# Hypocrates is a genetically encoded fluorescent biosensor for (pseudo)hypohalous acids and their derivatives

Alexander I. Kostyuk [1,2,3,16], Maria-Armineh Tossounian [4,5,6,14,16], Anastasiya S. Panova[1,2,3], Marion Thauvin [7,8], Roman I. Raevskii[1], Daria Ezeriņa [4,5,6], Khadija Wahni[4,5,6], Inge Van Molle [4,6], Anastasia D. Sergeeva[1,9], Didier Vertommen [10], Andrey Yu. Gorokhovatsky[1], Mikhail S. Baranov[1,11], Sophie Vriz [7,12,15], Joris Messens [4,5,6✉], Dmitry S. Bilan [1,2,3✉] & Vsevolod V. Belousov[1,2,3,13✉]

The lack of tools to monitor the dynamics of (pseudo)hypohalous acids in live cells and tissues hinders a better understanding of inflammatory processes. Here we present a fluorescent genetically encoded biosensor, Hypocrates, for the visualization of (pseudo) hypohalous acids and their derivatives. Hypocrates consists of a circularly permuted yellow fluorescent protein integrated into the structure of the transcription repressor NemR from *Escherichia coli*. We show that Hypocrates is ratiometric, reversible, and responds to its analytes in the $10^6\,M^{-1}s^{-1}$ range. Solving the Hypocrates X-ray structure provided insights into its sensing mechanism, allowing determination of the spatial organization in this circularly permuted fluorescent protein-based redox probe. We exemplify its applicability by imaging hypohalous stress in bacteria phagocytosed by primary neutrophils. Finally, we demonstrate that Hypocrates can be utilized in combination with HyPerRed for the simultaneous visualization of (pseudo)hypohalous acids and hydrogen peroxide dynamics in a zebrafish tail fin injury model.

[1] Shemyakin-Ovchinnikov Institute of Bioorganic Chemistry, 117997 Moscow, Russia. [2] Center for Precision Genome Editing and Genetic Technologies for Biomedicine, Pirogov Russian National Research Medical University, 117997 Moscow, Russia. [3] Laboratory of Experimental Oncology, Pirogov Russian National Research Medical University, 117997 Moscow, Russia. [4] VIB-VUB Center for Structural Biology, Vlaams Instituut voor Biotechnologie, B-1050 Brussels, Belgium. [5] Brussels Center for Redox Biology, Vrije Universiteit Brussel, B-1050 Brussels, Belgium. [6] Structural Biology Brussels, Vrije Universiteit Brussel, B-1050 Brussels, Belgium. [7] Center for Interdisciplinary Research in Biology (CIRB), Collège de France, CNRS, INSERM, PSL Research University, Paris 75231, France. [8] Sorbonne Université, Collège Doctoral, Paris 75005, France. [9] Biological Department, Lomonosov Moscow State University, 119992 Moscow, Russia. [10] de Duve Institute, MASSPROT platform, UCLouvain, 1200 Brussels, Belgium. [11] Laboratory of Medicinal Substances Chemistry, Pirogov Russian National Research Medical University, 117997 Moscow, Russia. [12] Université de Paris, Paris 75006, France. [13] Federal Center of Brain Research and Neurotechnologies, Federal Medical Biological Agency, 117997 Moscow, Russia. [14] Present address: Department of Structural and Molecular Biology, University College London, London WC1E 6BT, United Kingdom. [15] Present address: Laboratoire des biomolécules, LBM, Département de chimie, École normale supérieure, PSL University, Sorbonne Université, CNRS, 75005 Paris, France. [16] These authors contributed equally: Alexander I. Kostyuk, Maria-Armineh Tossounian. ✉email: joris.messens@vub.be; d.s.bilan@gmail.com; belousov@fccps.ru

The past decades have considerably increased our knowledge on the role of reactive oxygen species (ROS), particularly of hydrogen peroxide ($H_2O_2$), in physiological and pathophysiological processes. Indeed, $H_2O_2$ went from being perceived exclusively as an oxidative stress molecule to a secondary messenger compound that regulates cellular signal transduction pathways by modifying cysteine residues in proteins[1–3]. Other oxidants of physiological relevance include (pseudo)hypohalous acids, known to participate in immune response reactions. However, specific details regarding their spatio-temporal dynamics and contribution to various aspects of metabolism remain largely unexplored. (Pseudo)hypohalous acids, such as hypochlorous acid (HOCl), hypobromous acid (HOBr), and hypothiocyanous acid (HOSCN), are produced by a number of mammalian peroxidases: myeloperoxidase (MPO), eosinophil peroxidase (EPO), and lactoperoxidase (LPO)[4]. These enzymes catalyze the conversion of (pseudo)halide ions ($X^-$) to $OX^-$ in the presence of $H_2O_2$. The formed HOCl and HOBr can react with nucleophiles containing nitrogen and sulfur atoms, for example, with amines and thiols, as well as with aromatic rings and unsaturated bonds in organic molecules. In a cellular context, this means that possible targets for HOCl and HOBr include different amino acids in proteins, glutathione, lipids, carbohydrates, and nucleobases[5–8]. As already mentioned, (pseudo)hypohalous acids are one of the key players in immune system functioning. Reactive chlorine species generated by MPO from neutrophils provide defense against bacterial and fungal infections[9,10]. HOBr produced by EPO from eosinophils is a powerful agent used for the destruction of larger parasites[11], while HOSCN from LPO acts as a potent antibacterial compound primarily in saliva, tears, milk, and airways[12,13].

Levels of (pseudo)hypohalous acids need to be controlled, as their increase is often associated with pathological conditions such as atherosclerosis, diseases of the cardiovascular system and lungs, autoimmune diseases, Alzheimer's disease, and many others[14–17]. However, our knowledge about the precise mechanisms of their regulation, and, more broadly, of how they participate in cellular signaling, is rather limited. It is important to note that HOCl/HOBr and HOSCN differ markedly in their reactivity, with HOSCN being a much weaker oxidant and displaying a pronounced selectivity for thiols[18] and selenium-containing species[19]. Moreover, the exact role of N-chloramines, which are milder and longer-lived oxidants compared to HOCl that result from the reaction of amines with HOCl, is also not clear[20–22]. In light of the above, HOSCN and N-chloramines seem to be the best candidates to act as signal transmitters. Derivatives of (pseudo)hypohalous acids differ not only in their selectivity but also in stability. For example, it has been proposed that RS-Cl intermediates formed on cysteine (Cys) residues are characterized by short lifetimes and quickly undergo hydrolysis[23]. In contrast, the RS-SCN intermediates are much more stable with characteristic lifetimes of several minutes or even hours[23,24]. Thus, changes in the ratio between different (pseudo)hypohalous acids produced in cells and tissues might shape the spatial and temporal patterns of oxidative stress.

The most frequently used approaches to study hypohalous stress in tissues are by measuring MPO enzymatic activity with colorimetric methods or immunohistochemical visualization of MPO localization. In addition, mass spectrometry and gas chromatography allow the identification of compounds that have undergone halogenation as a result of hypohalous stress. A significant drawback of these approaches is that they do not monitor real-time dynamics within live cells. Therefore, over the past few years, the market has seen a large number of fluorescent dyes for measuring HOCl[25–31], and some selective synthetic indicators for HOBr detection[32]. Despite all the advantages of these dyes, they have several shortcomings compared to genetically encoded biosensors based on fluorescent proteins, which are able to follow their analyte in living systems of any level of complexity. It is generally accepted and safe to say that such biosensors have revolutionized research in redox biology[33]. In particular, probes of the HyPer family contributed to our understanding of the biological role of $H_2O_2$ by revealing its sites of production, intracellular trafficking, and dynamics, as well as modes of regulation[34–38]. The development of a similar genetically encoded protein-based biosensor for the visualization of (pseudo)hypohalous acids was thought to be an impossible task due to the high reactivity of these compounds and their low target selectivity. However, today, there is evidence that similarly to $H_2O_2$, the 'aggressive' oxidant HOCl is also involved in the regulation of proteins, modifying specific amino acid residues in both eukaryotic and prokaryotic organisms[39–43].

Here we set out to engineer a genetically encoded biosensor for the visualization of (pseudo)hypohalous acids in live cells and in vivo models. We base our biosensor on a circularly permuted yellow fluorescent protein (cpYFP) integrated into the modified structure of the E. coli transcription repressor NemR. Wild-type NemR contains several redox-active Cys residues and is sensitive to reactive chlorine species, including HOCl and chloramines[42,44], as well as to some electrophiles[45]. In previous work, it has been suggested that NemR oxidation with reactive chlorine species leads to the formation of a reversible sulfenamide bond between Cys106 and Lys175, which induces a local minor change in the protein conformation[42,44]. Therefore, to develop a biosensor specific for HOCl, we use NemR with only a single cysteine, Cys106 (NemR$^{C106}$). This biosensor displays specificity for (pseudo)hypohalous acids and is named Hypocrates (from Hypochlorite Ratiometric Sensor). This name is a homophone for the name of the "Father of Medicine", Hippocrates - the great ancient Greek physician who was one of the first to describe signs of inflammation and to reflect on the nature of this process.

## Results

**Hypocrates (NemR-cpYFP biosensor) architecture and design.** As a first step, we searched for prokaryotic transcription factors which sense hypochlorite anions ($ClO^-$). Out of all possible candidates, HypR and NemR were selected[42,43]. For our work, a NemR mutant with all Cys residues substituted for Ser, except for Cys106 (NemR$^{C106}$)[42], was used to avoid undesirable sensitivity for reactive electrophilic species (RES) and to minimize other non-specific redox reactions, such as disulfide-linked dimerization[45]. To determine which of the two candidates was more suited for our biosensor design, we measured the second-order rate constants of NemR$^{C106}$ and HypR by monitoring the change of their respective intrinsic tyrosine and tryptophan fluorescence with increasing NaOCl concentrations (Supplementary Fig. 1). We found that NemR$^{C106}$ ($\sim 3.0 \times 10^5$ $M^{-1}s^{-1}$) reacts ~200-times faster compared to HypR ($\sim 1.5 \times 10^3$ $M^{-1}s^{-1}$), while exposure to $H_2O_2$ had no effect in both cases (Supplementary Fig. 2). The negative y-intercept might be the result of several chlorination/oxidation events with some of these modifications not directly being within the microenvironment of the tryptophan, which was used as the reporter amino acid for determining the second-order rate constant of NemR$^{C106}$.

Based on these observations, we selected NemR$^{C106}$ as a molecular platform to design a biosensor for HOCl detection. NemR$^{C106}$ consists of a DNA-binding and a sensory domain (Fig. 1a). The latter has a flexible loop with the crucial Cys106 buried in a hydrophobic pocket surrounded by Trp167, Leu168, and Val109. Notably, its measured reaction rate with NaOCl is ~1000 times slower compared to what has been published for a

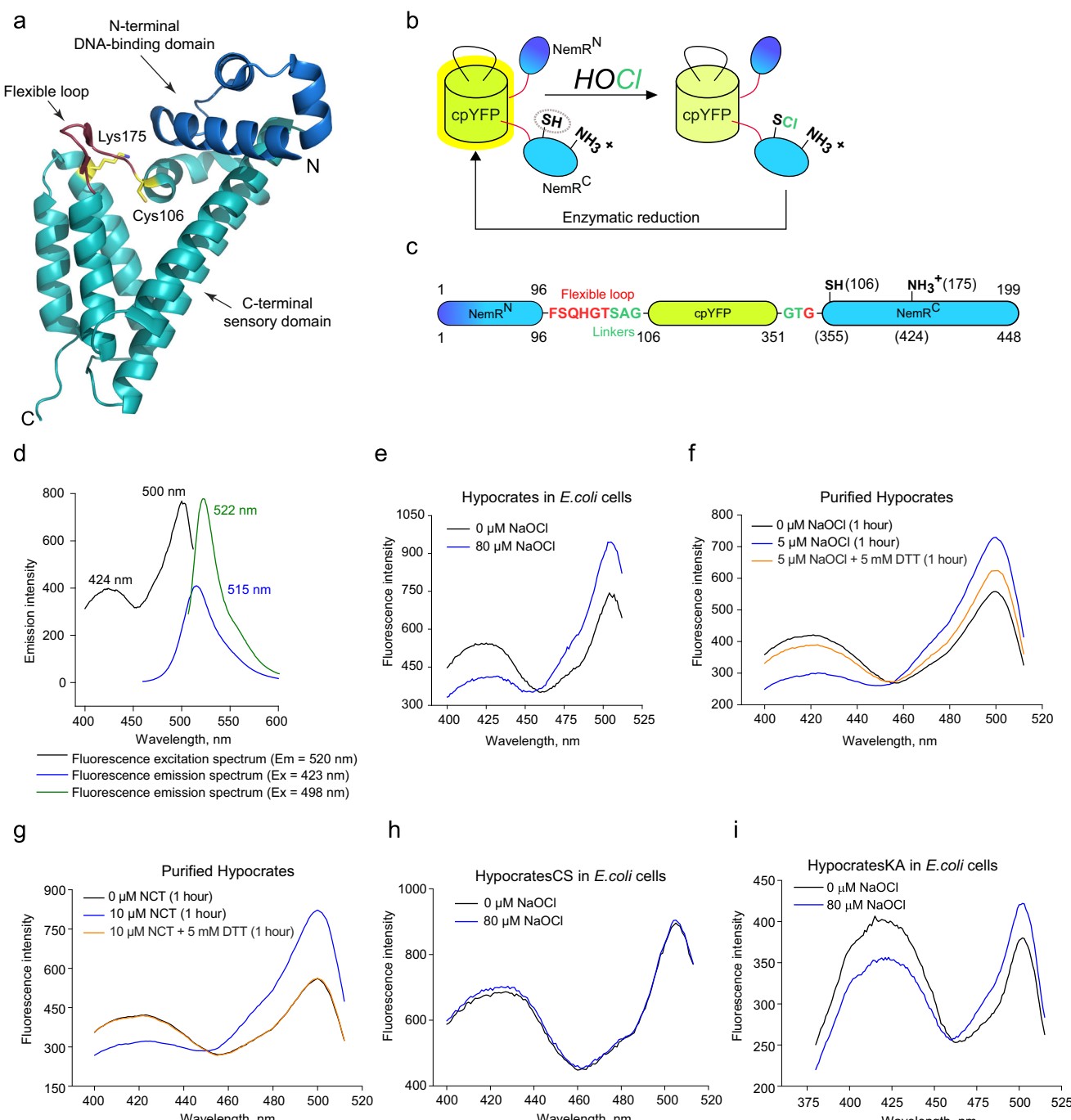

**Fig. 1 Hypocrates (NemR-cpYFP biosensor) design and spectral characteristics. a** The structure of NemR[C106] (PDB ID: 4YZE) shows the N-terminal DNA-binding domain (colored blue), the C-terminal sensory domain (colored cyan), Cys106 and Lys175 (colored yellow), and the flexible loop (colored red) into which cpYFP was inserted. The N- and C-termini are indicated with N and C, respectively. **b** The proposed simplified scheme of NemR-cpYFP biosensors functioning in living cells. **c** The structure of Hypocrates is presented with NemR[C106] colored blue/cyan, cpYFP colored yellow, the linkers between NemR[C106] and cpYFP colored green, and the flexible loop colored red. The upper numbers represent amino acid numbering corresponding to the intact NemR[C106], while the lower numbers represent numbering corresponding to the biosensor. **d** Optical properties of purified Hypocrates protein in PBS. **e** Hypocrates fluorescence excitation spectra in *E. coli* cells in reduced and NaOCl-oxidized forms. **f** Purified Hypocrates (0.5 µM) fluorescence excitation spectrum behavior in the presence of NaOCl in saturating concentration. **g** Purified Hypocrates (0.5 µM) fluorescence excitation spectrum behavior in the presence of *N*-chlorotaurine (NCT) in saturating concentration. **h** HypocratesCS fluorescence excitation spectra in *E. coli* cells in reduced and NaOCl-oxidized forms. **i** HypocratesKA fluorescence excitation spectra in *E. coli* cells in reduced and NaOCl-oxidized forms. Source data are provided as a Source Data file.

free cysteine[46], which indicates that the local environment strongly determines its reactivity. Cys106 has been observed in two conformations in the crystallographic asymmetric unit, with the transition reportedly caused by HOCl[42] (Supplementary Fig. 3). To visualize these conformational changes, we introduced cpYFP at several positions within the flexible loop. Similar to other cpYFP-based probes[34,38,47], we expected that, upon proper conformational coupling, structural shifts in the flexible loop induced by the reaction with HOCl would alter the optical properties of the chromophore (Fig. 1b).

This yielded 12 chimeras, in which cpYFP was separated from NemR[C106] by a variation of short linkers (SAG/G or SAG/GT) (Supplementary Fig. 4a). We hypothesized that shortening of the cpYFP integration region would improve the conformational coupling and, therefore, signal transmission from the sensory to the reporter unit of the sensor. Therefore, we designed four additional variants with one or two amino acid deletions in the flexible loop.

We recombinantly expressed all chimeras in *E. coli* and tested the changes in their fluorescence excitation spectra by adding NaOCl to the bacterial suspensions as well as purified recombinant proteins (Supplementary Fig. 4b). We selected the variant that had the maximum response amplitude of ~1.6-fold for further studies and named it Hypocrates (Fig. 1c) (Supplementary Fig. 4c). Purified recombinant Hypocrates protein is characterized by two excitation maxima (~425 nm and ~500 nm) and one fluorescence emission (~518 nm) maximum (Fig. 1d). The estimated brightness of the biosensor lies in the ~4400–13,900 range, depending on both the excitation wavelength and the redox state, which is ~7–22% of the Enhanced Yellow Fluorescent Protein (EYFP) brightness (Supplementary Table 1). The addition of NaOCl to *E. coli* cells expressing Hypocrates results in a ratiometric change in the fluorescence excitation spectrum (Fig. 1e). Thus, the biosensor signal can be expressed as an $Ex_{500}/Ex_{425}$ ratio. Recombinant purified Hypocrates protein behaves similarly and the ratiometric response can be almost completely reversed by the addition of a reducing agent (Fig. 1f). Treatment of the purified protein with NaOBr, another hypohalite salt, elicited a comparable signal shift of the probe (Supplementary Fig. 5). Next, we decided to test whether Hypocrates is also sensitive to the HOCl-derivative, *N*-chlorotaurine (NCT). Taurine is one of the most abundant amino acids in many tissues[48]. This is especially the case in neutrophils, making NCT one of the most common derivatives of reactive chlorine species in that cell type[49]. We demonstrated that NCT treatment led to a fully reversible ratiometric response of the protein (Fig. 1g). HOSCN, a pseudohypohalous acid, which is chemically similar to HOCl and HOBr, but acts as a weaker and more selective oxidizing agent, demonstrated comparable behavior (Supplementary Fig. 5). Thus, Hypocrates can be considered as a biosensor for (pseudo)hypohalous acids and their derivatives.

If Hypocrates works according to the above-proposed principles, then substitution of the key Cys355 (Cys106 in NemR) residue for a non-reactive Ser should disrupt its sensing mechanism. We constructed this mutant version and named it HypocratesCS. As expected, *E. coli* cells expressing HypocratesCS did not respond anymore to the addition of NaOCl (Fig. 1h). To confirm the formation of the sulfenamide bond between Lys424 (Lys175 in NemR) and Cys355, we engineered a HypocratesKA variant in which Lys424 was replaced by Ala. However, we found that HypocratesKA is still highly sensitive to NaOCl (Fig. 1i). Apparently, no sulfenamide bond is formed in Hypocrates during oxidation. This was also confirmed by mass spectrometry on Hypocrates protein incubated with NaOCl, NCT, HOSCN, and NaOBr. The presence of a sulfenamide bond would induce a trypsin miscleavage at the Lys leading to the formation of a longer peptide. The theoretical mass of this peptide (after removing 2 protons due to the formation of a sulfenamide bond) was not observed at the MS1 and MS2 levels in any of these conditions. All in all, the experimental data indicate that the presence of the key Cys355 is required for hypohalous acid response, while the formation of a sulfenamide with Lys residues (Lys424 and Lys359) located in the direct structural environment of Cys355 does not take place.

**The selectivity of Hypocrates.** We showed that Hypocrates is highly sensitive to (pseudo)hypohalous acids and their derivatives. It is noteworthy that high concentrations of NaOCl and NaOBr (~100 μM at a protein concentration of 0.5 μM), but not NCT, led to pronounced fluorescence quenching due to apparent protein damage, which further indicates that NCT reacts with the sensor in a more specific way (Supplementary Fig. 5). In addition, we investigated whether global structural changes of Hypocrates occurred in the presence of NaOCl using circular dichroism (CD). After adding NaOCl to the biosensor, we observed an increase of the molar ellipticity [θ] at 208 nm and 222 nm and a decrease at 194 nm (Fig. 2a). Upon $H_2O_2$ addition, no CD spectral changes were observed (Supplementary Fig. 6). To test whether the optical shift could be restored, we incubated the NaOCl-oxidized Hypocrates with the reducing agent DTT. After 5 min incubation with DTT, the spectrum of the oxidized biosensor showed a similar pattern as the one of the reduced form (Fig. 2b), indicating the reversibility of the structural changes.

As for other cpYFP-based biosensors (except HyPer7[38]), the ratiometric response of Hypocrates is pH-dependent (Fig. 2c). The pKa of purified Hypocrates is 9.10, and HypocratesCS has a pKa of 9.20. In the presence of NaOCl and NCT, the pKa of the sensor decreases to 8.90 and 8.84, respectively. We measured the ratiometric signal of Hypocrates in buffer solutions with different acidity in the function of increasing NCT concentration. Within the physiological range of pH fluctuations, the shape of the titration curve is stable (Supplementary Fig. 7). However, a shift in pH from 6 to 8 results in a 12-fold signal increase; therefore, appropriate pH controls, such as HypocratesCS, are required for proper data interpretation.

Next, we tested the selectivity of Hypocrates. We incubated the sensor with aliquots of various common oxidants (Fig. 2d). Only minor signal fluctuations were observed in the presence of high concentrations of $H_2O_2$, xanthine oxidase/xanthine system ($O_2^{\bullet-}$ generator), MAHMA NONOate ($NO^{\bullet}$ generator), and GSSG (Fig. 2d). However, Hypocrates showed a ratiometric response to $ONOO^-$ (Fig. 2d) (Supplementary Fig. 5). Therefore, if pronounced reactive nitrogen species production in the system can be expected, implementation of appropriate controls might become necessary. As active electrophiles have been shown to affect the DNA-binding affinity of NemR, as well as the expression of the nemRA–gloA operon[45], we also tested whether glyoxal, formaldehyde, and methylglyoxal would elicit the response of Hypocrates (Fig. 2d). We found that Hypocrates is insensitive to these compounds.

To validate whether the NaOCl-induced fluorescence changes are NemR[C106]-derived, we treated purified cpYFP (0.5 μM) and two other cpYFP-based biosensors (HyPer-2[35] and SypHer3s[50], 0.5 μM) with NaOCl (5–10 μM). HyPer-2 and SypHer3s showed no response. At the same concentration of NaOCl (5 μM), a slight decrease in the intensity of cpYFP fluorescence was detected. This can be explained by bleaching since a single cpYFP has a more open conformation, making the chromophore more accessible to the environment. With a significant increase in the concentration of NaOCl (up to 105 μM), the fluorescence is almost completely quenched. This change in the signal is practically irreversible as

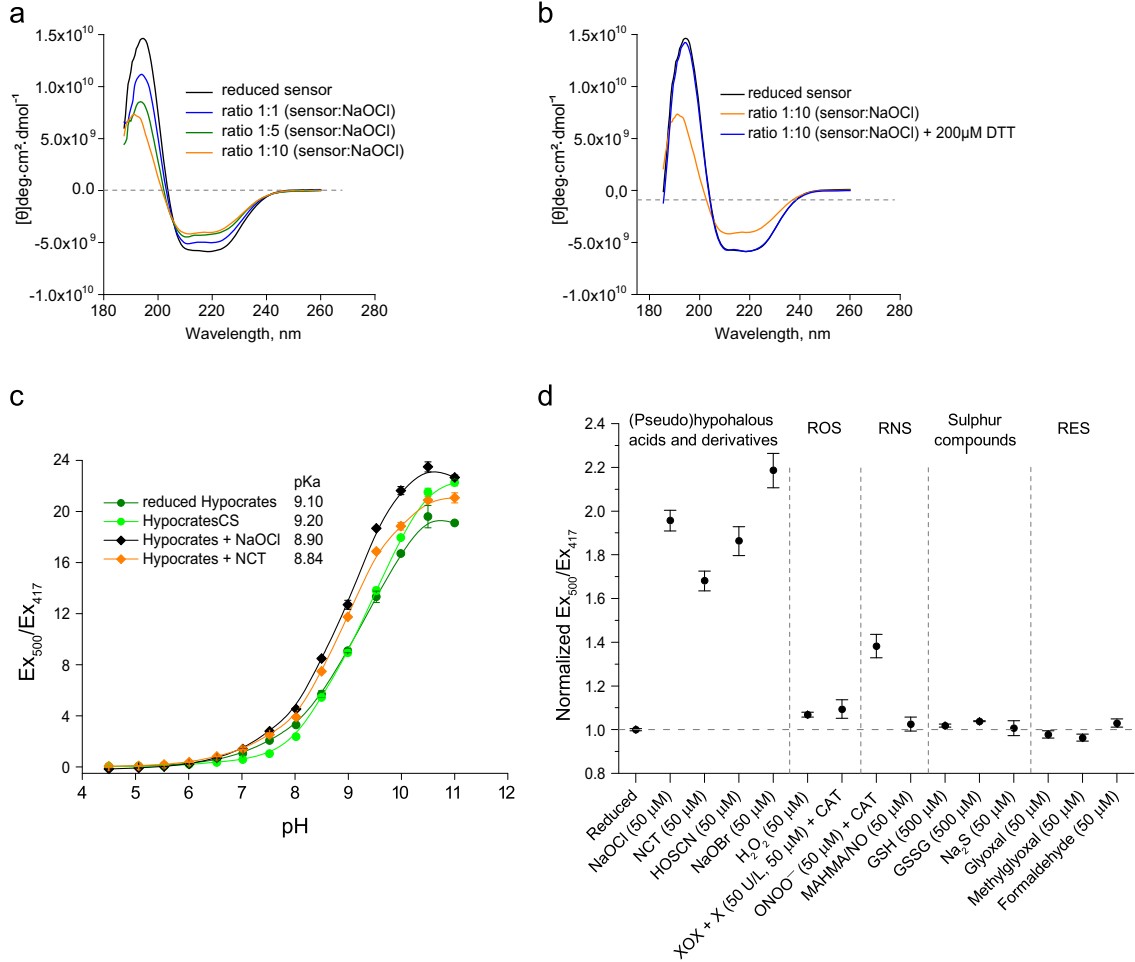

**Fig. 2 The selectivity of Hypocrates. a** Far-UV circular dichroism spectra of reduced and NaOCl oxidized Hypocrates. With increasing NaOCl concentration, an increase of the molar ellipticity [θ] at 208 nm and 222 nm and a decrease at 194 nm were observed. **b** Upon reduction with DTT, the NaOCl-treated biosensor restores its overall secondary structure to the reduced form. **c** The excitation ratio of reduced and oxidized Hypocrates depends on the pH value of the buffer solution. NCT – N-chlorotaurine. The data are presented as a mean ± SEM, $n = 4$ for reduced Hypocrates, $n = 3$ for other curves. **d** Selectivity profile of purified Hypocrates ($2\,\mu M$) towards a set of various redox compounds. ROS - reactive oxygen species, RNS - reactive nitrogen species, RES - reactive electrophilic species, XOX - xanthine oxidase, X - xanthine, CAT - catalase. The data are presented as the mean ± SEM, $n = 8$ for NaOCl, $n = 5$ for NCT, $n = 6$ for ONOO$^-$, $n = 3$ for other compounds. Source data are provided as a Source Data file.

shown by incubation with DTT, which indicates that non-specific degradation of the fluorescent protein proceeds under the tested conditions (Supplementary Fig. 8). As such, we concluded that cpYFP itself does not contribute to the ratiometric response. To obtain more direct evidence that the generated signal is NemR$^{C106}$-derived, the intrinsic Trp fluorescence change of NemR$^{C106}$ ($2\,\mu M$) was determined in the presence of oxidizing agents ($50\,\mu M$) (Supplementary Fig. 8e). The oxidants NaOCl, NCT, and NaOBr caused Trp-fluorescence shifts ($\lambda_{ex} = 295\,nm$, $\lambda_{em} = 350\,nm$), while HOSCN induced no response. This indicates that the NemR-derived domain in Hypocrates gained the ability to sense this compound due to the integration of cpYFP. This change in Trp-fluorescence is not related to direct oxidation of the Trp residue, which we could confirm with mass spectrometry (Supplementary Table 2).

**Hypocrates sensitivity and reaction rates.** The sensitivity of Hypocrates towards different oxidants (NaOBr, NaOCl, NCT, and HOSCN) was studied by titrating the biosensor ($0.5\,\mu M$) with increasing analyte concentrations (up to $15\,\mu M$) in sodium phosphate buffer (Fig. 3a–d). Hypocrates reaches saturation at $\sim 4$–$5\,\mu M$ (8–10:1 oxidant/sensor ratio) in the cases of NaOBr,

NaOCl, and NCT, and at $\sim 1\,\mu M$ (2:1 oxidant/sensor ratio) in the case of HOSCN, which might be connected to its higher selectivity towards Cys355. The Ex$_{500}$/Ex$_{417}$ ratio stabilizes at response values of $\sim 1.8$-fold (for NaOCl), $\sim 2.0$-fold (for NaOBr), $\sim 1.8$-fold (for HOSCN), and $\sim 1.7$-fold (for NCT) under saturating conditions. The highest sensitivities were observed for HOSCN and NaOBr, which were $\sim 1.11\,\mu M^{-1}$ and $\sim 0.60\,\mu M^{-1}$, respectively. This outcome shows that the biosensor is slightly more sensitive to HOSCN and NaOBr than to NaOCl and NCT. To estimate corresponding limits of detection (LOD), we implemented the $3S_{y|x}/b$ approach, where $S_{y|x}$ is the residual standard error and $b$ is the slope of the linear regression model. In the described system, the LOD values are $\sim 100\,nM$ for NaOBr, $120\,nM$ for HOSCN, $290\,nM$ for NCT, and $330\,nM$ for NaOCl.

To compare the reaction rates of Hypocrates towards NaOBr, NaOCl, and NCT, the second-order rate constants were measured on a stopped-flow instrument (Fig. 3e–g). We found that the biosensor reacts $\sim 100$-fold faster with NaOBr ($\sim 4.5 \times 10^6\,M^{-1}s^{-1}$) and NaOCl ($\sim 1.4 \times 10^6\,M^{-1}s^{-1}$) compared to NCT ($\sim 6.1 \times 10^4\,M^{-1}s^{-1}$). NemR$^{C106}$ is also less reactive to NCT compared to NaOCl (Supplementary Fig. 9), which can possibly be explained by the fact that NCT is a less aggressive compound. A previous study[21] showed that NCT reacts with glyceraldehyde-3-phosphate dehydrogenase at

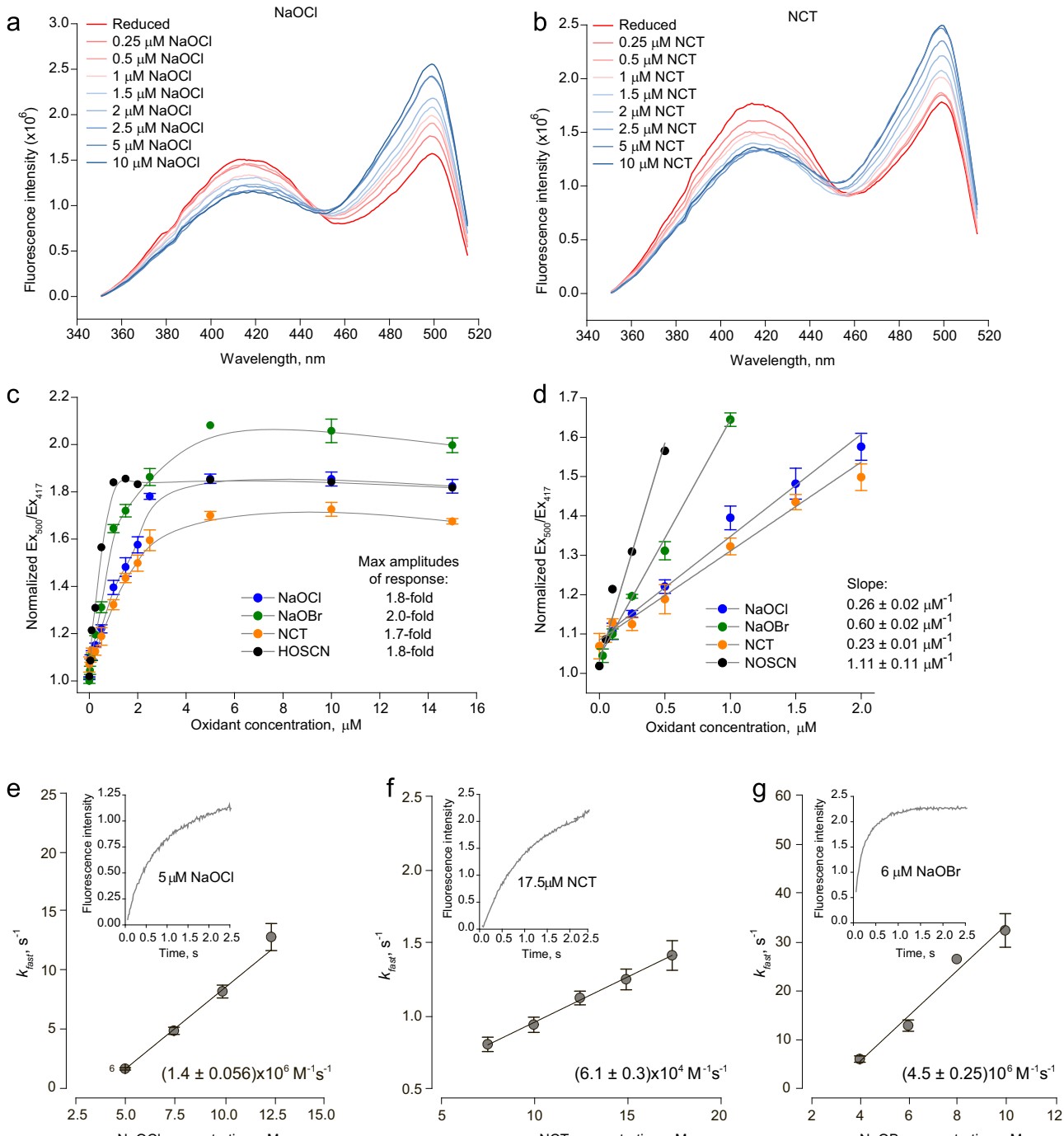

**Fig. 3 Hypocrates sensitivity and reaction rates.** Changes in the fluorescence excitation spectra of Hypocrates (0.5 μM) obtained by additions of **a** NaOCl or **b** N-chlorotaurine (NCT) aliquots. **c** Titration curves of Hypocrates (0.5 μM) in sodium phosphate buffer obtained by additions of NaOCl, NaOBr, HOSCN, or NCT aliquots. The data are presented as a mean ± SEM (for $n > 2$), $n \geq 2$. The maximum amplitudes of response are 2.0-, 1.8-, 1.8- and 1.7-fold for NaOBr, HOSCN, NaOCl, and NCT, respectively. In the presence of NaOBr, NaOCl, and NCT, the probe is saturated at ~5 μM, and for HOSCN at ~1 μM. **d** The same data as in **c**, Hypocrates sensitivity towards NaOCl, NaOBr, NCT, and HOSCN is shown. The data are presented as a mean ± SEM (for $n > 2$), $n \geq 2$. **e**–**g** Hypocrates reaction rates. Changes in cpYFP fluorescence at > 515 nm cut-off ($\lambda_{ex} = 485$ nm) were measured as a function of time (insert). The curves were fitted to a double exponential to obtain the observed rate constants ($k_{obs/fast}$), which were plotted as a function of different **e** NaOCl, **f** NCT, or **g** NaOBr concentrations. The second-order rate constants for NaOCl ($1.4 \pm 0.056) \times 10^6$ M$^{-1}$s$^{-1}$, NCT ($6.1 \pm 0.3) \times 10^4$ M$^{-1}$s$^{-1}$, and NaOBr ($4.5 \pm 0.25) \times 10^6$ M$^{-1}$s$^{-1}$ were determined from the slope of the straight line [$k_{fast} = k_{on} \cdot$[oxidant] + $k_{off}$]. The data are presented as a mean ± SD (for $n > 2$), $n \geq 2$. Source data are provided as a Source Data file.

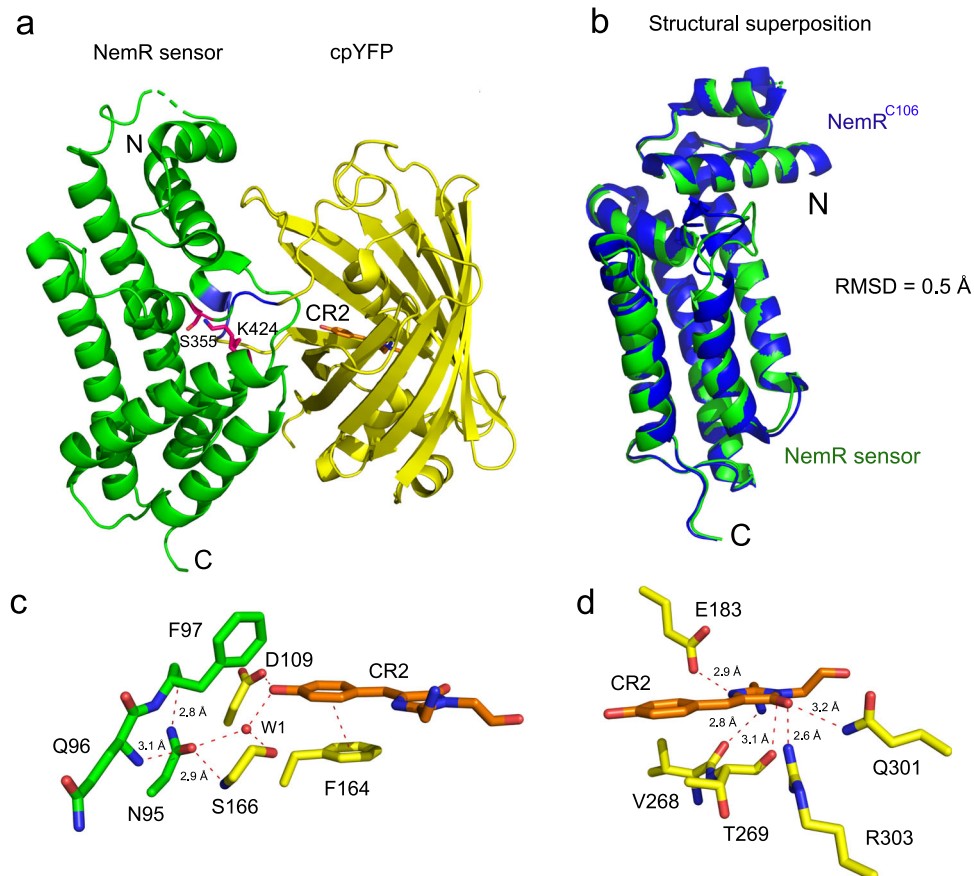

**Fig. 4 The structure of HypocratesCS, a cpYFP-based biosensor (PDB ID: 6ZUI). a** The NemR-sensor domain (green) and the cpYFP domain (yellow) are shown. The chromophore (CR2) in the cpYFP β-barrel is shown in orange stick representation. S355 (to which the reactive C355 was mutated to) and K424 (two orientations of the side chain) are shown in red stick representation. The linkers "SAG" and "GT" are colored blue. The N-terminal residues 1–7 and segments (residues 40-42 and 193–207) are missing. **b** Superposition of NemR^C106 (blue – PDB ID: 4YZE) with the NemR-sensor domain (green). RMSD - root-mean-square deviation. **c** N95 connects the sensor domain (green) with cpYFP (yellow). N95 interacts with the backbone of Q96, F97 of the NemR-sensor domain and with S166 of the cpYFP domain. The 4-hydoxybenzyl group of CR2 interacts with the phenyl-ring of F164 over a distance of 3.8 Å. **d** The imidazolinone ring of CR2 interacts with R303, Q301, V268, E183, and T269.

$300\,M^{-1}s^{-1}$, and with creatine kinase at $1.2 \times 10^2\,M^{-1}s^{-1}$, making Hypocrates 100 times more efficient in recognizing HOCl-modified taurine.

By using the intensiometric cpYFP fluorescence as a read-out, we obtain a negative $k_{off}$ for NaOCl and NaOBr and relatively high $k_{fast}$ values, which could be explained by the chlorination/bromination on several amino acids and methionine oxidation observed by mass spectrometry (Supplementary Table 3). Especially Tyr106, located close to Cys355, might be involved in coupling the chlorination and bromination to the chromophore (Supplementary Fig. 10).

**X-ray structure of HypocratesCS.** To gain insights into the biosensor architecture and sensing mechanism, we next crystallized Hypocrates and HypocratesCS. Only HypocratesCS gave diffraction-quality crystals. The orthorhombic crystals (C222₁, $a = 90.23$, $b = 95.44$, $c = 106.25$, $\alpha = \beta = \gamma = 90°$) contain one molecule of the biosensor per asymmetric unit and diffract to a resolution of 2.2 Å (Supplementary Table 4). The structure (PDB ID: 6ZUI [https://doi.org/10.2210/pdb6ZUI/pdb]) was solved by molecular replacement, using *E. coli* NemR^C106 (PDB ID: 4YZE [https://doi.org/10.2210/pdb4YZE/pdb]) and the cpYFP-based calcium sensor (PDB ID: 3O77 [https://doi.org/10.2210/pdb3O77/pdb]) as search models. HypocratesCS consists of a NemR^C106S-based sensor domain (green) and a cpYFP domain

(yellow) that contains the p-hydroxybenzylidene-imidazolidinone chromophore (orange), designated as "CR2" in the PDB (Fig. 4a). Superposition of the NemR^C106 (PDB ID: 4YZE [https://doi.org/10.2210/pdb4YZE/pdb] – blue) and HypocratesCS (sensor domain – green) shows a similar structure with a root-mean-square deviation (rmsd) of 0.506 Å for 159 atoms (Fig. 4b). As such, the insertion of cpYFP had only a minor structural effect on the overall structure of the NemR-sensor domain. The structure of HypocratesCS shows an X-ray structure of a cpFP-based redox biosensor with an integrated cpYFP domain (Fig. 4c). The chromophore is in cis-configuration (Supplementary Fig. 11) and consists of an imidazolinone ring connected to a planar 4-hydroxybenzyl ring. The 4-hydroxybenzyl ring of the chromophore is stabilized by π-π stacking interaction with the phenyl-ring of Phe164, which has an off-centered parallel orientation to the 4-hydroxybenzyl ring. Aromatic ring stacking is also observed in YFP and in the cpYFP-based calcium sensor but is absent in GFP (Supplementary Fig. 12)[51–53]. The distance between the 4-hydroxybenzyl ring and the Phe164 phenyl ring is 3.9 Å. Further, the imidazolinone ring of the chromophore has several interactions with neighboring residues, suggesting that the chromophore will not change position during excited-state proton transfer (ESPT) (Fig. 4d).

Asn95 is the connecting residue between the NemR-sensor domain and cpYFP (PDB ID: 6ZUI [https://doi.org/10.2210/pdb6ZUI/pdb]). Asn95 is located on the connecting loop between

the α4 helix of the NemR-sensor domain and the N-terminus of the cpYFP domain. Asn95 interacts with the backbone of Gln96 and Phe97 in the NemR-sensor domain, and with Ser166, a conserved serine of the ESPT pathway in fluorescent proteins (Supplementary Fig. 12a). In the $Ca^{2+}$ sensor, Case16, the domain connecting residue is Ser24 (Supplementary Fig. 12c)[52].

Further inspection of the structure showed that non-covalent interface interactions between cpYFP β-barrel and the NemR-sensor domain are relatively limited. In the corridor between both domains, we noticed possible hydrophobic π-π interactions between the rings of Tyr67 and Tyr106, and between the rings of Phe184 and Phe429. Also, hydrogen bond distance interactions between the side chains of Gln91 and Gln165, Gln19 and Asn131, and between the side chain of Arg24 and the backbone oxygen of Glu133 were observed. Whether these interactions play a role in signal coupling between the sensor domain and the chromophore embedded within the β-barrel is not known and could form a topic for a detailed mechanistical study.

The ESPT pathways are different for cpYFPs, YFP, and GFP (Supplementary Fig. 12a–d)[51–53] and consist of a hydrogen-bonding network surrounding the chromophore, the conserved serine and glutamate residues, and the conserved water molecules W1 and W2. In cpYFP (PDB ID: 6ZUI [https://doi.org/10.2210/pdb6ZUI/pdb]), the phenol oxygen of CR2 interacts with W1 and with Asp109, which might stabilize a negative charge on the phenol oxygen, similar to the phenol oxygen in GFP, where Thr203 takes over the role of Asp109 (Supplementary Fig. 12a, d). Changing the position of Asn95 in the sensor domain could affect the pKa of the phenol oxygen via changes in the H-bond network in which W1, Ser166, and Asp109 are involved (Supplementary Fig. 12a), and this change could trigger a different charge transfer from the phenol oxygen via the phenol and imidazolinone rings with Glu183 as the final acceptor. The most likely final step of the pathway is a deprotonation of the heterocyclic ring nitrogen and the protonation of Glu183, rendering both groups neutral, but determining the exact details of the ESPT pathway and the potential role of W2 is beyond the scope of the present study.

As NemR was shown to form a homo-dimer[42], we decided to test whether the integration of the fluorescent protein into its structure affected the oligomeric state. Our gel filtration data indicate that Hypocrates is a strict homo-dimer (99.85 kDa), and that dimer formation is protein concentration and redox state-independent (Supplementary Fig. 13).

**Hypocrates performance in vitro and in eukaryotic cell culture.** We tested the ability of Hypocrates to visualize myeloperoxidase (MPO) activity in vitro. Incubation of the purified protein (0.5 μM) in the presence of the MPO-$H_2O_2$ system leads to a ratiometric response with an amplitude shift of 1.79-fold after 10 min of incubation, while $H_2O_2$ even at a physiologically irrelevant high concentration (100 μM) induces only a minor oxidation shift of ~1.1-fold (Fig. 5a, b).

To test whether Hypocrates is functional in a eukaryotic system, we expressed the sensor in HeLa Kyoto cells and visualized the signal using fluorescence microscopy. To evaluate the sensitivity of the probe, we tested increasing concentrations of NaOCl and calculated the response as an $Ex_{500}/Ex_{425}$ ratio. The minimal NaOCl addition that induced detectable changes of the sensor fluorescence was ~4.2 nmol/($10^5$ cells) (Fig. 5c). Exposure to 17 nmol/($10^5$ cells) NaOCl led to a signal change of 1.8-fold, which is similar to the saturating response obtained with purified protein and in *E. coli* suspension (Fig. 1e, f). It should be noted that since NaOCl rapidly reacts with biological objects, in such experiments it is important to take into account not only the absolute concentration of the NaOCl solution but also the number of cells.

The oxidation of the biosensor in HeLa Kyoto cells was reversible – Hypocrates returned to the initial signal within ~3 min after NaOCl addition (Fig. 5d). In living cells, after reduction, the biosensor can be re-oxidized, which makes it possible to use Hypocrates as a reusable indicator (Supplementary Fig. 14a). We also transfected cells with HypocratesCS and with the specific pH-sensor SypHer3s[50]. Exposure to 17 nmol/($10^5$ cells) NaOCl did not significantly affect the signal of both probes (Supplementary Fig. 14b), indicating that the behavior of Hypocrates, observed in this system, specifically reflects a NaOCl-induced response.

To demonstrate the possibility of monitoring endogenous hypohalous stress with Hypocrates, we used human peripheral blood neutrophils, which contain a high level of MPO. For this, *E. coli* cells expressing the Hypocrates probe were added to the medium of freshly isolated neutrophils. We monitored changes of the fluorescent signal in the bacterial cytoplasm every time a cell was phagocytosed by a leukocyte. HypocratesCS also showed an increase in the signal, but to a lesser extent, which indicates that inside the neutrophils, in addition to hypohalous stress, bacterial cells experience an increase in cytoplasmic pH (Fig. 5e).

**Hypocrates monitors changes in the levels of hypohalous acids in vivo.** To test Hypocrates in vivo, we induced inflammation using tail fin amputation of zebrafish larvae. A similar experiment with the genetically encoded sensor HyPer[34] has previously shown that $H_2O_2$ concentration significantly increases in the wound margin and reaches its maximal value ~20 min post amputation[54]. As both $H_2O_2$ and hypohalous acids are involved in inflammation, we decided to simultaneously monitor the production of these compounds in this system. To this end, we combined Hypocrates with HyPerRed[37], a red sensor for $H_2O_2$ (Fig. 6). Signals of both indicators increased 15 min post amputation (mpa), then HyPerRed fluorescence decreased while the Hypocrates signal changed much slower. These differences in dynamics are expected, as neutrophils participate in the elimination of the $H_2O_2$ gradient due to the reaction catalyzed by MPO[55]. In parallel, we used HypocratesCS as control. Although HypocratesCS signal also increased, statistical analysis revealed that the difference between the responses of Hypocrates and HypocratesCS was significant. Thus, we demonstrated that Hypocrates is suitable for in vivo imaging with HypocratesCS as a control.

**Discussion**

Our idea of developing a genetically encoded sensor for (pseudo)hypohalous acids detection was born out of the increasing volume of incoming information about proteins that are specifically modified under conditions of hypohalous stress. We present the Hypocrates probe, which is a genetically encoded fluorescent biosensor for visualizing (pseudo)hypohalous acids in live systems. Hypocrates displays a ratiometric reversible change in signal when interacting with (pseudo)hypohalous acids and their derivatives with minimal response-inducing oxidant concentrations located in the 0.1–0.3 μM range (at a biosensor concentration of 0.5 μM). It is known that neutrophils produce high amounts of reactive chlorine species. However, it is difficult to calculate the exact concentrations of HOCl generated by cells since it quickly reacts with surrounding molecules. It has been reported that at the sites of inflammation, the concentration of HOCl generated by MPO from accumulated immune cells reaches levels of 1–2 mM[56]. Our biosensor demonstrates similar sensitivities to NaOCl, NaOBr, and HOSCN, which is why we are positioning Hypocrates as an indicator for visualizing the dynamics of the total pool of (pseudo)hypohalous acids. This is particularly relevant because (pseudo)hypohalous acids are

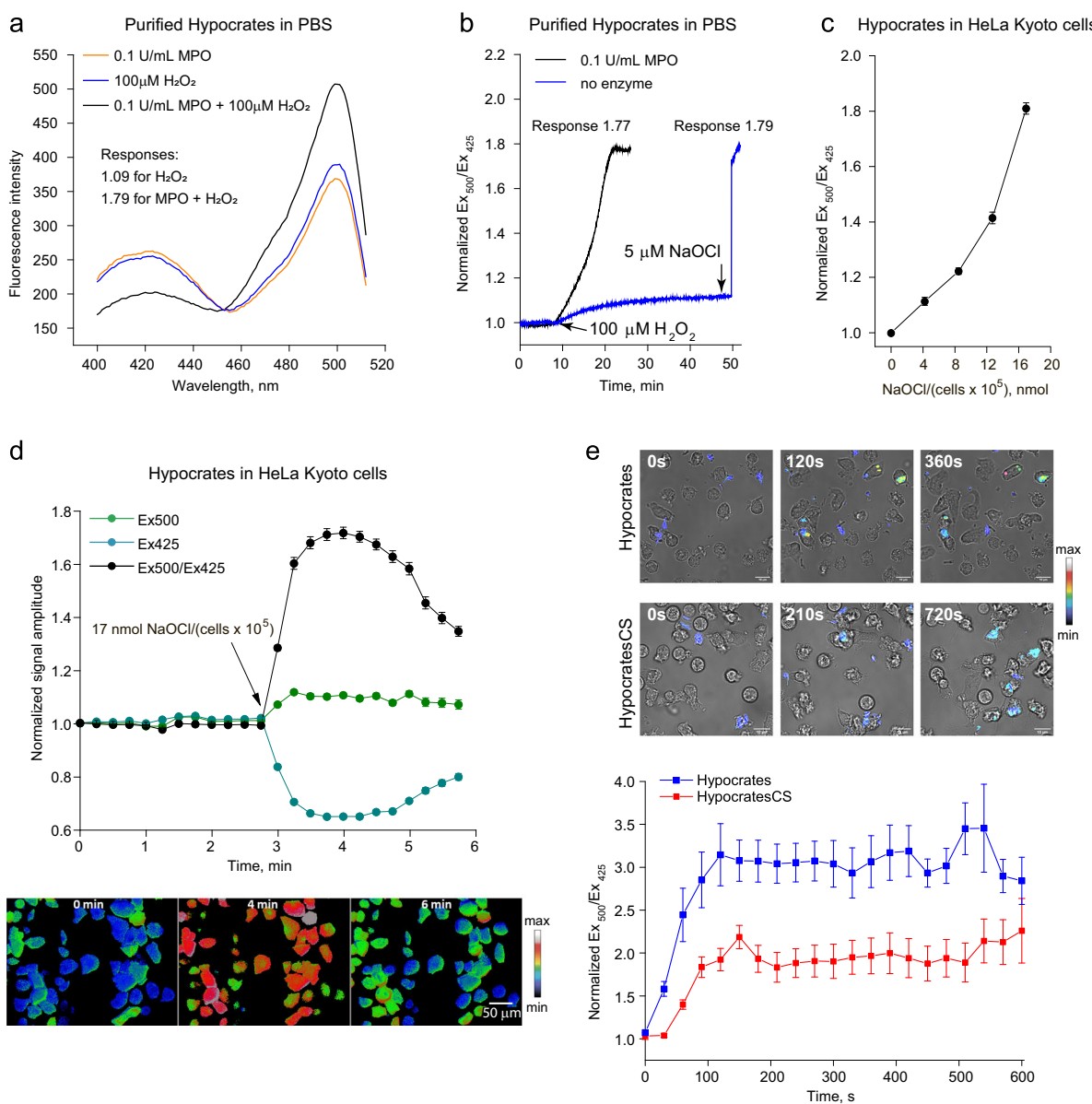

**Fig. 5 Hypocrates performance in vitro and in eukaryotic cell culture. a** Fluorescence excitation spectra of purified Hypocrates (0.5 μM) in the presence of individual myeloperoxidase (MPO), $H_2O_2$ and the MPO-$H_2O_2$ system. **b** Hypocrates (0.5 μM) signal as a function of time in the presence of individual $H_2O_2$ and the MPO-$H_2O_2$ system. HOCl, generated by MPO, leads to the development of a saturating response, while a physiologically irrelevant $H_2O_2$ concentration induces only minor signal changes. **c** The titration curve of Hypocrates in HeLa Kyoto cells exposed to different concentrations of NaOCl (values ± SEM, $N = 2$ experiments, $n \geq 25$ cells per experiment). **d** Upper part: Hypocrates fluorescence changes induced by 17 nmol NaOCl/($10^5$ cells) (values ± SEM, $N = 2$ experiments, $n \geq 30$ cells per experiment). Lower part: Images of Hypocrates in transiently transfected HeLa Kyoto cells exposed to 17 nmol/($10^5$ cells) NaOCl at different time points. Scale bar = 50 μm. The lookup table indicates changes in the $Ex_{500}/Ex_{425}$ ratio. **e** Upper part: Images of human neutrophils phagocytosing *E. coli* cells that express Hypocrates or the control version HypocratesCS. Scale bar = 10 μm. The lookup table indicates changes in the $Ex_{500}/Ex_{425}$ ratio. Lower part: Hypocrates (blue line) and HypocratesCS (red line) fluorescence changes in *E. coli* cells phagocytosed by human neutrophils. The starting point on the graph corresponds to the moment at which individual bacteria are phagocytosed by a neutrophil (values ± SEM, $N = 3$ experiments, 35 bacterial cells in total for each version of the sensor). Source data are provided as a Source Data file.

produced as a mixture in biological systems. Hypocrates also allows monitoring the dynamics of HOCl derivatives. Chloramines are characterized by longer lifetimes due to decreased reaction rates and altered selectivity profiles with more pronounced specificity for sulfur-containing groups. Due to the high concentrations of taurine in neutrophils, *N*-chlorotaurine (NCT) is one of the most common derivatives of reactive chlorine species[49]. It might seem that having a reaction rate of ~$10^6$ M$^{-1}$s$^{-1}$ for NaOCl, Hypocrates will not be able to visualize hypohalous stress under all conditions, since it could be outcompeted by

other sulfur-containing compounds, as some of them are present at millimolar concentrations and operate at $10^7$–$10^8$ M$^{-1}$s$^{-1}$. However, being an extremely reactive compound, HOCl hardly induces its biological effects directly. Apparently, HOCl is locally converted into less aggressive halamines, like NCT, and these compounds, in turn, might affect the cellular metabolism. Therefore, the ability of Hypocrates to sense NCT with a relatively high reaction rate makes it an efficient tool for the visualization of hypohalous stress. Recombinantly expressed and purified Hypocrates did not show any response to the major

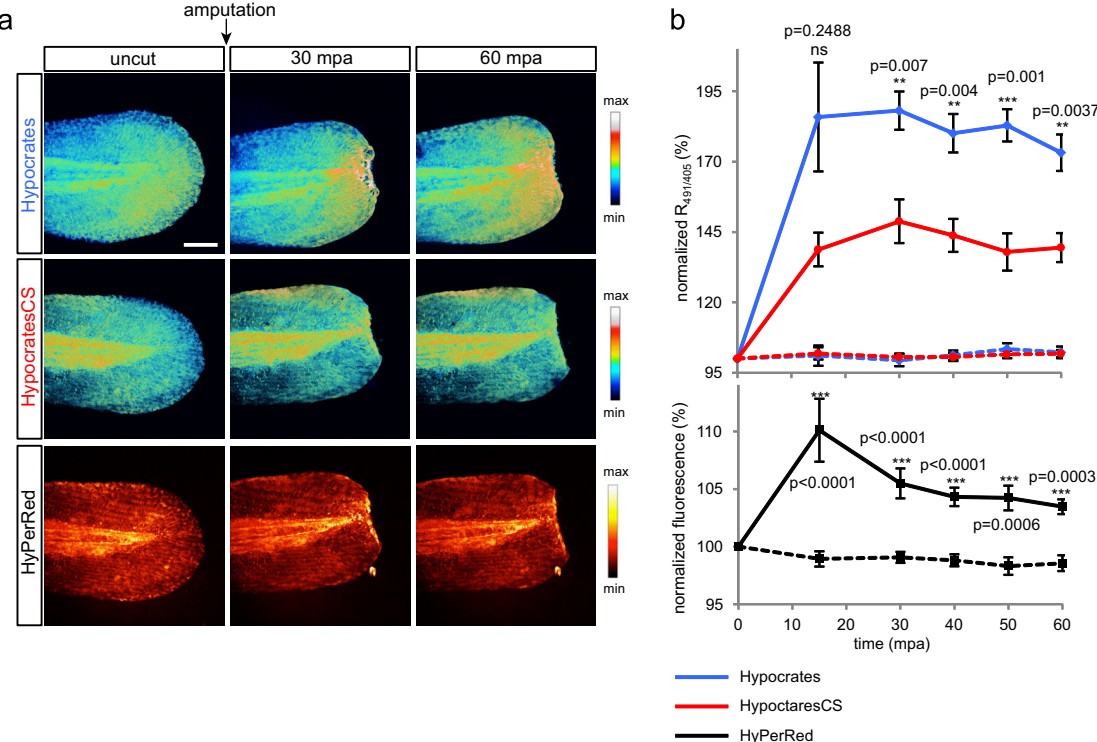

**Fig. 6 Hypohalous acids and H₂O₂ dynamics during zebrafish larvae wounding. a** Hypocrates and HyPerRed imaging. Zebrafish embryos were co-injected with Hypocrates or HypocratesCS and HyPerRed mRNAs at the 1-cell stage, and a tail fin amputation assay was performed on 48 h post-fertilization larvae. Images were taken before amputation, and time lapse imaging was performed up to 60 min post-amputation (mpa). The lookup tables indicate changes in the oxidation states of the sensors. Scale bar = 100 µm. **b** Hypocrates ratio and HyPerRed fluorescence were quantified at the amputation plane and normalized to the mean fluorescence on the uncut tail for each larva. Ratio quantification on larvae tail fin expressing Hypocrates (blue lines) or HypocratesCS (red lines) is shown. Two-way ANOVA test with a Tukey's multiple comparisons posttest was used to determine if the observed difference was statistically significant. Non-amputated embryos (dashed lines) expressing Hypocrates or HypocratesCS were also imaged as a control (values ± SEM; $N = 4$ experiments, $n \geq 3$ embryos/timepoint; ns no significant, **$P < 0.01$, ***$P < 0,001$, *versus* HypocratesCS cut larvae, $P$-values are shown). For control embryos (dotted lines), fluorescence has been normalized for each embryo to the first image of the time lapse ($t = 0$). HyPerRed fluorescence quantification on larvae tail fin expressing HyPerRed (black lines) is shown (values ± SEM, $N = 3$ experiments, $n \geq 7$ embryos/timepoint; ***$P < 0.001$, versus uncut larvae (dashed line), $P$-values are shown). Source data are provided as a Source Data file.

common intracellular oxidizing agents. However, we observed spectral changes of the sensor in the presence of ONOO⁻. The saturating concentration of ONOO⁻ that induced the maximum response of Hypocrates in vitro was ~10 µM at a biosensor concentration of 0.5 µM. Thus, in systems in which high ONOO⁻ production is expected, it is recommended to perform a control series of experiments, for example, using various inhibitors of NO• synthases, in order to assess the contribution of ONOO⁻ to the formation of the biosensor response. The acidity of the medium affects the acid-base equilibrium of the chromophores inside the fluorescent proteins; therefore, the relative abundance of their protonated and deprotonated forms largely depends on the pH of the solution. The dependence on H⁺ concentration is especially pronounced for circular permutants of fluorescent proteins, including cpYFP that we used, since their chromophores are more accessible to the environment. To inactivate the sensor, we substituted the key Cys355 residue with a Ser and obtained the HypocratesCS variant, which is insensitive to (pseudo)hypohalous acids and their derivatives. Since its spectral properties and pH sensitivity are identical to the parameters of the original indicator, HypocratesCS is the most appropriate pH control.

By analogy with the C355S mutation, we expected that a K424A substitution would completely inactivate Hypocrates. However, the HypocratesKA variant was found to act very similarly to the original sensor in terms of NaOCl sensitivity and response amplitude. This observation together with our mass

spectrometry data suggests that the NemR^C106-sensor domain of Hypocrates does not function in a similar way to the *E. coli* transcription repressor NemR in forming a reversible sulfena-mide bond[42,44], and that even the oxidation of Cys355 itself could be sufficient for signal changes in Hypocrates (Supplementary Fig. 10). We hypothesize that the oxidation of Cys355 (mass spectrometry data showed sulfinic and sulfonic acid formation – Supplementary Table 3), which is located on a flexible loop next to Gly354, couples to Gln99 and this residue, in turn, couples to Asn95, which is in contact with the 4-hydoxybenzyl group of the chromophore (CR2) via a water molecule. Hence, oxidation of Cys355 mediated by (pseudo)hypohalous acids might induce changes in the hydrogen-bonding network surrounding CR2 that lead to a ratiometric response. Next to Cys355 oxidation, our kinetic and mass spectrometric studies point in the direction of a potential alternative chlorination/bromination mechanism. A potential culprit residue in the active site of the sensing domain, located within a hydrogen-bonding distance of Cys355, is Tyr106 (Supplementary Fig. 10). This tyrosine is found to be chlorinated and brominated (Supplementary Table 3). It is tempting to speculate that the formation of chloramine on the flexible Lys424 (observed in two different side-chain orientations) via Cys355 could catalyze Tyr106 carbon 3 and carbon 5 chlorination[57–59]. To facilitate this process, a deprotonated sulfur of Cys355, located N-terminally of a short helix, and the Nε2 of His63 would help to delocalize electrons in

the direction of the OH of Tyr106 making both carbons on the Tyr phenol ring more susceptible for chlorination. This chlorination event takes place in a region of the sensing domain which could couple to the hydrogen-bonding network surrounding the chromophore resulting in a reversible ratiometric response. However, elucidating the precise sensing mechanism of Hypocrates is the subject of a separate study.

The crystal structure of HypocratesCS is a cpFP-based redox biosensor that reveals its CR2 chromophore environment within its overall structure (Fig. 4a,c). Overall structure comparison of the β-barrel of cpYFP in Hypocrates (PDB ID: 6ZUI [https://doi.org/10.2210/pdb6ZUI/pdb]) with the calcium biosensor, Case16 (PDB ID:3O77 [https://doi.org/10.2210/pdb3O77/pdb])[52], showed that these β-barrels are very similar (rmsd of 0.282 Å for 190 atoms). Further, both have a CR2 chromophore, while GFP is characterized by a CRO chromophore (Supplementary Fig. 12d). As such, the cpFP in Case16 is actually a cpYFP, and not a cpGFP as mentioned in Leder et al.[52]. The structure of HypocratesCS is not only important for understanding the functioning of this biosensor, but also for revealing the features of other cpYFP-based probes, which show subtle differences in their CR2 chromophore environment (Supplementary Fig. 12a, c), as well as for the rational structure-guided design of future cpYFP-based biosensors for other analytes.

As a defense mechanism against pathogens, immune cells generate hypohalous acids. We tested the functioning of Hypocrates by visualizing hypohalous stress in the bacterial cytoplasm of E. coli captured and swallowed by neutrophils. To accomplish this, E. coli cells expressing Hypocrates were added to human peripheral blood neutrophils while the signal of the probe was monitored with a fluorescence microscope. When individual bacteria entered the neutrophils, we registered a sharp increase in the ratiometric response. In the same system, the signal of HypocratesCS also increased, but to a lesser extent. These observations correspond to the previously obtained results from the literature, showing that the human neutrophil phagosome adopts an alkaline pH for several tens of minutes after phagocytosis[60,61]. Thus, the dynamics of the inactivated HypocratesCS version most likely reflect alkalization of the bacterial cytoplasm. With the use of HypocratesCS as a control, it, therefore, becomes possible to assess the contribution of the pH component to the Hypocrates signal.

Finally, Hypocrates is suitable for studying inflammatory reactions in vivo. In this work, we induced inflammation by injuring the caudal fin of Danio rerio larvae. Previously, it was shown with the HyPer biosensor that an $H_2O_2$ gradient is formed in the wound, which serves to attract neutrophils to the area of inflammation[54]. In turn, neutrophils subsequently participate in the elimination of this gradient due to the reaction catalyzed by MPO[55]. Here, we observed the simultaneous real-time dynamics of $H_2O_2$ and (pseudo)hypohalous acids in vivo in zebrafish tissues during inflammation using the red fluorescent biosensor HyPerRed[37] and green-emitting Hypocrates. This demonstrates how Hypocrates can be combined in a multiparameter microscopy mode with other biosensors with suitable spectral properties to disentangle the interactions between various reactive species involved in the inflammatory response, and beyond.

## Methods

**Ethics statement**. The D. rerio experiments were approved by French Ministry of Agriculture (n°C75-05-12). The human neutrophil experiments were approved by the local ethics committee of Pirogov Russian National Research Medical University and conducted in accordance with the Declaration of Helsinki. All blood donors were informed of the final use of their biological materials and signed an informed consent document.

**Expression and purification of S. aureus HypR**. The pET-11b-HypR plasmid[43] was transformed in E. coli BL21 (DE3) cells, which were grown in Lysogeny Broth (LB) at 37 °C until the $A_{600}$ reached 0.8. Isopropyl β-d-1-thiogalactopyranoside (IPTG; 1 mM) was added for the expression induction, followed by 3.5 h of incubation at 37 °C. Harvested cells were pelleted at 4 °C, 5000 rpm for 15 min using the Avanti® J-26xp centrifuge (Beckman Coulter®) with a JLA-8.1000 rotor and resuspended in lysis buffer composed of 20 mM HEPES/NaOH pH 7.5, 0.5 M NaCl, 1 mM dithiothreitol (DTT), 0.1 mg/ml 4-(2-aminoethyl) benzenesulfonyl fluoride hydrochloride (AEBSF), 1 μg/ml Leupeptin, 50 μg/ml Dnase I, and 20 mM MgCl₂. The cells were lysed using a Sonic VibraCell sonicator for 10 min, with 30 s sound/30 s pause with 61% amplitude. Cell debris was removed by centrifugation (45 min at 18,000 rpm, at 4 °C; Avanti® J-26xp centrifuge (BECKMAN COULTER®) with a JA-20 rotor), and the supernatant was in-batch incubated with Ni²⁺-Sepharose 6 Fast Flow beads (Cytiva) equilibrated with the binding buffer (20 mM HEPES/NaOH pH 7.5, 0.5 M NaCl and 10 mM imidazole) for 1 h at 4 °C. The beads were then packed in a column coupled to an AKTA™ Pure system (GE Healthcare, Life Sciences) controlled by UNICORN 6.3.0.731 software). HypR was eluted using a linear gradient with elution buffer: 20 mM HEPES pH 7.5, 0.5 M NaCl and 0–500 mM (0–100%) imidazole. Protein purity was assessed on a non-reducing SDS-PAGE gel, and the pure fractions were collected, dialyzed (~20 ml sample/2 L dialysis buffer) overnight at 4 °C against 20 mM HEPES pH 7.5 and 250 mM NaCl, and stored at −80 °C in 20% glycerol.

**Expression and purification of E. coli NemR$^{C106}$**. The pET-21b(+)-NemR$^{C106}$ plasmid[44], which contains only one cysteine (Cys106), was transformed in E. coli BL21 (DE3) cells. Cells were grown in LB supplemented with 50 μg/ml of kanamycin at 37 °C until the $A_{600}$ reached 0.8. IPTG (0.5 mM) was used for the expression induction, followed by 3 h of incubation at 37 °C. Harvested cells were then pelleted at 4 °C, 5000 rpm for 15 min using the Avanti® J-26xp centrifuge (Beckman Coulter®) with a JLA-8.1000 rotor and resuspended in lysis buffer composed of 50 mM Tris/HCl pH 8, 0.2 M NaCl, 1 mM DTT, 0.1 mg/ml AEBSF, 1 μg/ml Leupeptin, 50 μg/ml Dnase I, and 20 mM MgCl₂. Cells were disrupted and centrifuged as mentioned above. The supernatant was in-batch incubated with Ni²⁺-Sepharose 6 Fast Flow beads (Cytiva) equilibrated with 50 mM Tris/HCl pH 8, 0.2 M NaCl and 1 mM DTT for 1 h at 4 °C. The beads were packed in a column, and the AKTA™ Pure system (GE Healthcare, Life Sciences) controlled by UNICORN 6.3.0.731 software was used for purification. NemR$^{C106}$ was eluted using a linear gradient with elution buffer consisting of 50 mM Tris/HCl pH 8, 0.2 M NaCl, 1 mM DTT and 0 to 700 mM (0–100%) imidazole. Following purification, protein purity was assessed on a non-reducing SDS-PAGE gel, and the pure fractions were dialyzed (~20 ml sample/2L dialysis buffer) overnight at 4 °C against the binding buffer and stored at −20 °C.

**Molecular cloning procedures**. Tersus Plus PCR Kit (Evrogen) was used for all amplification procedures. Primers are listed in Supplementary Table 5. An overlap extension PCR protocol was implemented to engineer different NemR-cpYFP versions. Each reaction mix included the NemR$^{C106}$ N- and C-terminal fragments and the cpYFP fragment in equal molar amounts. The DNA concentration was estimated with horizontal DNA electrophoresis in an agarose gel. The pQE30-HyPer-2 plasmid[35] was used as a template to amplify the cpYFP part. Two versions of this fragment (with SAG/G and SAG/GT linker pairs) were generated with the use of №1/№19 and №2/№20 primer pairs, respectively. The pET-21b(+)-NemR$^{C106}$ plasmid was used as a template to amplify the NemR$^{C106}$ N- and C-terminal parts. All NemR$^{C106}$ N-terminal parts were generated with the use of primer №3 and one of the primers from the №22-33 subset. All NemR$^{C106}$ C-terminal parts were generated with the use of primer №21 and one of the primers from the №4-15 subset. Upon completion of the overlap extension PCR protocol, the target product was separated from the non-target by-products using horizontal DNA electrophoresis in an agarose gel and purified with Cleanup Standard Kit (Evrogen). To engineer pQE30-NemR-cpYFP plasmids, the purified NemR-cpYFP constructs and intact pQE30 vector were incubated with BamHI and HindIII FastDigest™ enzymes in the corresponding buffer (Thermo Scientific) at 37 °C for 20 min. The restricted polynucleotides were purified with Cleanup Standard Kit (Evrogen) and ligated with T4 DNA ligase in the corresponding buffer (Evrogen) at 14 °C overnight. The molar vector/insert ratio was ~1:3 in all cases. The DNA concentration was estimated with horizontal DNA electrophoresis in an agarose gel. After incubation, the samples were transformed into E. coli XL1-Blue cells, which were grown on LB-agar plates containing 100 μg/ml ampicillin for 14 h at 37 °C. To detect colonies bearing the target plasmid, ScreenMix Kit (Evrogen) was used according to the manufacturer's protocol. The positive colonies were then transferred to 100 μg/ml ampicillin LB and grown for 14 h at 37 °C, 200 rpm (New Brunswick™ Excella® E25). The resulting NemR-cpYFP-bearing vectors were purified using Plasmid Miniprep Kit (Evrogen) according to the manufacturer's protocol. The DNA concentration in the pure samples was measured with the use of a NanoDrop 2000 spectrophotometer (Thermo Scientific). The lack of any undesired mutations in the engineered genes was verified by DNA sequencing (Evrogen).

An overlap extension PCR protocol was implemented to engineer HypocratesCS and HypocratesKA versions. In each case, the reaction mix included Hypocrates N- and C-terminal fragments with the desired substitution in equal

molar amounts. The DNA concentration was estimated with horizontal DNA electrophoresis in an agarose gel. The pQE30-Hypocrates plasmid was used as a template to amplify both parts. The N- and C-terminal parts were generated with the use of №3/№34 and №16/№21 primer pairs, respectively, for HypocratesCS and №3/№35 and №17/№21 primer pairs, respectively, for HypocratesKA. The reaction mix after overlap extension PCR was subjected to the same procedures as described above.

To transfer NemR-cpYFP versions from the pQE30 vector to the pCS2+ vector, the corresponding gene was amplified with the use of the №18/№36 primer pair and purified with Cleanup Standard Kit (Evrogen). The obtained construct and intact pCS2+ vector were incubated with ClaI and XbaI FastDigest™ enzymes in the corresponding buffer (Thermo Scientific) at 37 °C for 20 min. The restricted polynucleotides were then subjected to the same procedures as described above.

**Functionality tests of NemR-cpYFP variants in _E. coli_ cells**. To obtain bacterial cells that express NemR-cpYFP variants, the pQE30 vector bearing the desired gene was transformed to _E. coli_ XL1-Blue cells, after which they were grown on LB-agar plates containing 100 μg/ml ampicillin for 14 h at 37 °C. In all cases, the bacterial density was controlled to achieve conditions in which the individual colonies were located at a distance of 1–2 mm from each other, as this parameter significantly affects the maturation and the redox state of the sensors (~5–10 ng of DNA per plate). The fluorescence intensity of the cells was estimated with the use of an Olympus US SZX12 fluorescent binocular microscope. On the first day, all NemR-cpYFP versions were characterized by weak fluorescence, which was attributed to the fact that circularly permuted fluorescent proteins have a destabilized structure and require more time for efficient maturation. Taking that into account, the LB-agar plates were incubated for an additional 24 h at 17–20 °C, as it is known that the maturation of circularly permuted fluorescent proteins proceeds better at lower temperatures.

To test the functionality of NemR-cpYFP variants, the bacterial biomass was transferred to 1 ml of phosphate buffer saline (PBS; 137 mM NaCl, 2.7 mM KCl, 10 mM Na$_2$HPO$_4$, 1.8 mM KH$_2$PO$_4$, pH 7.4, here and after) and resuspended with an automatic pipette. The fluorescence emission spectra ($\lambda_{ex}$ = 425 nm or 500 nm) and excitation spectra ($\lambda_{em}$ = 525 nm) were recorded with the use of a Varian Cary Eclipse Fluorescence Spectrophotometer controlled by Cary Eclipse Scan 1.1(132) Application. The suspensions were treated with NaOCl aliquots to achieve a final oxidant concentration of 80 μM, after which the spectral measurements were repeated. In all cases, the samples were mixed by pipetting prior to the final spectra registration until the signal stabilization was observed. The data were analyzed with OriginPro 9.0 (OriginLab).

**Expression and purification of NemR-cpYFP variants, EYFP, intact cpYFP, HyPer-2, and SypHer3s**. In the current work, two different protocols for Hypocrates expression and purification were used. Both of them led to obtaining the functional biosensor. Therefore, they should be considered equal.

Protocol 1. XL1-Blue cells were transformed with pQE30-Hypocrates plasmid, after which they were plated (LB-agar medium, 100 μg/ml ampicillin) and incubated for 14 h at 37 °C. The bacterial density was controlled as described above. To achieve better protein folding and maturation, the plates were additionally incubated for 24 h at 17–20 °C. Next, the cells were washed from the agar surface by ice-cold PBS, and the final volume of the suspension was adjusted to 24 ml with the same buffer. The number of plates used for a single purification procedure was twenty. The cells were disrupted with the use of a Sonic VibraCell instrument in an ice bath (5 s sonication + 10 s pause cycle; total sonication time – 9 min; the amplitude – 32%). The obtained lysates were centrifuged for 20 min at 21,000×_g_ and 4 °C (Centrifuge 5424 R, Eppendorf) to precipitate insoluble fractions. The supernatants were collected and applied to a column filled with 5 ml of TALON Metal Affinity resin (Takara) previously equilibrated with ice-cold PBS. The column was washed with 50 ml of the same buffer to get rid of non-target proteins. The elution step was performed by the addition of 10 ml of ice-cold PBS containing 250 mM imidazole, and the fraction with the target protein was collected on the basis of its bright yellow color. The elimination of imidazole was achieved by gel filtration on columns filled with 10 ml of Sephadex G-25 (GE Healthcare Life Sciences) previously equilibrated with ice-cold PBS. The pure protein sample was stored at 4 °C for no more than 3 days. The addition of any reducing agents (such as β-mercaptoethanol) did not alter the properties of the protein – the sensor was obtained in its fully reduced form, even in their absence. Hypocrates samples, purified according to this protocol, were used for the following tests: the measurements of optical parameters, fluorescence spectra stability and reversibility experiments, titration by _N_-chlorotaurine at different pH values, gel filtration, and measurements of MPO activity. Other primary NemR-cpYFP versions, EYFP, cpYFP, SypHer3s, and HyPer-2 were purified following the same protocol. However, in the case of the latter, all buffers, except for those used at the gel filtration step, contained 5 mM β-mercaptoethanol to avoid the oxidation of the sensor. The protein concentration in the final samples was measured with the use of Bicinchoninic Acid Kit for Protein Determination (Sigma-Aldrich) and a 96-well plate analyzer (Tecan Infinite 200 PRO) controlled by Tecan i-control 1.11.1.0 software.

Protocol 2. Shuffle® T7 or XL1-Blue cells were transformed with pQE30-Hypocrates plasmid, respectively. The cells were plated on LB-agar-ampicillin and incubated overnight at 37 °C (for XL1-Blue) and 30 °C (for Shuffle® T7). Plates were transferred to a 25 °C incubator until they expressed the protein, as indicated by yellow-colored colonies. At the next step, several colonies were transferred to 3 L of LB medium supplemented with 100 μg/ml ampicillin and incubated for 36 h at 25 °C with shaking at 180 rpm. Cells were harvested, and the pellet was resuspended in lysis buffer composed of 40 mM Tris pH 7.5, 150 mM KCl, 10 mM MgSO$_4$, 5 mM DTT, 0.1 mg/ml AEBSF, 1 μg/ml Leupeptin, 50 μg/ml DnaseI, and 20 mM MgCl$_2$. Cells were lysed and centrifuged as performed for NemR$^{C106}$, and the supernatant was in-batch incubated with Ni$^{2+}$-Sepharose beads (Thermo Scientific) equilibrated with binding buffer: 40 mM Tris pH 7.5, 150 mM KCl, 10 mM MgSO$_4$ and 1 mM DTT for 1 h at 4 °C. After column packing, the AKTA™ Pure system (GE Healthcare, Life Sciences) controlled by UNICORN 6.3.0.731 software was used to elute the protein using a binding buffer with 400 mM imidazole followed by size exclusion chromatography on a Superdex75 16/600 (GE Healthcare) column equilibrated in binding buffer. The purity of the protein was assessed on a non-reducing SDS-PAGE gel, and the pure fractions were collected and stored at −20 °C. Hypocrates and HypocratesCS samples, purified according to this protocol, were used for the following tests: circular dichroism experiments, pKa determination, fluorescence selectivity experiments, fluorescence sensitivity experiments, pre-steady-state kinetic measurements, HypocratesCS crystallization, and mass spectrometry.

**_N_-chlorotaurine, NaOBr, HOSCN, and NaONOO preparation**. The preparation of _N_-chlorotaurine (NCT) was carried out according to Patent DE4041703A1 (https://patents.google.com/patent/DE4041703A1/en). Chloramine T trihydrate (6.0 g, 21.3 mmol) was dissolved in dry methanol (50 ml). Finely powdered taurine (2.5 g, 20 mmol) was added, and the mixture was stirred for 20 h at room temperature (20–25 °C). The solvent was removed on a rotary evaporator, and the residue was washed with isopropyl alcohol (3 times, 10 ml) and diethyl ether (three times, 35 ml). The white solid was dried in vacuum (5 mmHg, 1 h). The NMR analysis of the product (DMSO-d6) showed the absence of aromatic protons. The product was stored at −20 °C.

The preparation of NaOBr solution was carried out according to Liu et al.[32] NaOH (11.7 g, 0.30 mol) was dissolved in water (100 ml). The solution was cooled in an ice bath to −5 °C. Liquid bromine (3.86 ml) was added dropwise upon stirring with a temperature not exceeding 0 °C. The mixture was stirred for 1 h at the same temperature, resulting in the NaOBr solution that was used in further experiments. The solution was stored in a dark, cold place (4 °C). The concentration of BrO$^-$ ions was determined before each experiment by spectrophotometry (Varian Cary 5000 Spectrophotometer controlled by Varian UV Scan Application 3.00(339)). For these measurements, the solution was diluted with NaOH (pure water, 0.1 M concentration). The concentration was determined using Lambert-Beer's law, taking: ε at 329 nm = 332 M$^{-1}$cm$^{-1}$ for BrO$^-$ ions.

The preparation of HOSCN solution was carried out according to Lane et al.[62] Lactoperoxidase (LPO (Sigma); 40 μM) was mixed with sodium thiocyanate (150 mM) in 100 mM sodium phosphate buffer pH 7.4. The obtained sample was incubated on ice for 15 min, and 5 × 10 μL aliquots of H$_2$O$_2$ (75 mM) were added with a space of 1 min apart. Catalase (Sigma; 1 mg/mL) was added to the reaction mixture for 5 min to remove the excess of H$_2$O$_2$. Next, LPO and catalase were removed using Vivaspin 500 (Merck; 10,000 Da cut-off) and the flow-through was collected. The concentration of HOSCN was determined in the presence of TNB (extinction coefficient of 14,150 M$^{-1}$cm$^{-1}$) To prepare the TNB solution, 2 mg of reagent were dissolved in 200 μl of 50 mM NaOH. After vortexing, and incubation at room temperature for 5 min, the solution was diluted 40x in 100 mM sodium phosphate buffer pH 7.4. HOSCN was added to the prepared TNB solution (1:200 ratio, respectively), and the absorbance at 412 nm was measured using the SpectraMax iD5 plate reader (Molecular Devices) controlled by Softmax Pro 7.1.

The preparation of NaONOO solution was carried out as described by Uppu[63]. NaOH (4.0 g, 0.10 mol) was dissolved in water (35 ml). The mixture was cooled in an ice bath to 5-0 °C, and a solution of 35% H$_2$O$_2$ (11 ml, ~0.11 mol) and EDTA (solid, 75 mg) were added. Liquid isoamyl nitrite (13.5 ml, 0.10 mol) was added, and the mixture was vigorously stirred at room temperature (~25 °C) for 5 h. The mixture was diluted with dichloromethane (100 ml), and the water phase was separated and washed additionally with dichloromethane (five times, 100 ml each). The unreacted H$_2$O$_2$ was then removed by passing the aqueous phase through manganese dioxide (10–15 g, 5 mm layer). The resulting solution was additionally filtered from traces of MnO$_2$, and the traces of dichloromethane were removed in vacuum (5 mmHg, 1 h). The resulting NaONOO solution was used in the further experiments. The solution was stored at −20 °C. The concentration of ONOO$^-$ ions was determined before each experiment by spectrophotometry (Varian Cary 5000 Spectrophotometer controlled by Varian UV Scan Application 3.00(339)). For these measurements, the solution was diluted with NaOH (pure water, 0.1 M concentration). The concentration was determined using Lambert-Beer's law, taking: ε at 302 nm = 1670 M$^{-1}$cm$^{-1}$ for ONOO$^-$ ions.

**Measurements of the optical parameters of NemR-cpYFP variants**. To measure the brightness of the primary NemR-cpYFP versions, the proteins were diluted in PBS to equimolar concentrations (according to the Bicinchoninic Acid Kit). Purified EYFP served as the comparison control. The absorbance and fluorescence excitation spectra ($\lambda_{em}$ = 513 nm and 533 nm for NemR-cpYFP variants and EYFP,

respectively) of the samples were recorded with the use of a Varian Cary 5000 Spectrophotometer (controlled by Varian UV Scan Application 3.00(339)) or a Varian Cary Eclipse Fluorescence Spectrophotometer (controlled by Cary Eclipse Scan 1.1(132) Application). The molar extinction coefficients ($\varepsilon$) were calculated according to the following equation $\varepsilon = A/(C \cdot L)$, where $A$ was the optical density at the studied absorption maximum, $C$ was the protein concentration (M), and $L$ was the optical path length (cm). The fluorescence quantum yields (QY) were calculated according to the following equation $QY_{NemR-cpYFP} = QY_{EYFP} \cdot (A_{EYFP} \cdot Em_{NemR-cpYFP}/(A_{NemR-cpYFP} \cdot Em_{EYFP}))$, where $A$ was the optical density at the studied absorption maximum, and Em was the emission intensity at the studied excitation maximum ($\lambda_{ex} = 425$ nm or 500 nm for NemR-cpYFP variants, and 519 nm for EYFP). $QY_{EYFP}$ is a standard value of 0.67 according to the literature (Fpbase ID: 8DNLG). The data were analyzed with OriginPro 9.0 (OriginLab).

The purified sensor samples might contain not fully folded and matured molecules, reducing the accuracy of the optical parameters' measurements. Therefore, the molar extinction coefficients and QYs of Hypocrates were investigated in more detail. To estimate the concentration of fully matured chromophores, the samples of Hypocrates and EYFP in PBS were mixed with 1 M NaOH at a volume ratio of 1:1 and incubated for 5 min. In the described conditions, yellow fluorescent proteins undergo denaturation, and mature chromophores are converted to the form absorbing at 445 nm with $\varepsilon = 44{,}000$ $M^{-1}cm^{-1}$ [64]. To investigate how reducing and oxidizing agents alter the optical parameters, some of the Hypocrates samples were incubated in the presence of 0.5 mM N-chlorotaurine or 5 mM DTT for 30 min prior to spectra registration. All of the following procedures were carried out as described above. The data were analyzed with OriginPro 9.0 (OriginLab).

**Fluorescence spectra stability and reversibility experiments.** To investigate whether high oxidant concentrations damage the proteins, purified Hypocrates samples (0.5 μM) were treated with saturating oxidant concentrations (5–10 μM), and their fluorescence excitation spectra ($\lambda_{em} = 520$ nm) were recorded. Next, aliquots of corresponding oxidants were added to achieve extremely high concentrations (105–110 μM), and the same measurements were carried out. In all cases, the samples were mixed by pipetting prior to the final spectra registration until the signal stabilization was observed. NaOCl and N-chlorotaurine were tested in PBS, while NaONOO and NaOBr were tested in 100 mM sodium phosphate buffer (pH 7.4) to avoid possible OCl⁻ generation in the system. In the latter two cases, the protein aliquots were buffer-exchanged using Amicon Ultra-0.5 Centrifugal Filter Units (Millipore) or by gel filtration on columns filled with 10 ml of Sephadex G-25 (GE Healthcare Life Sciences) previously equilibrated with ice-cold medium. The NaOCl sensitivities of intact cpYFP, SypHer3s and HyPer-2 purified proteins were investigated according to the same protocol. The measurements were performed with the use of a Varian Cary Eclipse Fluorescence Spectrophotometer controlled by Cary Eclipse Scan 1.1(132) Application. The data were analyzed with OriginPro 9.0 (OriginLab).

To investigate whether Hypocrates oxidation is reversible, purified protein samples (0.5–2 μM) were treated with saturating oxidant concentrations (5–50 μM) and incubated for 5 min, after which the fluorescence excitation spectra were recorded. Next, DTT was added to the reaction mix to the final concentration of 1–5 mM, and the probes were incubated for 40–60 min prior to the spectra registration. In some cases, two additional control probes (intact and with the same oxidant concentration) were prepared and incubated for the same time to control for possible artifacts caused by prolonged atmosphere exposure. NaOCl and N-chlorotaurine were tested in PBS, while NaOBr and HOSCN were tested in 100 mM sodium phosphate buffer (pH 7.4) to avoid possible OCl⁻ generation in the system. In the latter two cases, the protein aliquots were buffer-exchanged via Amicon Ultra-0.5 Centrifugal Filter Units (Millipore) or a Hi-Trap® desalting column (GE Healthcare) using the AKTA™ Pure system (GE Healthcare, Life Sciences) controlled by UNICORN 6.3.0.731 software. The reversibility of cpYFP signal change after NaOCl-mediated oxidation was investigated according to the same protocol. The measurements were performed with the use of either a Varian Cary Eclipse Fluorescence Spectrophotometer (controlled by Cary Eclipse Scan 1.1(132) Application) or an LS55 luminescence spectrophotometer (controlled by FL WinLAB 4.00.03). The data were analyzed with OriginPro 9.0 (OriginLab).

**Biosensor secondary structural changes with circular dichroism.** Changes in the overall secondary structure of Hypocrates between its reduced and oxidized (NaOCl or $H_2O_2$) forms were evaluated with circular dichroism (CD) spectroscopy. The protein was reduced with 30 mM DTT for 30 min at room temperature. A Hi-Trap® desalting column (GE Healthcare), equilibrated with 20 mM sodium phosphate buffer pH 7.4, was used to remove excess DTT. To prepare the oxidized samples, Hypocrates (25 μM) was incubated for 10 min at room temperature with different concentrations of NaOCl (1:1, 1:5, and 1:10 ratios) or $H_2O_2$ (1:1, 1:3, and 1:6 ratios of protein to oxidant concentration), with a reaction buffer composed of 20 mM sodium phosphate, pH 7.4, and 200 mM sodium fluoride. Micro Bio-Spin® Chromatography Columns (BIO-RAD), equilibrated with the same buffer, were used to remove the oxidants. Following sample preparation, a Jasco J-715 spectropolarimeter controlled by J-700 1.07.00 software was used to analyze 4 μM of each sample at 25 °C in a quartz cuvette with a 1-mm path length. Far-UV CD spectra (190–260 nm) were measured, and the data were analyzed with GraphPad Prism8 and OriginPro 9.0 (OriginLab).

To determine whether the overall secondary structure could be restored, DTT was used. NaOCl-oxidized Hypocrates (1:10 protein/oxidant ratio) was incubated with 200 μM DTT for 5 min at room temperature. The background from the buffer and the presence of 200 μM DTT were subtracted.

**pKa determination of reduced and oxidized NemR-cpYFP versions.** To determine the pKa of Hypocrates, the protein was reduced with DTT and buffer-exchanged into 100 mM sodium phosphate buffer (pH 7.4) using a Hi-Trap® desalting column (GE Healthcare). Reduced Hypocrates (0.5 μM) in the presence or absence of 12.5 μM oxidants (NaOCl and NCT) was diluted in a polybuffer solution with several pH values (0.5 pH unit intervals), and the excitation spectra (with $\lambda_{em} = 555$ nm) were recorded after 5 min of incubation at 25 °C using a SpectraMax iD5 plate reader (Molecular Devices) controlled by Softmax Pro 7.1. The polybuffer solution consisted of sodium acetate (10 mM), sodium phosphate (10 mM), sodium borate (10 mM), and sodium citrate (10 mM). The $Ex_{500}/Ex_{417}$ ratio was plotted as a function of increasing buffer pH. For each measurement, at least three independent replicates were performed, and the data were analyzed using GraphPad Prism8 and OriginPro 9.0 (OriginLab). The pKa of reduced HypocratesCS was determined as described for reduced Hypocrates.

**Fluorescence selectivity experiments.** Prior to the experiment, the purified protein was reduced with DTT and buffer-exchanged using a Hi-Trap® desalting column (GE Healthcare) into 100 mM sodium phosphate buffer (pH 7.4). Aliquots of the sensor (2 μM) were incubated with different oxidants: NaOCl (50 μM), NaOBr (50 μM), HOSCN (50 μM), N-chlorotaurine (50 μM), $H_2O_2$ (50 μM), glutathione (GSH; 500 μM), glutathione disulfide (GSSG; 500 μM), MAHMA NONOate (NO• generator; 50 μM), $Na_2S$ (50 μM), NaONOO (50 μM), glyoxal (50 μM), methylglyoxal (50 μM), formaldehyde (50 μM), and the xanthine (X)/xanthine oxidase (XOX) system ($O_2^{\bullet-}$ generator; 50 μM X + 0.05 U/ml XOX). Both NaONOO and X/XOX samples contained catalase (0.1 μM) as an additional control, which removed $H_2O_2$ generated during the reaction. The workability of the X/XOX system was checked by its ability to alter the shape of the cytochrome C absorption spectrum with the use of a Varian Cary 5000 Spectrophotometer controlled by Varian UV Scan Application 3.00(339). MAHMA NONOate was dissolved in 10 mM NaOH to decrease its degradation rate. The samples were incubated for 5 min at 25 °C, and the ratiometric fluorescence changes were monitored by an excitation scan ($\lambda_{em} = 515$ nm) using an LS55 luminescence spectrophotometer (PerkinElmer) controlled by FL WinLAB 4.00.03. For each concentration, at least three independent experimental measurements were performed. The selectivity of NemR[C106] was determined following the same protocol, except for the fact that the changes in intrinsic Trp fluorescence were measured as a readout using an emission scan ($\lambda_{ex} = 295$ nm). The data were analyzed with OriginPro 9.0 (OriginLab).

**Fluorescence sensitivity experiments.** Hypocrates sensitivity experiments were performed in 100 mM sodium phosphate buffer (pH 7.4). Aliquots of the purified protein (0.5 μM) were incubated with increasing concentrations of oxidants (NaOCl, NaOBr, HOSCN, and N-chlorotaurine) for 5 min at 25 °C. The excitation scans (with $\lambda_{em} = 555$ nm) were recorded with the use of a SpectraMax iD5 plate reader (Molecular Devices) controlled by Softmax Pro 7.1. The $Ex_{500}/Ex_{417}$ ratio, which represents the ratio between fluorescence excited at 500 nm and at 417 nm, was plotted as a function of increasing oxidant concentration. The initial linear part of the hyperbolic curve was analyzed using linear regression, where the slope values represent the sensitivity towards the corresponding oxidants. For each measurement, at least two independent replicates were performed, and the data were analyzed using GraphPad Prism8 and OriginPro 9.0 (OriginLab).

The sensitivities of Hypocrates at different pH values towards N-chlorotaurine were analyzed in 100 mM sodium phosphate buffers (pH 7.00, 7.30, 7.60, and 7.90). Aliquots of the purified protein (0.5 μM) were treated with serial N-chlorotaurine additions of 0.3–5 μM up to the final concentration of 15 μM. The fluorescence excitation spectra ($\lambda_{em} = 525$ nm) were recorded with the use of a Varian Cary Eclipse Fluorescence Spectrophotometer controlled by Cary Eclipse Scan 1.1(132) Application. At each titration step, the samples were mixed by pipetting prior to the final spectra registration until the signal stabilization was observed. The $Ex_{500}/Ex_{425}$ ratio, which represents the ratio between fluorescence excited at 500 nm and at 425 nm, was plotted as a function of increasing oxidant concentration. For each buffer, three independent replicates were performed, and the data were analyzed using OriginPro 9.0 (OriginLab).

**Pre-steady-state kinetic measurements.** Pre-steady-state kinetic measurements were performed using a stopped-flow apparatus coupled to a fluorescence detector (Applied Photophysics SV20) and controlled by Prodata SX 2.5.0. For HypR, changes in intrinsic Tyr fluorescence were measured using $a > 305$ nm cut-off filter ($\lambda_{ex} = 274$ nm). For NemR[C106], changes in intrinsic Trp fluorescence were measured using $a > 320$ nm cut-off filter ($\lambda_{ex} = 295$ nm). For Hypocrates, changes in the cpYFP chromophore fluorescence were measured using $a > 515$ nm cut-off filter ($\lambda_{ex} = 485$ nm). The excitation and emission wavelengths for each protein were determined prior to the stopped-flow experiments using an LS55 luminescence spectrophotometer (PerkinElmer) controlled by FL WinLAB 4.00.03. The

oxidant concentration range required to lead to fluorescence changes for each protein was also determined.

Prior to the experiment, the samples were reduced with 30 mM DTT for 30 min at room temperature. A Hi-Trap® desalting column (GE Healthcare), equilibrated with argon-flushed 100 mM sodium phosphate pH 7.4 buffer, was used to remove excess DTT. To determine the second-order rate constants, 0.5 μM Hypocrates or 1 μM NemR or HypR was mixed with increasing concentrations of an oxidant (NaOCl, NaOBr or N-chlorotaurine) in a reaction medium of 100 mM sodium phosphate buffer at 25 °C. Changes in fluorescence were monitored, and the obtained curves were fitted with double exponential equations which gave trustworthy diagnostic residual plots. From the double exponential equations, the observed $k_{fast}$ values were used to determine the second-order rate constants (GraphPad Prism8 and OriginPro 9.0).

**Mass spectrometry.** Hypocrates protein was reduced with 20 mM DTT and desalted via a Hi-Trap® desalting column (GE Healthcare) using the AKTA™ Pure system (GE Healthcare, Life Sciences) controlled by UNICORN 6.3.0.731. The desalting buffer solution (100 mM sodium phosphate buffer pH 7.4) was argon flushed prior to use. The protein was then transferred into the Whitley A35 anaerobic workstation (Don Whitley Scientific). A reduced biosensor (30 μM) was incubated for 10 min with different oxidants (NaOCl, NaOBr, NCT, or HOSCN) at a 1:20 ratio of protein to oxidant concentration. The samples were then desalted using Micro Bio-Spin® Chromatography Columns (BIO-RAD) to remove excess oxidants. Finally, 10 mM NEM was added to block available free thiols. The samples were then stored at −20 °C until mass spectrometry analysis.

Proteins were precipitated with methanol/chloroform and digested overnight with trypsin at 37 °C, the reaction was stopped with 0.1% (v/v) TFA. Peptides were dissolved in solvent A (0.1% TFA in 2% ACN), directly loaded onto reversed-phase pre-column (Acclaim PepMap 100, Thermo Scientific) and eluted in backflush mode. Peptides were separated by reverse-phase chromatography on an analytical column (0.075 x 250 mm Pepmap C18, Thermo Sientific) by a linear gradient of solvent B (0.1% FA in 80% ACN) min at a flow-rate of 300 nl/min. The different steps were: 4%–27.5% solvent B for 100 min, 27.5%–40% for 10 min, 40%–95% for 1 min, and holding at 95% for the last 10 min. The peptides were analyzed by an Orbitrap Fusion Lumos tribrid mass spectrometer (ThermoFisher Scientific) controlled by XCalibur 4.3 and LUMOS tune application 3.3 software. The peptides were ionized by an NSI source set at 2.1 kV coupled online to the nano-LC (Ultimate 3000 RSLC, Thermo Scientific). Precursor ions were detected in the Orbitrap at a resolution of 120,000. Peptides were selected for MS/MS using CID setting at 30; daughter ions were detected in the Ion Trap working at a normal scan rate. A data-dependent scan routine (DDA) that alternated between a full-MS scan followed by MS/MS scans was applied for 5 seconds for ions above a threshold ion count of 1.0E4 in the MS1 master scan with 40.0 s dynamic exclusion. Full-MS spectra were acquired with an AGC target of 4E5 ions and a maximum injection time of 50 ms, and MS/MS spectra were obtained with an AGC target of 1E4 ions and a maximum injection time of 35 ms. For MS1 scans, the m/z scan range was set from 350 to 1800. The recorded raw data were inspected with Freestyle 1.6. The MS/MS data were processed using Sequest HT search engine within Proteome Discoverer 2.4 SP1 against a Escherichia coli K12 protein database obtained from Uniprot (4349 entries) and containing the Hypocrates sequence. Trypsin was specified as the protease allowing up to 2 missed cleavages, 4 modifications per peptide, and up to 5 charges. The mass error was set to 10 ppm for precursor ions and 0.5 Da for fragment ions. Oxidation on Met and Trp (+15.995 Da), dioxidation on Cys and Trp (+31.990 Da), trioxidation on Cys (+47.985 Da), bromination or chlorination on Phe, His, and Trp (+77.991 and +33.961 Da, respectively) and N-ethylmaleimide on Cys (+125.048 Da) were considered as variable modifications. False discovery rate (FDR) was assessed using Percolator and thresholds for protein, peptide, and modification site were specified at 1%. For abundance comparison, abundance ratios were calculated by Label Free Quantification (LFQ) of the precursor intensities within Proteome Discoverer 2.4 SP1. The mass spectrometry data have been deposited to the ProteomeXchange Consortium via the PRIDE[65] partner repository with the dataset identifier PXD029624 and https://doi.org/10.6019/PXD029624.

**HypocratesCS crystallization, X-ray data collection, and structure determination.** HypocratesCS was crystallized at 7 mg/ml concentration at 10 °C or 20 °C using the hanging-drop vapor diffusion method with Tris (0.1 M, pH 8), CaCl₂ (0.1 M), MgCl₂ (0.1 M), and PE15/4 (15%) as a precipitant solution. The drops were composed of 1 μl of protein and 1 μl of precipitant solution. To obtain larger and better diffracting crystals, the small needles obtained within the above crystallization condition were used for microseeding.

For X-ray data collection, the cryo-protectant used was the same as the precipitant solution, but with 30% PE15/4. X-ray data were collected at 100 K at the Proxima 2 beamline of the Soleil synchrotron facility, at a wavelength of 0.980113 Å, and processed using XDS 0.6.5.5[66]. The structure of HypocratesCS was solved by molecular replacement using Phase 2.7.16[67] from the Phenix suite[68], using both the E. coli NemR (PDB: 4YZE [https://doi.org/10.2210/pdb4YZE/pdb]) (100% identical to the sensory domain of HypocratesCS) and the cpYFP-based calcium biosensor (PDB: 3O77 [https://doi.org/10.2210/pdb3O77/pdb]) (98% identity to the cpYFP) as search models. Coot 0.8.9.2 EL[69] was used to manually

complete the building of the structure, and the refinement was done using Phenix.Refine 1.11.1_2575[70] from the Phenix suite. Analysis of the Ramachandran plot showed that 97.82% of the residues are in the most favored areas of the Ramachandran plot, 1.70% in additionally allowed areas, and 0.49% in disallowed areas. The data collection statistics and refinement parameters are summarized in Supplementary Table 4.

**Gel filtration experiments.** Gel filtration experiments were performed using a Superdex® 200 Increase 10/300 GL column (Cytiva) equilibrated with 100 mM sodium phosphate buffer (pH 7.4) at 24 °C and at a flowrate of 0.75 ml/min. The column was connected to an Agilent 1260 Bio-Inert LC system equipped with Agilent 1260 diode array detector and calibrated using cytochrome C (12.4 kDa), carbonic anhydrase (29 kDa), bovine serum albumin (66 kDa), alcohol dehydrogenase (150 kDa), b-amilase (200 kDa) and ferritin (450 kDa). The equipment was controlled by Agilent OpenLAB CDS ChemStation Edition C.01.07 SR3 software. Hypocrates samples were incubated in the presence of 5 mM DTT or with N-chlorotaurine at 1:20 protein/oxidant ratio for 5 min before the injection (50 μl). The sensor was visualized by measuring absorbance at 415 nm to detect only molecules with the mature chromophore. The data were analyzed with OriginPro 9.0 (OriginLab).

**Measurements of MPO activity with purified Hypocrates.** To test whether Hypocrates is capable of visualizing MPO activity in vitro, the purified sensor was incubated with 0.1 U/ml human MPO and 100 μM H₂O₂ for 10 min in PBS, after which the fluorescence excitation spectrum ($\lambda_{em}$ = 525 nm) was recorded with the use of a Varian Cary Eclipse Fluorescence Spectrophotometer controlled by Cary Eclipse Scan 1.1(132) Application. The probes that contained only MPO or H₂O₂ were treated according to the same protocol to control for non-specific fluorescence changes. The data were analyzed using OriginPro 9.0 (OriginLab).

To record the dynamics of HOCl production by MPO in vitro, purified Hypocrates was incubated in PBS in the presence of 0.1 U/ml human MPO, and the intensities of fluorescence ($\lambda_{em}$ = 525 nm) excited at 425 nm and 500 nm were collected every 2.4 s with the use of a Varian Cary Eclipse Fluorescence Spectrophotometer controlled by Cary Eclipse Kinetics 1.1(133) Application. To start the MPO reaction, 100 μM H₂O₂ was added to the reaction mix. A probe without MPO was treated according to the same protocol to control for H₂O₂-induced fluorescence changes. A probe without MPO and without H₂O₂ addition was treated according to the same protocol to control for fluorescence changes attributed to prolonged incubation. For all three samples, the $Ex_{500}/Ex_{425}$ ratio was calculated as a function of time, and the first two curves were normalized to the third one. The data were analyzed using OriginPro 9.0 (OriginLab).

**NaOCl visualization with Hypocrates, HypocratesCS, and SypHer3s in HeLa Kyoto cells.** HeLa Kyoto cells were cultured in DMEM (PanEko) supplemented with 10% FBS (Biosera), 2 mM L-glutamine (PanEko), 50 units/ml penicillin (PanEko) and 50 μg/ml streptomycin (PanEko) at 37 °C in an atmosphere containing 5% CO₂. Cells were passaged every 2–3 days. For transfection, cells were seeded into 35-mm glass-bottom dishes (SPL Lifesciences). After 24 h, cells were transfected with the plasmid of the required sensor using FuGene HD transfection reagent (Promega) according to the manufacturer's protocol. Fluorescence microscopy was performed on the next day after transfection with a Leica DMI 6000 microscope, equipped with an HCX PL Apo CS 40.0 × 1.25 Oil UV objective, CFP (excitation filter BP 436/20, dichromatic mirror 455, suppression filter BP 480/40) and GFP (excitation filter BP 470/40, dichromatic mirror 500, suppression filter BP 525/50) filter cubes. The microscope was controlled by LAS X 2.6.4.8702 software. A 10 mM stock solution of NaOCl (EMPLURA) in Milli-Q water was freshly prepared before cell imaging. Cell culture medium was replaced with 900 μl of PBS, and baseline fluorescence was detected for several minutes. PBS was chosen as an inorganic imaging medium because NaOCl, being a strong oxidant, can react with the nitrogen-containing components of the medium and thus introduce inaccuracy to the results. The desired amount of NaOCl stock was diluted in 100 μl of PBS just before addition, and the final concentration of NaOCl in the sample was in the range of 10–40 μM (~4.2–17 nmol/(10⁵ cells)). All measurements were taken at room temperature because the maximum response amplitude of Hypocrates is being reduced as a result of heating. The responses of sensors were calculated as ratios of fluorescence intensities excited at 500 nm and at 425 nm ($Ex_{500}/Ex_{425}$ ratio) and normalized to the signal of the probe on the first image of the series. Processing of the images and quantification of results were performed using Fiji 2.0.0-rc-69/1.52p/Java 1.8.0_172 (https://fiji.sc), Excel 2016 (Microsoft), and OriginPro 9.0 (OriginLab).

**Polymorphonuclear leukocytes isolation.** Polymorphonuclear leukocytes (PMNs), including neutrophils, were isolated from the whole blood of healthy volunteers. All donors were informed of the final use of their blood and signed an informed consent document. The study was approved by the local ethics committee of Pirogov Russian National Research Medical University and conducted in accordance with the Declaration of Helsinki. The blood was collected in tubes with anticoagulant (EDTA) and separated on Polymorphprep™ (Alere Technologies AS) gradient according to the manufacturer's protocol. After blood separation by

centrifugation, the layers containing serum and PMNs were transferred to individual tubes. The collected serum was further used for the opsonization of bacteria. PMNs fraction was diluted 10-fold in the red blood cell lysis solution (Miltenyi Biotec) and incubated for 10 min at room temperature, after which the cells were collected by centrifugation at $400 \times g$, 10 min. This procedure was repeated two times to remove all residual erythrocytes. Next, the obtained PMNs were washed twice with PBS by centrifugation ($400 \times g$, 10 min) and finally resuspended in RPMI-1640 medium (PanEko) supplemented with 2 mM L-glutamine (PanEko) and 0.5% FBS (Biosera). The concentration of the PMNs was adjusted to $1.5 \times 10^6$ cells/ml. For activation, the PMNs were incubated with 1 μg/ml recombinant human IFN-γ (Gibco$^{TM}$) for 1.5 h in the CO$_2$ incubator (37 °C, 5% CO$_2$). Then, the liquid was replaced with the same medium without IFN-γ by centrifugation and cells were kept in the CO$_2$ incubator (37 °C, 5% CO$_2$) until the imaging experiments.

**Live-cell imaging of phagocytosis**. To obtain bacterial cells expressing Hypocrates or HypocratesCS, chemically competent E. coli XL1-Blue cells were transformed with the pQE30 vectors encoding the required constructs. The bacterial colonies were grown on LB-agar plates with ampicillin (150 μg/ml) for 14 h at 37 °C. To achieve better protein maturation, the plates were additionally kept at room temperature for two more days. For opsonization, the bacteria were diluted in 1 ml of PBS and human serum mixture (1:1), after which the sample was incubated in a thermoshaker for 30 min at 37 °C, 300–400 rpm. Opsonized bacteria were washed three times with PBS by centrifugation ($1200 \times g$, 2 min) and the collected cellular mass was resuspended in the same buffer to obtain an OD$_{600}$ of 0.5.

For live-cell imaging of the PMNs, the medium was changed to RPMI-1640 without phenol red and sodium bicarbonate (Sigma-Aldrich) supplemented with 2 mM L-glutamine (Sigma-Aldrich), 0.5% FBS (Biosera), and 20 mM HEPES. This procedure was performed by centrifugation at $400 \times g$ for 10 min. At the next step, 800 μl of the PMNs suspension ($1.8 \times 10^6$ cells/ml) were transferred into a 35 mm confocal dish (SPL Lifesciences) for microscopy. To induce phagocytosis, 75 μl of opsonized E. coli cells in PBS (OD$_{600}$ = 0.5) were added per dish. Microscopy was performed with an ECLIPSE Ti2 inverted microscope (Nikon Instruments Inc.) equipped with Plan Apo VC 100X NA 1.40 oil objective. The microscope was controlled by NIS-Elements 5.21.03 software. The signal was detected in three channels: 395 nm diode (exposure 100 ms, diode intensity 9%), 470 nm diode (exposure 200 ms, diode intensity 9%), and brightfield.

The signals of Hypocrates and HypocratesCS in bacteria were calculated as ratios of fluorescence intensities excited at 470 nm and 395 nm (Ex$_{470}$/Ex$_{395}$). To measure the response amplitudes, the signals were normalized to the values observed before the onset of phagocytosis. Image processing and quantification of results were performed using Fiji 2.0.0-rc-69/1.52p/Java 1.8.0_172 (https://fiji.sc), Excel 2016 (Microsoft), and OriginPro 9.0 (OriginLab).

***Danio rerio* tail fin inflammation model**. The study was approved by French Ministry of Agriculture (n°C75-05-12). For the tail fin amputation experiment, mRNAs of Hypocrates, HypocratesCS, and HyPerRed were in vitro synthesized using the mMessage mMachine Transcription Kit (Invitrogen) according to manufacturer's manual. For transient expression of the biosensors in zebrafish larvae (wild-type Tübingen (TU) strain, male and female), 80 ng/μl of Hypocrates or HypocratesCS mRNA and 50 ng/μl of HyPerRed mRNA were co-injected into 1-cell-stage embryos. The zebrafish embryos were maintained in egg water containing 0.2 mM N-phenylthiourea (PTU; Sigma) to prevent pigment formation at 28 °C. Fluorescence imaging was performed 48 h post-fertilization (hpf). Larvae were anesthetized in 0.02% MS-222 tricaine (Sigma), embedded in low-melting agarose (0.8%) and then subjected to tail fin amputation under a stereoscopic microscope. Imaging was performed with a CSU-W1 Yokogawa spinning disk coupled to a Zeiss Axio Observer Z1 inverted microscope equipped with a sCMOS Hamamatsu camera and a 25x (Zeiss 0.8 Imm WD: 0.19 mm) oil objective. The microscope was controlled by Metamorph Premier 7.8. DPSS 100 mW 405 nm and 150 mW 491 nm lasers and a 525/50 bandpass excitation filter were used for Hypocrates and HypocratesCS imaging. A 100 mW 561 nm laser and a 595/50 bandpass filter were used for HyPerRed imaging. To quantify the response, fluorescent signal at the amputation plane was normalized to the mean fluorescence of the tail before amputation. A statistical two-way ANOVA test with a Tukey's multiple comparisons posttest was then performed. Image processing and quantification of results were performed using Fiji 2.0.0-rc-69/1.52p/Java 1.8.0_172 (https://fiji.sc), Excel 2016 (Microsoft), and OriginPro 9.0 (OriginLab).

**Reporting summary**. Further information on research design is available in the Nature Research Reporting Summary linked to this article.

## Data availability

The X-ray crystal structure of HypocratesCS was deposited in the protein data bank under accession code 6ZUI. The entry has been assigned the following PDB DOI: [https://doi.org/10.2210/pdb6ZUI/pdb]. The mass spectrometry data have been deposited to the ProteomeXchange Consortium via the PRIDE partner repository with the dataset identifier PXD029624 [https://doi.org/10.6019/PXD029624]. The additional raw data files underlying the Source Data are available upon reasonable request from the corresponding authors, requests will be answered within 2 weeks. Source data are provided with this paper.

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

## Acknowledgements

The work was supported by the Russian Foundation for Basic Research (RFBR) Grant 18-34-20032 (to D.S.B.); RFBR and Moscow city Government according to the project № 21-34-70031 (to D.S.B.); the Russian Science Foundation (RSF) Grant 17-15-01175 (to D.S.B) for work related to the preparation and testing of Hypocrates biosensor in the eukaryotic system; the Ministry of Science and Higher Education, grant 075-15-2019-1933 (to A.I.K., A.S.P., D.S.B., and V.V.B.); RFBR Grant 18-29-08049 for gel filtration experiments (to A.Yu.G.); the Grants from the Vlaams Instituut voor Biotechnologie (to J.M.); the Research Foundation-Flanders grant for the anaerobic workstation no. 1508316N (to IVM); the Research Foundation-Flanders–Fonds de la Recherche Scientifique Excellence of Science project no. 30829584 (to J.M. and D.V.); a FWO Ph.D. fellowship grant (to M-A.T.); CNRS, INSERM, Collège de France and Université de Paris (MT and SV). We thank Prof. Ursula Jakob for providing the NemR$^{C106}$ plasmid; Prof. Haike Antelmann for providing the HypR plasmid; Prof. Maria Lagarkova for the consultation on the neutrophil isolation technique; the beamline scientists at the Proxima 2 beamline of the Soleil synchrotron facility and the beamline scientist Pierre Legrand at the Proxima 1 beamline of the Soleil Synchrotron facility for his help with data processing. This work was partially performed using the equipment of Center for Precision Genome Editing and Genetic Technologies for Biomedicine, Pirogov Russian National Research Medical University, Moscow, Russia.

## Author contributions

A.I.K. developed architecture and design of the working version of Hypocrates biosensor, performed the in vitro experiments, analyzed and combined data, and wrote the manuscript. M-A.T. crystallized, collected data, and solved the structure of HypocratesCS, performed the in vitro circular dichroism experiments, pKa determination, fluorescence selectivity and sensitivity experiments, pre-steady-state kinetic measurements of Hypocrates and HypR/NemR, prepared Hypocrates MS samples with different oxidizing agents, analyzed data, and wrote the manuscript. A.S.P. performed experiments in eukaryotic cell culture. M.T. performed experiments in zebrafish. R.I.R. performed the in vitro experiments and analyzed the data. D.E. performed selectivity and kinetic assays, analyzed the data, and edited the manuscript. K.W. helped with the crystal conditions optimization and performed kinetics

measurements. I.V.M. helped with X-ray structure refinement, and X-ray data deposition. A.D.S. performed experiments with neutrophils. D.V. collected the mass spectrometry data and analyzed the MS data. A.Yu.G. performed gel filtration experiments. M.S.B. synthesized chemical compounds (NCT, NaOBr, ONOO⁻). S.V. supervised the in vivo work. J.M. supervised the in vitro kinetic and structural studies, analyzed data, and wrote the manuscript. D.S.B. and V.V.B. created the general concept of the project, supervised the work of the project at all stages, wrote the manuscript.

## Competing interests

The authors declare no competing interests.
