## [Peer Review File · Nature Communications]

Hypocrates is a genetically encoded fluorescent biosensor for (pseudo)hypohalous acids and their derivativesReviewers' Comments:

Reviewer #1:

Remarks to the Author:

This is an interesting manuscript describing the production and characterisation of a genetically encoded, ratiometric YFP -based probe that recognises hypochlorous acid and taurine chloramine. It is a neat approach to use the bacterial transcription factor NemR, this is the first such probe available and I can see multiple uses for it. The study is generally well performed and sound, but I do have a number of queries about validation that need to be addressed.

Major comments

1. Page 4 Line 24, Supplementary Figure 1 and kinetic data on p9 and Fig 3. I have major concerns about the rate constants determined for NemR and Hypocrates. Relative to values determined for other thiols, the HOCl ones are incredibly slow and the TauCl ones much higher. For a sensor supposedly sensitive to HOCl, a low rate seems counterintuitive and most unlikely. For example, the rate constant of $1.2 \times 10^8 \text{ M}^{-1}\text{s}^{-1}$ measured by Pattison and Davies for HOCl and GSH at pH 7.4 (FRBM 2014) is 1000 times faster than the values reported here. Reported rate constants for other thiols are also all at least in the 10^7 range (see refs 6 and 47). In contrast (see ref 13) the rate constant for TauCl with GSH is $118 \text{ M}^{-1}\text{s}^{-1}$. With other thiols it does increase with the extent of ionization, but the highest is 970 with thiodinitrobenzoic acid (which tends to be the most reactive thiol in experimental studies). So could it be that the NemR centre is super-reactive with TauCl but very unreactive with HOCl? As this is so unlikely, the assay needs to be revisited and the values validated by another method. Several points come to my mind. Are the fluorescent changes actually measuring the initial reaction of the HOCl/chloramine, to give a sulfenyl chloride, or are they monitoring a secondary structural change in the protein? Could there be a problem with removing all the DTT used to generate the reduced starting material? I note in the methods 30 mM DTT and one spin column was used. If only 0.1% remained, this would interfere. In our experience in related situations, we have had to use a second spin column.

2. Page 7 last para and Fig 2D. There are several oxidants that have not been tested but highly relevant when studying peroxidases and HOCl. HOBr, bromamines and HOSCN are produced by MPO and related peroxidases, and I would expect the bromine derivatives and possibly HOSCN to react with the probe. This is not necessarily a disadvantage, but it is important to know. It would be desirable, therefore, to test these reagents. At the very least, the authors need to be more cautious in claiming specificity without this information.

3. Page 9. See above re the kinetic data. I also wonder if the stoichiometry can be addressed in more detail with an explanation for why considerable excess of HOCl was required for maximum fluorescence change. In reference 32, the authors speculate (but do not prove) that the NemR oxidation product is a sulfenamide. Is there any supporting evidence in the current study? Others who have studied -S-N- crosslinking (see for example Fu et al Biochemistry 2002, 41, 1293) have observed sulfenamides converting to sulfinamides and sulphonamides at higher HOCl exposure. Is this relevant here?

Minor comments

1. Page 7 last sentence. I'm not sure what can be concluded from the Trp fluorescence change. It could be due to direct oxidation (see refs 6 and 47) or relate to C106.

2. In addressing sensitivity of cells to HOCl, the authors should consider (as reported by others) that because HOCl is so rapidly consumed by cells, the amount of oxidant per given number of cells is usually more relevant than absolute concentration. For example, a solution of $x \text{ M}$ HOCl will have more effect if the number of cells is halved, as will doubling the volume of solution applied. Wording could be adjusted here and in Discussion to account for this.

3. Figure 6 legend could be clearer. It is stated that values were normalised against uncut embryos. Does this imply that dotted lines were normalised against themselves? Please identify the two graphs in B.

4. Page 16 line 18. These pseudo 1st order rate constants have no meaning without knowing the conditions. Some dye probes have been used successfully in real time. I don't think it is necessary to knock these in order to justify the probe developed here.

5. Page 17. The estimate of mM HOCl was actually made by Weiss (NEJM 1989) and just referred to in ref 47. As the authors point out, HOCl tends not to accumulate, and even Weiss's calculation relates to how much is produced rather than an absolute concentration, so I am very cautious about using such figures. As discussed in point 2, it is more a matter of the amount produced and certainly this can be large at inflammatory sites, and the authors could consider changing their wording accordingly. Similarly, in relation to peroxynitrite, comparing absolute concentrations may not be the most meaningful as amounts produced and alternative reactions are important. I recommend being more cautious in discussing ONOO⁻ based on concentrations and not going into such detail in trying to exclude it. Pointing out the apparently lower sensitivity and ability to control for it may be sufficient.

Reviewer #2:

Remarks to the Author:

This manuscript by Belousov and co-workers developed a genetically encoded hypochlorite sensor, Hypocrates. Inspired by the hypochlorite-responsive transcriptional factor NemR, the authors constructed a cpYFP-based hypochlorite sensor, Hypocrates. The authors next characterized the selectivity, sensitivity and kinetics of the sensor protein. The crystal structure of the control form of the sensor, HypocratesCS, was also solved and analyzed. Finally, the authors performed experiments to detect hypochlorite in cells and in a zebrafish tail fin injury model.

The sensor development part of this work is generally well done. However, the introduction and study on the physiological role of hypochlorite in this manuscript is not enough, which may diminish the scientific significance of developing a hypochlorite sensor. There are also several issues on characterization of the sensor. Thus, this reviewer considers the paper needs major revisions to satisfy the requirements to be published in Nature Communications.

Major comments:

1. The authors should make a more detailed introduction on the biological role of hypochlorite and its derivatives. For example, what is the difference between hydrogen peroxide and hypochlorite on their sources, damage effects and/or signal transduction? The authors mentioned that high taurine concentrations in neutrophils lead to formation of N-chlorotaurine (NCT). For the formation and functions of N-chloramine, do they vary largely between different types of cells? In order to explain the importance of this work better, the backgrounds need to be well-organized.

2. In Fig. 2c, the authors demonstrated the pH-responsive curve of Hypocrates. Since the response of Hypocrates to pH is much larger than the response to the analyte, this reviewer suggests the authors to add a chart/figure showing the Ex500/Ex417 ratio change (might be normalized, y axis) vs. oxidant concentrations (x axis) under different pH values, which may better indicate the performance of Hypocrates over a wide range of pH conditions (just like fig. 1f in Nat. Methods 2017, 14, 720-728). In addition, different excitation wavelengths were used in Fig. 2,3 (Ex500/Ex417) and Fig. 5 (Ex500/Ex425). This reviewer suggests Fig. 2,3 to be replotted as Ex500/Ex425 since 425 nm is the excitation peak of the sensor and should be available in the measurement *in vitro*.

3. In Page 7, the first paragraph of the section "The selectivity of Hypocrates", the authors demonstrated that ~100 μ M NaClO led to fluorescence quenching due to apparent protein damage. However, in the next paragraph the authors incubated Hypocrates with 250 μ M NaClO in the circular dichroism (CD) experiment (according to Page 32, Line 11) and found the conformational change under this condition was able to be reversed by 200 μ M DTT. Therefore, the authors should modify the demonstration on "apparent protein damage" since their CD experiment did not support this conclusion. It is possible that high concentration of NaClO or ONOO⁻ only affects the chromophore of Hypocrates, but in order to support this speculation, the authors should add

HypocratesCS or cpYFP, and DTT treatment as controls in Supplementary Fig. 5 to prove that high concentration of NaClO cause damage to the chromophore of the sensor. Current Supplementary Fig. 7A is not enough because only low concentration of NaClO was tested.

4. For the selectivity profiling of Hypocrates, recently there was another research on development of genetically encoded sensor published, which is also based on formation of Cys-Lys crosslinking and the sensor is responsive to formaldehyde (FA) and methylglyoxal (MG) (Nat. Commun. 2021, 12, 581). Since aldehyde electrophiles including glyoxal (GO), MG and acrolein were shown to affect the DNA-binding affinity of NemR and/or induce the expression of nemRA-gloA operon (ref. 35), the authors may consider to profile sensor selectivity on some other aldehyde electrophiles besides GO, e.g. FA, MG and acrolein.

5. The authors should inspect their data processing of the crystal structure. The anisotropic correction by STARANISO server can truncate the non-significant data and the ellipsoidal completeness are nearly always better than the spherical completeness. However, in Supplementary Table 2 ellipsoidal completeness appears to be worse. Was the diffraction data processed appropriately (e.g. space group, scaling)? The authors should explain this in the method part. Also, the resolutions of different directions given by STARANISO server should be reported. If there is low anisotropy, the authors may consider not to perform anisotropic correction, since the overall spherical completeness is acceptable. Another issue is the resolution limit. Ideally the completeness of the highest shell should be over 85%. The authors may cut down the resolution to 2.2 Å or lower and this will not affect the conclusions in the manuscript.

6. For description on the crystal structure of HypocratesCS, the authors should put more efforts on discussion of the Hypocrates functioning. For example, how will the linker change affect the structure? Would fusion of cpYFP affect dimerization of NemR? Are there any other interactions on the surface between cpYFP and NemR?

7. It is important to verify whether Hypocrates is able to sense endogenous HClO, which is lacked in the current manuscript. The zebrafish tail fin injury model is not appropriate because there are too many signal changes incorporated during the wounding. In the introduction section and Fig. 5A,B, the author stated that the level of HClO is closely relate to heme enzyme myeloperoxidase (MPO). This supplied a simpler model to test performance of Hypocrates. The authors may change the MPO level with overexpression, inhibition, knockdown or knockout of MPO in cells and test the HClO concentration changes with Hypocrates.

8. For the application of Hypocrates in the zebrafish tail fin injury model, the author should include the control version of HyPerRed (HyPerRed-C199S) as they did in ref. 27. Also, the authors should make more explanation and discussion on their results. It was shown that after tail amputation, the hydrogen peroxide level increased in 20 min and then decreased, while the hypochlorite level is more persistent. Why did the changes of these two oxidants' level differ from each other? The authors may compare the localization of signal increase between hydrogen peroxide and hypochlorite, to analyze whether they are produced in the same area of the injured tissue. The authors may also investigate the role of hypochlorite production in inflammation, for example, to examine whether the immune cells are recruited to the same area where hypochlorite is produced, and whether the cytokine level in this area is increased.

Minor comments:

1. In Supplementary Fig.1, the authors applied 2.1 mM NaOCl in measurement of HypR intrinsic fluorescence. Would ~2 mM NaOCl induce apparent protein damage on HypR? (also see major comments 3)

2. In Page 7, Line 28, the authors wrote "the probability of such a response in biological systems is low because of the low sensitivity to ONOO- and its low cellular concentration". The authors should cite reference here, or note that "details in discussion".

Reviewer #3:

Remarks to the Author:

This report describes the first fluorescent protein based biosensor for HOCl. There are many strengths, such as the originality, the discussion of the chromophore, the crystal structure, the zebrafish study, etc. The following concerns should be addressed:

1. On line 18 page 4, the authors refer to "HypR" but this has not been defined.
2. The sensor detects both HCIO and NCT. Why does this not create concern about selectivity/interference?
3. The authors criticize other (small molecule) probes for slow kinetics, for taking "dozens of seconds and even minutes" to respond to HOCl. However, in the experiments herein, the authors describe a similar response time by their sensor (e.g., to DTT treatment to \show reversibility). Despite the authors' claims, there are examples of published small abiotic probes for HOCl that are rapid (a few seconds or less) as well as examples that are reversible and ratiometric. It is not fair to the researchers who created such probes to dismiss all of them together as having big drawbacks.
4. Another criticism has to do with comments about other probes not being useful for reversibility over several cycles; however, have the authors proven the validity of this property in their sensor.
5. The probe is pH-sensitive. Why is this not an issue since pH is not the same in all regions of the cell, under different physiological states, etc.?

In summary, this is an interesting paper about a probe with potential utility for HOCl study. However, there are some claims that are not substantiated and a lack of clarity about interfering species (NCT) and pH that leave the reader wondering.

Response to Reviewers

Dear Reviewers,

We would like to thank you for the careful review of our paper. We have considered all your comments, introduced additional experiments, and made the necessary corrections.

Reviewer 1

Comment 1 (R1): *Page 4 Line 24, Supplementary Figure 1 and kinetic data on p9 and Fig 3. I have major concerns about the rate constants determined for NemR and Hypocrates. Relative to values determined for other thiols, the HOCl ones are incredibly slow and the TauCl ones much higher. For a sensor supposedly sensitive to HOCl, a low rate seems counterintuitive and most unlikely. For example, the rate constant of $1.2 \times 10^8 \text{ M}^{-1}\text{s}^{-1}$ measured by Pattison and Davies for HOCl and GSH at pH 7.4 (FRBM 2014) is 1000 times faster than the values reported here. Reported rate constants for other thiols are also all at least in the $e7$ range (see refs 6 and 47). In contrast (see ref 13) the rate constant for TauCl with GSH is $118 \text{ M}^{-1}\text{s}^{-1}$. With other thiols it does increase with the extent of ionization, but the highest is 970 with thiodinitrobenzoic acid (which tends to be the most reactive thiol in experimental studies). So could it be that the NemR centre is super-reactive with TauCl but very unreactive with HOCl? As this is so unlikely, the assay needs to be revisited and the values validated by another method. Several points come to my mind. Are the fluorescent changes actually measuring the initial reaction of the HOCl/chloramine, to give a sulfenyl chloride, or are they monitoring a secondary structural change in the protein? Could there be a problem with removing all the DTT used to generate the reduced starting material? I note in the methods 30 mM DTT and one spin column was used. If only 0.1% remained, this would interfere. In our experience in related situations, we have had to use a second spin column.*

Response 1 (R1): We thank the Reviewer for pointing this out. From the Pattison and Davies paper in Chem. Res. Toxicol. of 2001 it became clear that the reactivity of HOCl with different compounds strongly depends on the target and the reaction with sulfurs in Cys and Met reaches values in the range of $10^7 \text{ M}^{-1}\text{s}^{-1}$, while on the other hand reactivity with alpha-amino group ($10^5 \text{ M}^{-1}\text{s}^{-1}$) and Lys ($5 \times 10^3 \text{ M}^{-1}\text{s}^{-1}$) are in a different range. The reaction rates were in 2014 re-evaluated by Storkey et al. in FRBM using a competition kinetic approach where reaction rates towards Cys, Met, and GSH become even more spectacular reaching levels of $\sim 10^8 \text{ M}^{-1}\text{s}^{-1}$. Before such a reaction can occur both reagents need to be accessible, and this is not the case for C106 in NemR which suggests that the direct

environment of C106 determines its reactivity. If we take a closer look at the site of C106 in the structure of NemR (PDB ID: 4YZE), you will notice that C106 is in a hydrophobic core surrounded by Leu, Val, Trp. In order to bring this Cys out of this hydrophobic pocket and to make it possible to react with Lys to form a sulfenamide, like pointed out by Gray et al. in JBC 2013 and ARS 2015, a local structural rearrangement is needed. All in all, the reaction rates that we measure here are in the range that we can expect for NemR with a more buried Cys residue.

Of note, by incorporating NemR in cpYFP we also lose a factor of 10 in reaction rate. We noticed a similar phenomenon for the reaction of H₂O₂ with OxyR in comparison with the reaction of H₂O₂ with the biosensor Hyper7.

In order to take this possible confusion away for the reader, we have adapted the text:

«Based on these observations, we selected NemR^{C106} as the molecular platform to design a biosensor for HOCl detection. NemR^{C106} consists of a DNA-binding and a sensory domain (Fig. 1A). The latter has a flexible loop with the crucial Cys106 buried in a hydrophobic pocket surrounded by Trp167, Leu168, and Val109. Notably, its measured reaction rate with NaOCl is ~1000 times slower compared to what has been published for a free cysteine⁴⁶, which indicates that the local environment strongly determines the reactivity. The Cys106 residue has been observed in two conformations in the crystallographic asymmetric unit, with the transition reportedly caused by HOCl⁴² (Supplementary Fig. 3). In order to visualize these conformational changes, we introduced cpYFP in several positions of the flexible loop. Like for other cpYFP-based probes^{34,38,47}, we expected that, upon proper conformational coupling, structural shifts in the flexible loop induced by the reaction with HOCl would alter the optical properties of the chromophore (Fig. 1B).»

Regarding the concern of the use of biospin columns to remove DTT, there might have been a slight confusion. For all the kinetic data obtained, after reducing the protein with DTT, HiTrap desalting column on an AKTA pure system was used. Within the materials and methods section the corresponding information is written for all the *in vitro* assays, which required a reduced protein at the start. We only have used the biospin columns in one experiment where the protein was already reduced by DTT and desalted using the HiTrap column. The reduced protein was then treated with NaOCl or H₂O₂. The biospin column was used to remove the oxidant and the protein was immediately used for the CD data measurement. We hope that this clarifies the confusion.

Comment 2 (R1): *Page 7 last para and Fig 2D. There are several oxidants that have not been tested but highly relevant when studying peroxidases and HOCl. HOBr, bromamines and HOSCN are produced by MPO and related peroxidases, and I would expect the bromine derivatives and possibly HOSCN to react with the probe. This is not necessarily a disadvantage, but it is important to know. It would be desirable, therefore, to test these reagents. At the very least, the authors need to be more cautious in claiming specificity without this information.*

Response 2 (R1): Initially, when we started to develop Hypocrates biosensor based on a bacterial NemR, we were guided by the information in which NemR is characterized as a natural sensor for HOCl and electrophiles [Gray et al., Does the Transcription Factor NemR Use a Regulatory Sulfenamide Bond to Sense Bleach? Antioxid Redox Signal. 2015; Gray et al., NemR is a bleach-sensing transcription factor. J Biol Chem. 2013]. We used the version NemR^{C106} with only a single Cys106 to exclude interactions with electrophiles. However, as Reviewer 1 correctly noted, we did not really test the interaction of Hypocrates with HOBr and HOSCN.

To do this, we synthesized pure NaOBr and HOSCN. Indeed, Hypocrates demonstrates a pronounced change in the spectrum upon oxidation by these components in a similar manner to NaOCl and NCT. Thanks to your comment, we completely changed the concept of our paper and now we are positioning Hypocrates as a biosensor for the monitoring of (pseudo)hypohalous acids and their derivatives. Hypocrates exhibits a similar sensitivity to HOCl, HOBr and HOSCN, which is especially important given that these compounds are produced as a mixture in biological systems.

In connection with the new concept, we completely adapted the text and conducted the necessary experiments:

1. We investigated the change in spectral characteristics:

« Treatment of the purified protein with NaOBr, another hypohalite anion, elicited a comparable signal shift of the probe (Supplementary Fig. 5). Next, we decided to test whether Hypocrates is also sensitive to the HOCl-derivative, N-chlorotaurine (NCT). Taurine is one of the most abundant amino acids in many tissues⁴⁸. Moreover, NCT is one of the most common derivatives of reactive chlorine species, because taurine concentrations in neutrophils are relatively high⁴⁹. We demonstrated that NCT treatment led to a fully reversible ratiometric

response of the protein (Fig. 1G). HOSCN, a pseudohypohalous acid, which is chemically similar to HOCl and HOBr, but acts as a weaker and more selective oxidizing agent, demonstrated comparable behavior (Supplementary Fig. 5). Thus, Hypocrates can be considered as a biosensor for (pseudo)hypohalous acids and their derivatives».

And we changed Supplementary Fig. 5:

Supplementary Figure 5. (a-d) The comparison of Hypocrates fluorescence excitation spectra in the presence of several oxidants: (a) NaOCl, (b) NCT, (c) NaOBr, and (d) HOSCN, as well as the

reversibility of responses by DTT. (e-h) The degradation resistance of Hypocrates in the presence of high concentrations of several oxidants: (e) NaOCl, (f) NCT, (g) NaOBr, and (h) NaONOO. Only NCT does not induce fluorescence quenching, apparently due to its lower reactivity and higher specificity toward sulfur containing amino acids. In all panels, except for d, protein concentrations were 0.5 μ M. In panel d, protein concentration was 2 μ M. Panels a, b, e, f were recorded in PBS. Panels c, d, g, h were recorded in 100 mM sodium phosphate buffer, pH 7.4 to avoid possible OCl⁻ generation.

- We have expanded the biosensor selectivity diagram and changed the Fig.2D:

- We found that NaOCl, NCT and NaOBr caused Trp-fluorescence changes (λ_{ex} = 295 nm, λ_{em} = 350 nm) of original NemR^{C106}, while HOSCN showed no response, indicating that the NemR-derived domain in Hypocrates gained the ability to sense this compound due to the integration of cpYFP. We also showed that this Trp is not oxidized (see response to Comment 4 and Supplementary Table 2). We have completed the diagram in Suppl Fig.8 D:

Supplementary Figure 8. The response of different cpYFP-based probes to NaOCl. **(a)** In the presence of **low concentrations of NaOCl**, the fluorescence excitation spectrum of purified cpYFP shows minor quenching due to apparent protein damaging. **With an increase in the concentration of the oxidant, the signal becomes dramatically quenched.** **(b)** The spectral changes of cpYFP treated with NaOCl are almost irreversible and cannot be reduced by DTT. **(c)** The fluorescence excitation spectrum of the purified pH biosensor SypHer3s is resistant to NaOCl treatment. **(d)** Purified H₂O₂ biosensor HyPer2 does not react with NaOCl, while the addition of H₂O₂ induces a pronounced ratiometric response. **(e)** The intrinsic Trp fluorescence changes of initial NemR^{C106} in the presence of NaOCl, NaOBr, HOSCN, and NCT. The data are presented as a mean ± SEM, n = 3. Protein concentration was 0.5 μM in panels a-d and 2 μM in panel e. **Panels a-d were recorded in PBS, while panel e was recorded in 100 mM sodium phosphate buffer to avoid possible OCl⁻ generation.**

- We have updated the information on the sensitivity and reaction rates of Hypocrates, as a result we have changed Figure e3:

Figure 3. Hypocrates sensitivity and reaction rates. Changes in the fluorescence excitation spectra of Hypocrates (0.5 μM) obtained by additions of (a) NaOCI or (b) NCT aliquots. (c) Titration curves of Hypocrates (0.5 μM) in sodium phosphate buffer obtained by additions of NaOCI, NaOBr, HOSCN or NCT aliquots. The data are presented as the mean \pm SEM, $n \geq 2$. The maximum amplitudes of response are 2.0-, 1.8-, 1.8- and 1.7-fold for NaOBr, HOSCN, NaOCI and NCT, respectively. In the presence of NaOBr, NaOCI, and NCT, the probe is saturated at approximately 5 μM , and for HOSCN at approximately 1 μM . (d) Hypocrates sensitivity towards NaOCI, NaOBr, NCT, and HOSCN is shown. The data are presented as the mean \pm SEM, $n \geq 2$. (e-g) Hypocrates reaction rates. Changes in cpYFP fluorescence at >515 nm cut-off ($\lambda_{\text{ex}} = 485$ nm) were measured as a function of time (insert). The curves were fitted to a single exponential to obtain the observed rate constants (k_{obs}), which were plotted as a function of different (e) NaOCI, (f) NCT or (g) NaOBr concentrations. The second-order rate constants for NaOCI ($3.7 \times 10^5 (\pm 1.1 \times 10^3) \text{M}^{-1} \text{s}^{-1}$)

$1.1 \times 10^3 \text{ M}^{-1}\text{s}^{-1}$), NCT ($4.3 \times 10^4 (\pm 1.8 \times 10^3) \text{ M}^{-1}\text{s}^{-1}$) and NaOBr ($4 \times 10^5 (\pm 1.3 \times 10^4) \text{ M}^{-1}\text{s}^{-1}$) were determined from the slope of the straight line. The data are presented as the mean \pm SD, $n \geq 3$.

5. We also tried to characterize Hypocrates with respect to HOBr-derivatives. However, bromamine synthesis protocols imply the presence of OCl^- impurities. Therefore, we were unable to synthesize a pure bromamine preparation. Alternatively, we tested the interaction of the biosensor (0.5 μM) with a potent brominating agent NBS (N-bromosuccinimide, a model of a bromimide). Low concentrations of NBS (about 1 μM) lead to the characteristic ratiometric response of Hypocrates. But already starting from a concentration of 10 μM , the biosensor begins to degrade, we registered a decrease in fluorescence:

It is noteworthy that upon oxidation caused by 1 μM of NBS, the sensor can be completely reduced by DTT. However, at higher concentrations of NBS, the protein is irreversibly degraded:

NBS is a potent brominating agent, which appears to participate in non-specific interactions with the protein at high concentrations. Since this substance is not a component that can be found in cell biochemistry, we did not include this experiment to this manuscript.

6. We have made all the necessary edits to the Sections: abstract, introduction, results, discussion and methods.

Comment 3 (R1): Page 9. See above re the kinetic data. I also wonder if the stoichiometry can be addressed in more detail with an explanation for why considerable excess of HOCl was required for maximum fluorescence change. In reference 32, the authors speculate (but do not prove) that the NemR oxidation product is a sulfenamide. Is there any

supporting evidence in the current study? Others who have studied -S-N- crosslinking (see for example Fu et al Biochemistry 2002, 41, 1293) have observed sulfenamides converting to sulfinamides and sulphonamides at higher HOCl exposure. Is this relevant here?

Response 3 (R1): When choosing a sensory domain for a future biosensor, designers are looking for characterized proteins with the required properties of interest. We did the same, choosing NemR protein as the basis for Hypocrates biosensor. We were not faced with the task of testing the mechanism of NemR functioning, so we referred to the work conducted earlier. Having engineered NemR-cpYFP construct, we characterized in detail its properties, and came to the conclusion that this protein molecule can be considered as a biosensor. Most biosensors follow this path of creation. And in some cases, the properties of the sensory domain may differ from the properties of the original selected protein, as well as the properties of the fluorescent protein in the composition of the chimeric construct may differ from the original ones. Therefore, in the case of any biosensor, the most important thing is to characterize in detail its properties and behavior in various systems, which we have done in the case of Hypocrates. We agree that the mechanism of functioning of the probe represents valuable information. In the context of this comment, we conducted the following studies:

1. In a previous version of the manuscript, we described the development of HypocratesCS variant in which Cys355 was substituted for Ser355. This variant did not respond to oxidation. By analogy, we decided to develop a variant with K424A mutation, since Lys424 is supposed to be involved in the formation of the -S-N- bond. However, we found that this version, HypocratesKA, is highly sensitive to NaOCl. Thus, Cys355 is necessary for the functioning of the biosensor, but Lys424 is not. This observation suggests that the formation of the -S-N- bond is not required for the response of Hypocrates. We added a new graph (Fig.11):

2. In the work referred by the Reviewer, the authors used mass spectrometry for analysis. However, in their studies, this approach was applied for a short peptide; in our case, it is necessary to analyze the protein, which greatly complicates the implementation of the task. However, we performed such analysis for Hypocrates. We didn't observe the -S-N- formation with the MS analysis of the biosensor treated with NaOCl, NCT, HOSCN or NaOBr.

3. We have adapted the text accordingly.

4. As for the other part of the comment, the increased stoichiometry can be explained by the fact that different residues can be halogenated. However, even if they are modified, this is not reflected in the signal. We are testing a biosensor, so it is the behavior of the fluorescent response that needs to be considered, even if other interactions may occur that do not lead to a change in spectral characteristics. The same applies to the issue of the -S-N- bond formation. Even if it is not formed, the biosensor demonstrates changes in signal when interacting with hypohalous acids. In doing so, we have shown that Cys355 plays a key role in that matter. In our opinion, the study of protein modifications under conditions of hypohalous stress is the subject of a separate paper. It cannot be ruled out that the oxidation of Cys355 alone may be sufficient to generate a response.

Comment 4 (R1): Page 7 last sentence. I'm not sure what can be concluded from the Trp fluorescence change. It could be due to direct oxidation (see refs 6 and 47) or relate to C106.

Response 4 (R1): Using mass spectrometric analysis, we have shown that this Trp in the tested conditions is not modified (Supplementary Table 2).

We have added:

«This change in Trp-fluorescence is not related to direct oxidation of the Trp residue, which we could confirm with mass spectrometry (Supplementary Table 2)».

Supplementary Table 2. Label free quantification of Trp416 oxidation. Relative abundance of the $[M+3H]^{3+}$ precursor ion corresponding to the peptide sequence ENHSLTFSGEPLQQAQVLYALWLGANLQAK was determined from extracted ion chromatograms (XIC) and quantified by the area under the curve (AUC) taking into account the formation of oxindolylalanine (+15.99 Da) or N-formylkynurenine (+31.98 Da) under the different conditions. Oxidation of the Trp residue was confirmed by MS/MS fragmentation of the different parent ions (data not shown).

% Oxidation	W	control	NaOCl	NCT	HOSCN	NaOBr
mono oxidation						
		8.1	4.5	5.9	12.1	4.4
di oxidation						
		2.0	2.8	1.6	3.9	1.7

Comment 5 (R1): In addressing sensitivity of cells to HOCl, the authors should consider (as reported by others) that because HOCl is so rapidly consumed by cells, the amount of oxidant per given number of cells is usually more relevant than absolute concentration. For example, a solution of x M HOCl will have more effect if the number of cells is halved, as will doubling the volume of solution applied. Wording could be adjusted here and in Discussion to account for this.

Response 5 (R1): We have added:

«Exposure to 40 μ M NaOCl led to a signal change of 1.8-fold, which is similar to the saturating response obtained with purified protein and in *E. coli* suspension (Fig. 1E,F). It should be noted that since NaOCl rapidly reacts with biological objects, in such experiments it is important to take into account not only the absolute concentration of the NaOCl solution, but also the number of cells.»

We also added information about the number of cells in the caption to Figure 5:

«(D) Upper part: Hypocrates fluorescence changes induced by 40 μ M NaOCl (values \pm SEM, N = 2 experiments, n \geq 30 cells per experiment, 236,110 \pm 34,560 cells per dish). Lower part: Images of Hypocrates in transiently transfected HeLa Kyoto cells exposed to 40 μ M of NaOCl at different time points. Scale bar = 50 μ m. The lookup table indicates changes in the Ex₅₀₀/Ex₄₂₅ ratio.»

Comment 6 (R1): *Figure 6 legend could be clearer. It is stated that values were normalised against uncut embryos. Does this imply that dotted lines were normalised against themselves? Please identify the two graphs in B.*

Response 6 (R1): We have added:

«For control embryos (dotted lines), fluorescence has been normalized for each embryo to the first image of the time lapse (t = 0).»

Comment 7 (R1): *Page 16 line 18. These pseudo 1st order rate constants have no meaning without knowing the conditions. Some dye probes have been used successfully in real time. I don't think it is necessary to knock these in order to justify the probe developed here.*

Response 7 (R1): We have removed this paragraph.

Comment 8 (R1): *Page 17. The estimate of mM HOCl was actually made by Weiss (NEJM 1989) and just referred to in ref 47. As the authors point out, HOCl tends not to accumulate, and even Weiss's calculation relates to how much is produced rather than an absolute concentration, so I am very cautious about using such figures. As discussed in point 2, it is more a matter of the amount produced and certainly this can be large at inflammatory sites, and the authors could consider changing their wording accordingly. Similarly, in relation to peroxyntite, comparing absolute concentrations may not be the most meaningful as amounts produced and alternative reactions are important. I recommend being more*

cautious in discussing ONOO⁻ based on concentrations and not going into such detail in trying to exclude it. Pointing out the apparently lower sensitivity and ability to control for it may be sufficient.

Response 8 (R1): We have changed the discussion section and made all the wording more accurate, without naming specific concentrations.

Discussion

Our idea of developing a genetically encoded sensor for (pseudo)hypohalous acids detection was born out of increasing amounts of incoming information about proteins that are specifically modified under conditions of hypohalous stress. We present the Hypocrates probe, which is the first genetically encoded fluorescent biosensor for visualizing (pseudo)hypohalous acids in live systems. Hypocrates displays a ratiometric reversible change in signal when interacting with (pseudo)hypohalous acids and their derivatives with minimal response-inducing oxidant concentrations located in the 0.1-0.3 μM range (at a biosensor concentration of 0.5 μM). It is known that neutrophils produce high amounts of reactive chlorine species. However, it is difficult to calculate the exact concentrations of HOCl generated by cells since it quickly reacts with surrounding molecules. It has been reported that at the sites of inflammation, the amounts of HOCl generated by MPO from accumulated immune cells reach concentrations of 1-2 mM⁵⁶. The biosensor demonstrates similar sensitivities to NaOCl, NaOBr, and HOscn, which is why we are positioning Hypocrates as an indicator for visualizing the dynamics of the total pool of (pseudo)hypohalous acids. This is particularly relevant because (pseudo)hypohalous acids are produced as a mixture in biological systems. Hypocrates also allows monitoring the dynamics of HOCl derivatives. Chloramines are characterized by longer lifetimes due to decreased reaction rates and altered selectivity profiles with more pronounced specificity for sulfur-containing groups. Due to the high concentrations of taurine in neutrophils, N-chlorotaurine is one of the most common derivatives of reactive chlorine species⁴⁹. Recombinantly expressed and purified Hypocrates did not show any response to the major common intracellular oxidizing agents. However, we observed spectral changes of the sensor in the presence of ONOO⁻. The saturating concentration of ONOO⁻ that induced the maximum response of Hypocrates *in vitro* was \sim 10 μM at a biosensor concentration of 0.5 μM . Thus, in systems in which high ONOO⁻ production is expected, it is recommended to perform a control series of experiments, for example, using various inhibitors of NO[•] synthases, in order

to assess the contribution of ONOO⁻ to the formation of the biosensor response. The acidity of the medium affects the acid-base equilibrium of the chromophores inside the fluorescent proteins; therefore, the relative abundance of their protonated and deprotonated forms in solution largely depends on the pH value. The dependence on H⁺ concentration is especially pronounced for circular permutants of fluorescent proteins, including cpYFP that we used, since their chromophores are more accessible to the environment. To inactivate the sensor, we substituted the key Cys355 residue with a Ser and obtained the HypocratesCS variant, which is insensitive to hypohalous acids and their derivatives. Since its spectral properties and pH sensitivity are identical to the parameters of the original indicator, HypocratesCS is the most appropriate pH-control version.

By analogy with the C355A mutation, we expected that a K424A substitution would completely inactivate Hypocrates. However, the HypocratesKA variant was found to act very similar to the original sensor in terms of NaOCl sensitivity and response amplitude. This observation together with our mass spectrometrical data suggest that the NemR-sensor domain of Hypocrates does not function in a similar way to the *E. coli* transcription repressor NemR in forming a reversible sulfenamide bond^{42,44}, and that even the oxidation of Cys355 itself could be sufficient for signal changes in Hypocrates (Supplementary Fig. 14). We hypothesize that the oxidation of Cys355, which is located on a flexible loop next to Gly354, couples to Gln99 and this residue, in turn, couples to Asn95, which is in contact with the 4-hydroxybenzyl group of the chromophore (CR2) via a water molecule. Hence, oxidation of Cys355 mediated by (pseudo)hypohalous acids could induce changes in the hydrogen bonding network surrounding CR2 that lead to a ratiometric response. However, elucidating the precise sensing mechanism of Hypocrates is the subject of a separate study.

The crystal structure of HypocratesCS is the first cpFP-based redox biosensor that reveals its CR2 chromophore environment within its overall structure (Fig. 4A,C). Overall structure comparison of the β -barrel of cpYFP in Hypocrates (PDB ID: 6ZUI) with the calcium biosensor, Case16 (PDB ID:3O77)⁵², showed that both β -barrels are very similar (rmsd of 0.282 Å for 190 atoms). Further, both have a CR2 chromophore, while GFP is characterized by a CRO chromophore (Supplementary Fig. 11D). As such, the cpFP in Case16 is actually a cpYFP, and not a cpGFP as mentioned in Leder *et al.*⁵². The structure of HypocratesCS is not only important for understanding the functioning of this biosensor, but also for revealing the features of other cpYFP-based probes, which show subtle differences in their CR2 chromophore environment (Supplementary Fig. 11A,C), as well as for the rational structure-guided design of future cpYFP-based biosensors for other analytes.

As a defense mechanism against pathogens, immune cells generate hypohalous acids. We tested the functioning of Hypocrates by visualizing hypohalous stress in the bacterial cytoplasm of *E. coli* captured and swallowed by neutrophils. To accomplish this, *E. coli* cells expressing Hypocrates were added to human peripheral blood neutrophils while the signal of the probe was monitored with a fluorescence microscope. When individual bacteria entered the neutrophils, we registered a sharp increase in the ratiometric response. In the same system, the signal of HypocratesCS also increased, but to a lesser extent. These observations correspond to the previously obtained results from the literature, showing that the human neutrophil phagosome adopts an alkaline pH for several tens of minutes after phagocytosis^{57,58}. Thus, the dynamics of the inactivated HypocratesCS version most likely reflect alkalization of bacterial cytoplasm. With the use of HypocratesCS as a control, it, therefore, becomes possible to assess the contribution of the pH component to the Hypocrates signal.

Finally, Hypocrates is suitable for studying inflammatory reactions *in vivo*. In this work, we induced inflammation by injuring the caudal fin of *Danio rerio* larvae. Previously, it was shown with the HyPer biosensor that an H₂O₂ gradient is formed in the wound, which serves to attract neutrophils to the area of inflammation⁵⁴. In turn, neutrophils subsequently participate in the elimination of this gradient due to the reaction catalyzed by MPO⁵⁵. Here, for the first time, we observed the simultaneous real-time dynamics of H₂O₂ and (pseudo)hypohalous acids *in vivo* in zebrafish tissues during inflammation using the red fluorescent biosensor HyPerRed³⁷ and green emitting Hypocrates. This demonstrates how Hypocrates can be combined in a multiparameter microscopy mode with other biosensors with suitable spectral properties to disentangle the various reactive species involved in the inflammatory response, and beyond.

Reviewer 2

Comment 1 (R2): *The authors should make a more detailed introduction on the biological role of hypochlorite and its derivatives. For example, what is the difference between hydrogen peroxide and hypochlorite on their sources, damage effects and/or signal transduction? The authors mentioned that high taurine concentrations in neutrophils lead to formation of N-chlorotaurine (NCT). For the formation and functions of N-chloramine, do they vary largely between different type of cells? In order to explain the importance of this work better, the backgrounds need to be well-organized.*

Response 1 (R2): At the request of Reviewer 1, we checked whether Hypocrates biosensor is capable of interacting and changing its spectral properties in the presence of other biologically important compounds that are similar in properties to HOCl, primarily HOBr and HOSCN. NemR protein that we selected as a sensory domain was previously characterized as a natural sensor for HOCl and active electrophiles. However, we synthesized required compounds and showed that Hypocrates is equally sensitive to HOCl, HOBr and HOSCN. Thus, at the moment we are positioning Hypocrates as a biosensor for (pseudo)hypohalous acids and their derivatives. This is especially important in light of the fact that in biological systems, these compounds are produced by the same enzymes and form a mixture. We carried out all the necessary experiments in order to characterize the behavior of the fluorescent signal of Hypocrates after its oxidation with HOBr and HOSCN, by analogy with HOCl. Thus, we have significantly modified the paper, which includes changing the introduction section with notifications about the differences in the characteristics of (pseudo)hypohalous acids. Regarding taurine, taurine is one of the most abundant amino acids in many tissues (nervous system, muscles, retina, cells of the immune system and many others). We have also included this information in the text.

Comment 2 (R2): *In Fig. 2c, the authors demonstrated the pH-responsive curve of Hypocrates. Since the response of Hypocrates to pH is much larger than the response to the analyte, this reviewer suggests the authors to add a chart/figure showing the Ex500/Ex417 ratio change (might be normalized, y axis) vs. oxidants concentrations (x axis) under different pH values, which may better indicate the performance of Hypocrates over a wide range of pH conditions (just like fig. 1f in Nat. Methods 2017, 14, 720-728). In addition, different excitation wavelength was used in Fig. 2,3 (Ex500/Ex417) and Fig. 5 (Ex500/Ex425). This reviewer suggests Fig.2,3 to be replotted as Ex500/Ex425 since 425 nm is the excitation peak of the sensor and should be available in the measurement in vitro.*

Response 2 (R2): By analogy with the article referred to by the Reviewer 2, we conducted similar experiments. As a result, we have added Supplementary Figure 7.

Supplementary Figure 7. (a-d) Hypocrates fluorescence excitation spectra in the presence of different NCT concentrations at (a) pH 7.0; (b) pH 7.3; (c) pH 7.6; (d) pH 7.9 in 100 mM sodium phosphate buffers. (e) The same data represented as titration curves. Protein concentration was 0.5 μM in all cases.

Regarding the difference between EX_{417} and EX_{425} in Fig. 2,3 ($\text{EX}_{500}/\text{EX}_{417}$) and Fig. 5 ($\text{EX}_{500}/\text{EX}_{425}$), we explain this difference by the technical features of the devices that were used in these experiments. In one laboratory a spectrofluorometer with filters was used, in another – with a monochromator. All this information is detailed in the Materials and Methods section. In any case, the excitation wavelength does not have to correspond to the excitation maximum. For example, most laboratories use cpYFP-biosensors with the most popular 405 nm laser.

Comment 3 (R2): *In Page 7, the first paragraph of the section “The selectivity of Hypocrates”, the authors demonstrated that $\sim 100 \mu\text{M}$ NaClO led to fluorescence quenching*

due to apparent protein damage. However, in the next paragraph the authors incubated Hypocrates with 250 μM NaClO in the circular dichroism (CD) experiment (according to Page 32, Line 11) and found the conformational change under this condition was able to be reversed by 200 μM DTT. Therefore, the authors should modify the demonstration on “apparent protein damage” since their CD experiment did not support this conclusion. It is possible that high concentration of NaClO or ONOO- only affects the chromophore of Hypocrates, but in order to support this speculation, the authors should add HypocratesCS or cpYFP, and DTT treatment as controls in Supplementary Fig. 5 to prove that high concentration of NaClO cause damage to the chromophore of the sensor. Current Supplementary Fig. 7A is not enough because only low concentration of NaClO was tested.

Response 3 (R2): A slight confusion has occurred here. In the fluorescence spectra stability tests, we used the purified protein at a concentration of 0.5 μM . Therefore at 100 μM concentration of HOX, the protein/oxidant ratio was around 1:200. In circular dichroism tests, we used the purified protein at a concentration of 25 μM . Therefore, at 250 μM concentration of HOX, the protein/oxidant ratio was around 1:10. This information is mentioned in the Methods section. Moreover, the oxidant/protein ratio is directly written in Fig2A-B. As can be seen in the sensitivity tests (Fig.3), 1:10 is the saturating ratio for Hypocrates. Therefore, in circular dichroism tests we did not face the problem of protein destruction. To avoid possible misunderstandings, we have changed the wording in the text:

«We showed that Hypocrates is highly sensitive to (pseudo)hypohalous acids and their derivatives. It is noteworthy that high concentrations of NaOCl and NaOBr ($\sim 100 \mu\text{M}$ at a protein concentration of 0.5 μM), but not NCT, led to pronounced fluorescence quenching due to apparent protein damage, which further indicates that NCT reacts with the sensor in a more specific way (Supplementary Fig. 5).»

In any case, we decided to test whether the observed emission quenching in the presence of high oxidant concentrations is related to the damage of the fluorescent chromophore-containing domain. We have expanded the Supplementary Fig. 7A. We have shown that with an increase in the concentration of NaOCl, the purified cpYFP is dramatically quenched. This phenomenon is almost irreversible and the signal is not restored after incubation with a DTT. Thus, we are confident that extremely high concentration of NaOCl nonspecifically destroys the fluorescent protein, while the ratiometric response of the biosensor is determined by its sensory part.

Supplementary Figure 8. The response of different cpYFP-based probes to NaOCl. (a) In the presence of low concentrations of NaOCl, the fluorescence excitation spectrum of purified cpYFP shows minor quenching due to apparent protein damaging. With an increase in the concentration of the oxidant, the signal becomes dramatically quenched. (b) The spectral changes of cpYFP treated with NaOCl are almost irreversible and cannot be reduced by DTT. (c) The fluorescence excitation spectrum of the purified pH biosensor SypHer3s is resistant to NaOCl treatment. (d) Purified H₂O₂ biosensor HyPer2 does not react with NaOCl, while the addition of H₂O₂ induces a pronounced ratiometric response. (e) The intrinsic Trp fluorescence changes of initial NemR^{C106} in the presence of NaOCl, NaOBr, HOSCN, and NCT. The data are presented as a mean \pm SEM, n = 3. Protein concentration was 0.5 μ M in panels a-d and 2 μ M in panel e. Panels a-d were recorded in PBS, while panel e was recorded in 100 mM sodium phosphate buffer to avoid possible OCl⁻ generation.

We have also changed the main text:

«To validate whether the NaOCl-induced fluorescence changes are NemR^{C106}-derived, we treated purified cpYFP (0.5 μ M) and two other cpYFP-based biosensors (HyPer-2³⁵ and SypHer3s⁴⁹, 0.5 μ M) with NaOCl (5-10 μ M). HyPer-2 and SypHer3s showed no response. At the same concentration of NaOCl (5 μ M), a slight decrease in the intensity of cpYFP fluorescence was detected. This can be explained by bleaching since single cpYFP has a more open conformation, making the chromophore more accessible to the environment. With a significant increase in the concentration of NaOCl (up to 105 μ M), the fluorescence is almost completely quenched. This change in the signal is practically irreversible as shown by incubation with DTT, which indicates that non-specific degradation of the fluorescent protein

proceeds under tested conditions (Supplementary Fig. 8). As such, we concluded that cpYFP itself does not contribute to the ratiometric response.»

Comment 4 (R2): For the selectivity profiling of Hypocrates, recently there was another research on development of genetically encoded sensor published, which is also based on formation of Cys-Lys crosslinking and the sensor is responsive to formaldehyde (FA) and methylglyoxal (MG) (Nat. Commun. 2021, 12, 581). Since aldehyde electrophiles including glyoxal (GO), MG and acrolein were shown to affect the DNA-binding affinity of *NemR* and/or induce the expression of *nemRA-gloA* operon (ref. 35), the authors may consider to profile sensor selectivity on some other aldehyde electrophiles besides GO, e.g. FA, MG and acrolein.

Response 4 (R2): We decided to profile Hypocrates for three reactive electrophiles (GO, MG, and FA). No responses were observed in all cases (50 μ M GO, MG and FA). The new data are included in the manuscript and we have concluded that Hypocrates is not sensing these reactive electrophiles.

Comment 5 (R2): *The authors should inspect their data processing of the crystal structure. The anisotropic correction by STARANISO server can truncate the non-significant data and the ellipsoidal completeness are nearly always better than the spherical completeness. However, in Supplementary Table 2 ellipsoidal completeness appears to be worse. Was the diffraction data processed appropriately (e.g. space group, scaling)? The authors should explain this in the method part. Also, the resolutions of different directions given by STARANISO server should be reported. If there is low anisotropy, the authors may consider not to perform anisotropic correction, since the overall spherical completeness is acceptable. Another issue is the resolution limit. Ideally the completeness of the highest shell should be over 85%. The authors may cut down the resolution to 2.2 Å or lower and this will not affect the conclusions in the manuscript.*

Response 5 (R2): As the anisotropy of the data was indeed relatively weak (a^* 2.028 Å, b^* 2.047 Å, c^* 2.390 Å), and the contribution of anisotropic correction is neglectable, we decided to use the spherically processed data, as suggested by the Reviewer. In addition, we cut the resolution of the data to 2.2 Å. We adapted the processing and refinement parameters in Supplementary Table 3 accordingly, and updated the materials and methods section and results section.

Supplementary Table 3. X-ray data collection and refinement statistics.

HypocratesCS	
Data collection	
Space group	C2221
Cell dimensions	
a, b, c (Å)	90.23, 95.44, 106.25
α, β, γ (°)	90.000, 90.000, 90.000
Resolution (Å)	47.72-2.20 (2.27-2.20)**
R_{merge} (%)	8.2 (49.5)
$I / \sigma I$	15.8 (2.8)
Spherical completeness (%)	95.0 (87.9)
Redundancy	9.1 (6.2)
Refinement	
Resolution (Å)	47.72-2.20
No. reflections	22261 (2029)

$R_{\text{work}} / R_{\text{free}}$	19.85/27.30
No. atoms	
Protein	3299
Ligand/ion	na
Water	185
B-factors	
Protein	42.53
Ligand/ion	na
Water	43.92
R.m.s. deviations	
Bond lengths (Å)	0.01
Bond angles (°)	1.01

1 crystal was used to solve the HypocratesCS crystal structure

*Values in parentheses are for highest-resolution shell.

Comment 6 (R2): *For description on the crystal structure of HypocratesCS, the authors should put more efforts on discussion of the Hypocrates functioning. For example, how will the linker change affect the structure? Would fusion of cpYFP affect dimerization of NemR? Are there any other interactions on the surface between cpYFP and NemR?*

Response 6 (R2): How the linker would affect the structure is hard to predict without solving structures with different linkers. Why this specific combination of linkers (SAG/GT) results in the maximum response amplitude is experimentally determined and is hard to rationalize, even today, now that we have a view on the structure of HypocratesCS. One needs to realize that we are designing and creating a new protein and the quantity of flexibility needed to give a maximum read-out during the transfer of subtle conformational changes from the sensing domain to the beta-barrel is a complex process in which linker length and linker composition play an important role. The final ratiometric read-out is a change in the hydrogen bonding network surrounding the matured chromophore located in the center of the beta-barrel. For a deep rational structural view on this process in which also a water molecule is involved, we would need to have neutron protein crystallographic data, but this is beyond the scope of current study. With NPX data, we would also be able to optimize the ESPT pathway in favor of the response amplitude.

Reactive electrophiles showed the formation of a disulfide linked dimer as documented in DOI: [10.1111/mmi.12192](https://doi.org/10.1111/mmi.12192). To avoid disulfide linked dimerization, we decided

to create a sensing domain in Hypocrates with only one Cys left, being C106. This is now clearly mentioned in the Results section:

«For our work, a NemR mutant with all Cys residues substituted for Ser, except for Cys106 (NemR^{C106})⁴², was used to avoid undesirable sensitivity for reactive electrophilic species (RES) and to minimize other non-specific redox reactions, such as disulfide linked dimerization⁴⁵.»

Regarding the surface interaction between cpYFP beta-barrel and the NemR sensing domain in Hypocrates: The non-covalent interface interactions are relatively limited. Manual inspection showed possible hydrophobic π - π interaction between the rings of Tyr67 and Tyr106, and between the rings of Phe184 with Phe429. Further, we noticed H-bonding between the side chains of Gln91 and Gln165, and between Gln19 and Asn131 and between the side chain of Arg24 and the backbone oxygen of Glu133. The surface interactions are now included in the revised version of the manuscript.

Further, we hypothesize that the oxidation of Cys355, which is sitting on a flexible loop next to Gly354, couples to Gln99 and this one on its turn to Asn95 which is in contact with the 4-hydroxybenzyl group of the CR2 chromophore via a water molecule. The changes in this hydrogen bonding network induce changes of the hydrogen bonding network surrounding CR2 that results in a ratiometric response upon Cys oxidation by NaOCl. We included this idea in the discussion.

«By analogy with the C355A mutation, we expected that a K424A substitution would completely inactivate Hypocrates. However, the HypocratesKA variant was found to act very similar to the original sensor in terms of NaOCl sensitivity and response amplitude. This observation together with our mass spectrometrical data suggest that the NemR-sensor domain of Hypocrates does not function in a similar way to the *E. coli* transcription repressor NemR in forming a reversible sulfenamide bond^{42,44}, and that even the oxidation of Cys355 itself could be sufficient for signal changes in Hypocrates (Supplementary Fig. 14). We hypothesize that the oxidation of Cys355, which is located on a flexible loop next to Gly354, couples to Gln99 and this residue, in turn, couples to Asn95, which is in contact with the 4-hydroxybenzyl group of the chromophore (CR2) via a water molecule. Hence, oxidation of Cys355 mediated by (pseudo)hypohalous acids could induce changes in the hydrogen bonding network surrounding CR2 that lead to a ratiometric response. However, elucidating the precise sensing mechanism of Hypocrates is the subject of a separate study».

Moreover, we experimentally examined how the integration of a fluorescent protein into the NemR structure affected its oligomeric state. Using the gel filtration method, we

established that Hypocrates biosensor is a strict dimer regardless of its redox status and concentration in the solution.

We have added:

«As NemR was shown to form a dimer⁴², we decided to test whether the integration of the fluorescent protein into its structure affected the oligomeric state. Our gel filtration data indicate that Hypocrates is a strict dimer (99.85 kDa), and dimer formation is protein concentration and redox state independent (Supplementary Fig. 12).»

Supplementary Figure 12. Oligomeric state of Hypocrates. Gel filtration elution profiles of reduced Hypocrates at different concentrations and after treatment with NCT at 1:20 protein/oxidant ratio.

Comment 7 (R2): *It is important to verify whether Hypocrates is able to sense endogenous HClO, which is lacked in the current manuscript. The zebrafish tail fin injury model is not appropriate because there are too many signal changes incorporated during the wounding. In the introduction section and Fig. 5A,B, the author stated that the level of HClO is closely relate to heme enzyme myeloperoxidase (MPO). This supplied a simpler model to test performance of Hypocrates. The authors may change the MPO level with overexpression, inhibition, knockdown or knockout of MPO in cells and test the HClO concentration changes with Hypocrates.*

Response 7 (R2): We fully agree that our work lacked direct visualization of endogenous hypohalous stress. Experiments with cells containing high levels of MPO are

most preferred. Using human neutrophils and *E. coli* cells, we have shown the dynamics of hypohalous stress in phagocytosed bacteria.

We have added a new experiment and Fig. 5E:

«To demonstrate the possibility of monitoring endogenous hypohalous stress with Hypocrates, we used human peripheral blood neutrophils, which contain a high level of MPO. For this, *E. coli* cells expressing the Hypocrates probe were added to the medium of freshly isolated neutrophils. We monitored changes of the fluorescent signal in the bacterial cytoplasm every time a cell was phagocytosed by a leukocyte. HypocratesCS also showed an increase in the signal, but to a lesser extent, which indicates that inside the neutrophils, in addition to hypohalous stress, bacterial cells experience an increase in cytoplasmic pH (Fig. 5E).»

And a fragment for discussion:

«As a defense mechanism against pathogens, immune cells generate hypohalous acids. We tested the functioning of Hypocrates by visualizing hypohalous stress in the bacterial cytoplasm of *E. coli* captured and swallowed by neutrophils. To accomplish this, *E. coli* cells expressing Hypocrates were added to human peripheral blood neutrophils while the signal of the probe was monitored with a fluorescence microscope. When individual bacteria entered the neutrophils, we registered a sharp increase in the ratiometric response. In the same system, the signal of HypocratesCS also increased, but to a lesser extent. These observations correspond to the previously obtained results from the literature, showing that the human neutrophil phagosome adopts an alkaline pH for several tens of minutes after phagocytosis^{56,57}. Thus, the dynamics of the inactivated HypocratesCS version most likely reflect alkalization of bacterial cytoplasm. With the use of HypocratesCS as a control, it, therefore, becomes possible to assess the contribution of the pH component to the Hypocrates signal».

Figure 5. Hypocrates performance in vitro and in eukaryotic cell culture. (A) Fluorescence excitation spectra of purified Hypocrates (0.5 μ M) in the presence of individual MPO, H₂O₂ and MPO-H₂O₂ system. (B) Hypocrates (0.5 μ M) signal as a function of time in the presence of individual H₂O₂ and MPO-H₂O₂ system. HOCl, generated by MPO, leads to the development of a saturating response, while a physiologically irrelevant H₂O₂ concentration induces only minor signal changes. (C) The titration curve of Hypocrates in HeLa Kyoto cells exposed to different concentrations of NaOCl (values \pm SEM, N = 2 experiments, n \geq 25 cells per experiment). (D) Upper part: The timing of Hypocrates fluorescence changes induced by 40 μ M NaOCl (values \pm SEM, N = 2 experiments, n \geq 30 cells per experiment, 236,110 \pm 34,560 cells per dish). Lower part: Images of Hypocrates in transiently transfected HeLa Kyoto cells exposed to 40 μ M of NaOCl at different time points. Scale bar = 50 μ m. The lookup table indicates changes in the Ex₅₀₀/Ex₄₂₅ ratio. (E) Upper part: Images of human neutrophils phagocytosing *E. coli* cells that express Hypocrates or control version HypocratesCS. Scale bar = 10 μ m. The lookup table indicates changes in the Ex₅₀₀/Ex₄₂₅ ratio. Lower part: The timing of Hypocrates (blue line) and HypocratesCS (red line) fluorescence changes in *E. coli* cells phagocytosed by human neutrophils. The starting point on the graph corresponds to the moment at which individual bacteria are

phagocytosed by a neutrophil (values \pm SEM, N = 3 experiments, 35 bacterial cells in total for each version of the sensor).

Comment 8 (R2): *For the application of Hypocrates in the zebrafish tail fin injury model, the author should include the control version of HyPerRed (HyPerRed-C199S) as they did in ref. 27. Also, the authors should make more explanation and discussion on their results. It was shown that after tail amputation, the hydrogen peroxide level increased in 20 min and then decreased, while the hypochlorite level is more persistent. Why did the changes of these two oxidants' level differ from each other? The authors may compare the localization of signal increase between hydrogen peroxide and hypochlorite, to analyze whether they are produced in the same area of the injured tissue. The authors may also investigate the role of hypochlorite production in inflammation, for example, to examine whether the immune cells are recruited to the same area where hypochlorite is produced, and whether the cytokine level in this area is increased.*

Response 8 (R2): The profile of H₂O₂ production after tail amputation has been already published (Niethammer et al (2009) Nature 459, 996-999). This is a fairly common model of inflammation and has been used repeatedly with many genetically encoded biosensors. We only introduced the HyPerRed probe in this experiment to measure both H₂O₂ and (pseudo)hypohalous acids at the same time in the same embryos and to demonstrate with no ambiguity that these metabolites are accumulated at different rates. We believe that the mechanism of (pseudo)hypohalous acids production upon injury and their function in caudal fin regeneration are beyond the scope of this manuscript.

Comment 9 (R2): *In Supplementary Fig.1, the authors applied 2.1 mM NaOCl in measurement of HypR intrinsic fluorescence. Would ~2 mM NaOCl induce apparent protein damage on HypR? (also see major comments 3)*

Response 9 (R2): Many thanks for mentioning this as a discussion point. HypR and NemR are 2 different transcription factors coming from 2 different bacteria. HypR is a *S. aureus* transcription factor and NemR is a NaOCl responsive transcription factor from *E. coli*. Also mechanistically they are different: while NemR forms a sulfenamide (Gray et al. in JBC 2013 and ARS 2015), HypR involves a Cys33-Cys99 intersubunit disulfide formation upon NaOCl stress (Loi et al. ARS 2018). Evolutionary both transcription factors are different.

Further, whether a reaction occurs will not only depend on the reaction rate but also on the concentration of NaOCl and on the local protein concentrations in the cell. As far as we know, the intracellular concentrations of HypR and NemR are not known. From our data it became immediately clear that HypR does not react with NaOCl in the micromolar range, while for NemR we saw a clear response. In the case if mM NaOCl concentrations were fatal for HypR under pseudo first order reaction rate conditions, we would not be able to obtain a nice linear curve with increasing concentrations of NaOCl as can be observed from Supplementary Fig1.

Comment 10 (R2): *In Page 7, Line 28, the authors wrote “the probability of such a response in biological systems is low because of the low sensitivity to ONOO- and its low cellular concentration”. The authors should cite reference here, or note that “details in discussion”.*

Response 10 (R2): At the request of Reviewer 1, we revised this paragraph and removed this sentence, focusing on the recommendation to use NO generation controls.

Reviewer 3

Comment 1 (R3): *On line 18 page 4, the authors refer to "HypR" but this has not been defined.*

Response 1 (R3): We have made a clarification.

Comment 2 (R3): *The sensor detects both HClO and NCT. Why does this not create concern about selectivity/interference?*

Response 2 (R3): In biological systems, NCT can emerge via nucleophilic substitution reactions between taurine and HOCl or some compounds that contain N-Cl bonds (DOI: [10.1089/ars.2007.1927](https://doi.org/10.1089/ars.2007.1927); DOI: [10.2174/092986706778773095](https://doi.org/10.2174/092986706778773095)). There is also limited evidence published that MPO can chlorinate taurine directly (DOI: [10.1016/S0021-9258\(17\)37143-0](https://doi.org/10.1016/S0021-9258(17)37143-0)). Since HOCl is produced by MPO and N-Cl-bond containing molecules are generated in a similar manner as NCT, they all should be referred to as MPO-related oxidants that participate in hypochlorous stress progression. In other words, the total

concentration of these chemicals corresponds to the general activity of MPO. We believe that in most physiological research models it is much more reasonable to detect the total pool of reactive chlorine species rather than individual components.

If we compare a common cellular oxidant, H_2O_2 , with HOCl in the terms of their reactivity and stability, it will immediately become obvious that their temporal and spatial patterns of action are significantly different. The estimated lifetime of H_2O_2 in physiological conditions is ~100-fold higher than the corresponding value for HOCl (10 μs [DOI: 10.1038/nrm2240] vs 0.1 μs [DOI: 10.1038/nmicrobiol.2016.268]). Therefore, while it is known that H_2O_2 can diffuse over considerable distances, the radius of HOCl action is suggested to be less than 0.1 μm (1/10 of *E. coli* cell length). Thus, if the selectivity of a sensor was strictly limited to HOCl detection, such instrument would be able to only visualize very narrow regions of HOCl production and would tell nothing about real spread of oxidative stress.

The traditional view of hypochlorous stress claims that HOCl can hardly act as a direct toxic MPO-related agent since concentrations of nucleophilic N-containing molecules in live systems are very high: they can be found in free amino acids, in proteins, lipids, nucleic acids, enzymatic cofactors etc (DOI: 10.1016/s0076-6879(86)32043-3). As HOCl reacts with N-containing groups with quite fast rates, it is very likely that almost all HOCl is converted to secondary products at the regions of its emergence. This idea can be illustrated by a classical model when red blood cells (RBCs) are jointly incubated with activated neutrophils in order to measure the degree of oxidative stress experienced by cells in the presence of working MPO (DOI: 10.1016/S0021-9258(17)39793-4). It is experimentally shown that in such conditions no significant cell lysis takes place, while Hb is converted to oxidized met-form. This is markedly different from the situation when exogenous HOCl is added to RBCs – this protocol hardly affects Hb redox state, but leads to pronounced destruction of membranes. The ability to induce metHb formation is a known property of NH_2Cl and other hydrophobic halamines, and it is suggested that in the case of RBCs-neutrophils co-incubation the observed results are attributed to the emergence of NH_2Cl from HOCl and metabolically produced NH_4^+ . If any significant amounts of free HOCl leaked from neutrophils, then a detectible increase in the number of lysed cells would be observed.

It was shown that activated neutrophils secrete a relatively wide spectrum of N-Cl compounds and NCT is one of the most abundant agents (DOI: 10.1016/S0021-9258(18)90979-8). Its polar character due to the presence of a negatively charged group prevents it from easily crossing biological membranes and interacting with hydrophobic

interfaces of polypeptides. Therefore, it seems that NCT plays a dual role – on the one hand, it might lower the degree of the oxidative stress at some region (for example, addition of taurine to RBCs-neutrophils suspension rescues Hb from oxidation), but on the other hand, it might travel long distances and transfer its Cl^+ moiety to more reactive N-containing molecules there, which will expand the region of the hypochlorous damage.

A traditional and classic method for analytical visualization of hypochlorous stress is a combination of chromatography/mass spectrometry for the detection of chlorinated compounds among which 3-chlorotyrosines are ones of the most known. However, even though HOCl can undoubtedly react with Tyr residues directly, there is evidence published that chloramines play a key role in this process (DOI: [10.1074/jbc.270.28.16542](https://doi.org/10.1074/jbc.270.28.16542); DOI: [10.1074/jbc.M309046200](https://doi.org/10.1074/jbc.M309046200)). In this light, the ability of Hypocrates to sense both N-chloramines and HOCl is not different from the properties of other well-established and accepted techniques for the detection of hypochlorous stress.

At the request of the Reviewer 1, we have tested the sensitivity of Hypocrates to other (pseudo)hypohalous acids, HOBr, and HOSCN, and it turned out that the probe reacts with them in a similar manner as with HOCl. We, therefore, now position Hypocrates as a sensor for the total pool of (pseudo)hypohalous acids and their derivatives. Again, we do not think that this property of the developed instrument is a disadvantage.

First, in physiological conditions, heme peroxidases generate a mixture of (pseudo)hypohalous acids. Thus, the alkalization of the medium up to pH 7.8 (it can be observed in the lumen of the phagosome) results in enhancement of HOBr production by MPO, which might account for 40% of H_2O_2 consumed (DOI: [10.1016/j.abb.2005.07.005](https://doi.org/10.1016/j.abb.2005.07.005)). It is also known, that the concentration of SCN^- in the blood of active smokers can elevate from $\sim 30 \mu\text{M}$ to $\sim 250 \mu\text{M}$, and in such conditions MPO will generate HOCl and HOSCN in a similar proportions (DOI: [10.1042/bj3270487](https://doi.org/10.1042/bj3270487); DOI: [10.1016/j.freeradbiomed.2011.08.008](https://doi.org/10.1016/j.freeradbiomed.2011.08.008)).

Second, (pseudo)hypohalous acids can rapidly and non-enzymatically interact with (pseudo)halide ions, and this process results in the conversion of these species to each other (DOI: [10.1016/j.watres.2007.07.025](https://doi.org/10.1016/j.watres.2007.07.025)). Thus, kinetic modeling studies predict that in the blood plasma almost quantitative conversion of HOBr to HOSCN takes place (DOI: [10.1021/tx050338c](https://doi.org/10.1021/tx050338c)). Given that HOSCN is a much more stable and much more selective agent (like N-chloramines), a similar line of reasoning, why it might be good for a sensor to detect this compound, can be drawn.

Comment 3 (R3): *The authors criticize other (small molecule) probes for slow kinetics, for taking "dozens of seconds and even minutes" to respond to HOCl. However, in the experiments herein, the authors describe a similar response time by their sensor (e.g., to DTT treatment to show reversibility). Despite the authors' claims, there are examples of published small abiotic probes for HOCl that are rapid (a few seconds or less) as well as examples that are reversible and ratiometric. It is not fair to the researchers who created such probes to dismiss all of them together as having big drawbacks.*

Response 3 (R3): We have removed this paragraph.

Comment 4 (R3): *Another criticism has to do with comments about other probes not being useful for reversibility over several cycles; however, have the authors proven the validity of this property in their sensor.*

Response 4 (R3): We conducted an experiment confirming that the biosensor can be used in repetitive oxidation/reduction cycles.

We have added Supplementary Figure 13A:

Supplementary Figure 13. Hypocrates in eukaryotic cell culture. (a) Hypocrates Ex₅₀₀/Ex₄₂₅ ratio changes after several serial additions of 25 μM NaOCl (values ± SEM, N = 2 experiments, n = 26 cells). The response of the sensor is reversible; therefore, Hypocrates is capable of registering multiple oxidation/reduction events. **(b)** Hypocrates, HypocratesCS and SypHer3s Ex₅₀₀/Ex₄₂₅ ratio changes after addition of 40 μM NaOCl (values ± SEM, N = 3 experiments, n ≥ 28 cells per experiment).

Comment 5 (R3): *The probe is pH-sensitive. Why is this not an issue since pH is not the same in all regions of the cell, under different physiological states, etc.?*

Response 5 (R3): The pH sensitivity of Hypocrates is determined by the pH sensitivity of the cpYFP chromophore. This is a well-known fact and a given that we have to work with. Most of today's widely used biosensors exhibit similar pH sensitivity, but this does not make them less popular: Frex (DOI: [10.1016/j.cmet.2011.09.004](https://doi.org/10.1016/j.cmet.2011.09.004)), HyPer-1 (DOI: [10.1038/nmeth866](https://doi.org/10.1038/nmeth866)), HyPer-2 (DOI: [10.1016/j.bmc.2010.07.014](https://doi.org/10.1016/j.bmc.2010.07.014)), HyPer-3 (DOI: [10.1021/cb300625g](https://doi.org/10.1021/cb300625g)), RexYFP (DOI: [10.1016/j.bbagen.2013.11.018](https://doi.org/10.1016/j.bbagen.2013.11.018)), MetSOx (DOI: [10.1038/nchembio.1787](https://doi.org/10.1038/nchembio.1787)), OHSer (DOI: [10.1021/ja1071114](https://doi.org/10.1021/ja1071114)), GEVALs (DOI: [10.1038/nmeth.4404](https://doi.org/10.1038/nmeth.4404)), FHisJ (DOI: [10.1038/srep43479](https://doi.org/10.1038/srep43479)), Perceval (DOI: [10.1038/nmeth.1288](https://doi.org/10.1038/nmeth.1288)) among others. The key to the success of such biosensor depends on the selection or creation of a reliable pH control. It is necessary that the dependence of the spectral properties on the pH value for the biosensor and its control version ideally coincides. For example, for the well-known biosensors of the HyPer family probes, SypHer versions are an ideal pH control. HyPer and SypHer differ in a single mutation at a key amino acid residue. We used the same approach and created HypocratesCS version. This inactivated version does not show a response when interacting with the target analyte, but it has the same spectral properties and pH sensitivity. In this work, we strongly recommend using HypocratesCS for monitoring pH changes. This approach is generally accepted for most genetically encoded biosensors. At the request of the Reviewer 2, we made an additional experiment in which we titrated biosensor by oxidant NCT at different pH values. We believe this will also help to avoid measurement artifacts associated with fluctuations in pH.

Supplementary Figure 7. (a-d) Hypocrates fluorescence excitation spectra in the presence of different NCT concentrations at (a) pH 7.0; (b) pH 7.3; (c) pH 7.6; (d) pH 7.9 in 100 mM sodium phosphate buffers. (e) The same data represented as titration curves. Protein concentration was 0.5 μM in all cases.

Reviewers' Comments:

Reviewer #1:

Remarks to the Author:

The authors have done a good job of addressing most of the reviewers' comments and revising their manuscript to include reactivity of the probe with other reactive halogen species. However, I still have a major and a minor comment.

Major Comment

Re original comment 1. I still have concerns about the rate constants measure for Hypocrates, as they are so out of line with the reactivity of other thiols and are also hard to reconcile with NemR being an efficient sensor for HOCl. While the rationale based on accessibility proposed by the authors for why Hypocrates reacts ~1000-fold slower than GSH could have some credence, their rate constant for NCT (a fully ionised sulfonic acid) is ~100-fold faster for Hypocrates than GSH. This does not fit with an accessibility explanation. Also, GAPDH and creatine kinase are as reactive as GSH, if not more so, with HOCl (ref 21). Thus the probe (and sensor) would compete extremely poorly with GSH and other protein thiols, N-terminal amines, disulfides (see Pattison and Davies) that would be present in cells in vast excess. Yet it appears to be oxidised if sufficient HOCl is added to cells and in phagocytosed bacteria (assuming the response is not all a pH effect). Does this mean all there is sufficient excess to oxidise all these other constituents (ie sensitivity is not great) or the probe-HOC reaction is actually faster. I would like to see the stopped flow data validated by another method. This could be done by competitive kinetics. For example, with NCT, the rate constant measured here implies that methionine or ascorbate (rate constants 39 and 13 M⁻¹s⁻¹) should not inhibit even in considerable excess. Ascorbate and perhaps N-acetylcystine, and N-acetylTrp could be employed with HOCl and you should even see inhibition with N-acetylTyr.

There is also an unexpected y intercept in the kinetic data in Fig 3F&G.

If this cannot be resolved, more caution should be applied to interpreting the measured rate constants.

Minor comment

Re original comment 5. Lines 384-385 and legend to Fig 5. Rather than the statement about cell number in the text (which would need also to mention volume), it would be more useful to readers if the legend for d and e stated the number of nmol HOCl per 10⁵ cells. Is the accuracy of the cell number perhaps somewhat excessive? Clarify if SEM refers to within or between experiments. Range not SEM should be used when n=2.

Reviewer #2:

Remarks to the Author:

In the revised version of the manuscript "Hypocrates, a genetically encoded fluorescent biosensor for (pseudo)hypohalous acids and their derivatives", the authors have provided new data and made extensive textual edits to support their claims. Detailed responses have been made to most of the comments of the 3 reviewers. The sensing mechanism, kinetics, specificity, and the application study for Hypocrates are all strengthened with the new experiment results. This reviewer only has several additional suggestions that will improve the readability of the main text.

1. The Figure 1b might be modified to understate formation of sulfenamide. This reviewer understand that this subfigure is just a proposed scheme and reasonable during sensor design. Yet it would be better to tone down the description on formation of sulfenamide as the new experiment results indicated that Lys424 was not involved in hypohalous acids sensing of Hypocrates.

2. Page 12, line 3, the description of crystal cell dimensions should be the same with that in the Supplementary table 3.

3. Page 18, line 481, "By analogy with the C355A mutation", should be C355S here?

Reviewer #3:

Remarks to the Author:

The paper has been revised appropriately.

#3 Response to Reviewers

Dear Reviewers,

We would like to thank you again for the careful review of our paper. Here we answer to all the questions.

Reviewer 1

Comment 1 (R1): Re original comment 1. I still have concerns about the rate constants measure for Hypocrates, as they are so out of line with the reactivity of other thiols and are also hard to reconcile with NemR being an efficient sensor for HOCl. While the rationale based on accessibility proposed by the authors for why Hypocrates reacts ~1000-fold slower than GSH could have some credence, their rate constant for NCT (a fully ionised sulfonic acid) is ~100-fold faster for Hypocrates than GSH. This does not fit with an accessibility explanation. Also, GAPDH and creatine kinase are as reactive as GSH, if not more so, with HOCl (ref 21). Thus the probe (and sensor) would compete extremely poorly with GSH and other protein thiols, N-terminal amines, disulfides (see Pattison and Davies) that would be present in cells in vast excess. Yet it appears to be oxidised if sufficient HOCl is added to cells and in phagocytosed bacteria (assuming the response is not all a pH effect). Does this mean all there is sufficient excess to oxidise all these other constituents (ie sensitivity is not great) or the probe-HOC reaction is actually faster. I would like to see the stopped flow data validated by another method. This could be done by competitive kinetics. For example, with NCT, the rate constant measured here implies that methionine or ascorbate (rate constants 39 and 13 M⁻¹s⁻¹ should not inhibit even in considerable excess. Ascorbate and perhaps N-acetylcystine, and N-acetylTrp could be employed with HOCl and you should even see inhibition with N-acetylTyr.

There is also an unexpected y intercept in the kinetic data in Fig 3F&G.

If this cannot be resolved, more caution should be applied to interpreting the measured rate constants.

Response 1 (R1): Regarding possible pH effect: we made control experiments with SypHer3s and HypocratesCS, so pH effect is not an issue.

Generally, we do understand the concern of the Reviewer. Therefore, we went back to the raw data, and we checked the residual plot diagnostics for all the data (see curves as an example for one progress curve).

It turned out that we obtain a more reliable fitting of the progress curves using double association exponential curves for Hypocrates, NemRC106 and HypR. With the new obtained k_{fast} values, the respective second order rate constants were determined, which resulted in new Fig. 3 e,f,g panels:

(e-g) Hypocrates reaction rates. Changes in cpYFP fluorescence at $>515 \text{ nm}$ cut-off ($\lambda_{\text{ex}} = 485 \text{ nm}$) were measured as a function of time (insert). The curves were fitted to a double exponential to obtain the observed rate constants ($k_{\text{obs/fast}}$), which were plotted as a function of different **(e)** NaOCl, **(f)** NCT or **(g)** NaOBr concentrations. The second-order rate constants for NaOCl (1.4 ± 0.056) $\times 10^6 \text{ M}^{-1} \text{ s}^{-1}$, NCT (6.1 ± 0.3) $\times 10^4 \text{ M}^{-1} \text{ s}^{-1}$ and NaOBr (4.5 ± 0.25) $\times 10^6 \text{ M}^{-1} \text{ s}^{-1}$ were determined from the slope of the straight line. The data are presented as a mean \pm SD, $n \geq 2$.

Supplemental Figure S1 was adapted:

Supplementary Figure 1. NemR^{C106} senses NaOCl 200-fold faster than HypR. The intrinsic Tyr (a) and Trp (b) fluorescence changes of HypR and NemR^{C106}, respectively, at increasing NaOCl concentration are shown as functions of time (inserts). Note, as HypR has no Trp in its sequence, we followed the Tyr fluorescence change. The curves were fitted to a double exponential to obtain the observed rate constants ($k_{\text{obs/fast}}$), which were plotted as functions of increasing NaOCl concentration. From the slopes, the second-order rate constants were obtained. The second-order rate constant of NemR^{C106} is $(2.96 \pm 0.21) \times 10^5 \text{ M}^{-1}\text{s}^{-1}$ and the one of HypR is $(1.48 \pm 0.47) \times 10^3 \text{ M}^{-1}\text{s}^{-1}$. The data are presented as a mean \pm SD, $n \geq 1$.

The main text was also adapted accordingly: “We found that NemR^{C106} ($\sim 2.96 \times 10^5 \text{ M}^{-1}\text{s}^{-1}$) reacts 200-times faster compared to HypR ($\sim 1.48 \times 10^3 \text{ M}^{-1}\text{s}^{-1}$), while exposure to H₂O₂ had no effect in both cases (Supplementary Fig. 2).”

Supplemental Figure S9 was adapted:

Supplementary Figure 9. The kinetics of NemR^{C106} for N-chlorotaurine. Changes in intrinsic Trp fluorescence were measured as a function of time (insert). The curves were fitted to a double exponential and k_{fast} values were used as the observed rate constants (k_{obs}), which were plotted as a function of increasing NCT concentration (20-100 μM). From the slope the second-order rate constant of $(4.39 \pm 0.24) \times 10^3 \text{ M}^{-1}\text{s}^{-1}$ was determined. The data are presented as a mean \pm SD, $n \geq 2$.

Also, the main text was adapted and now reads:

“To compare the reaction rates of Hypocrates towards NaOBr, NaOCl, and NCT, the second-order rate constants were measured on a stopped-flow instrument (Fig. 3e-g). We found that the biosensor reacts ~100-fold faster with NaOBr ($\sim 4.5 \times 10^6 \text{ M}^{-1}\text{s}^{-1}$) and NaOCl ($\sim 1.4 \times 10^6 \text{ M}^{-1}\text{s}^{-1}$) compared to NCT ($\sim 6.1 \times 10^4 \text{ M}^{-1}\text{s}^{-1}$). NemR^{C106} is also less reactive to NCT compared to NaOCl (Supplementary Fig. 9), which possibly corresponds to the fact that NCT is a less aggressive compound. A previous study²¹ showed that NCT reacts with glyceraldehyde-3-phosphate dehydrogenase at $300 \text{ M}^{-1}\text{s}^{-1}$, and with creatine kinase at $1.2 \times 10^2 \text{ M}^{-1}\text{s}^{-1}$, making Hypocrates 100 times more efficient in recognizing HOCl-modified taurine.”

Further, the material and method section has been adapted. It now reads:

«Changes in fluorescence were monitored, and the obtained curves were fitted with double exponential equations which gave trustworthy diagnostic residual plots. From the double exponential equations, the observed k_{fast} values were used to determine the second order rate constants.»

Finally, Abstract has been adapted, too. It now reads:

“We show that Hypocrates is ratiometric, reversible, and responds to its analytes in the $10^6 \text{ M}^{-1}\text{s}^{-1}$ range.”

As a result of this double exponential fitting, the k -values for NaOCl and NaOBr increased with one log, while the k -value for NCT stayed approximately the same. A question remains: can we really trust these new second order rate constants? To find out, we performed additional competitive inhibition experiments, which also provide answers for some of the remarks from the Reviewer.

To further confirm the stopped flow secondary rate constants of Hypocrates for NCT and NaOCl, we decided to perform competition experiments similar to the one described in the Storkey et al. 2014 paper (<http://dx.doi.org/10.1016/j.freeradbiomed.2014.04.024>). The general idea is here to determine the k-values of the biosensor based on the slope using the reworked competition equation from the Storkey et al. 2014 paper (equation 2), assuming that the k-values for NAC, GSH and other possible competitors are correctly reported in the literature. If this is the case, we should find values similar to those determined in the stopped-flow experiments.

We used the following experimental set-up: 0.5 μM biosensor and 5 μM NaOCl or NCT (V_{max} conditions for the biosensor to obtain a reliable readout). We varied the competitive compounds (GSH, NAC, Ala) between 10 μM and 80 μM in an argon flushed buffer solution of 100 mM NaPO_4 , pH 7.4. The competitive compounds were prepared in the same buffer and pH adjusted.

The following k-values were used (Pattison et al. 2001, Storkey et al. 2014):

- For HOCl

GSH: $1.2 \times 10^8 \text{ M}^{-1}\text{s}^{-1}$; NAC: $2.9 \times 10^7 \text{ M}^{-1}\text{s}^{-1}$; Ala: $5.4 \times 10^4 \text{ M}^{-1}\text{s}^{-1}$.

- For NCT

GSH: $1.15 \times 10^2 \text{ M}^{-1}\text{s}^{-1}$; NAC: $4.6 \times 10 \text{ M}^{-1}\text{s}^{-1}$.

We observed the following:

1. GSH with HOCl: already at the lowest GSH concentration (10 μM), the “inhibition degree” jumps to its maximum. This is what was expected as Hypocrates reacts at $\sim 10^6 \text{ M}^{-1}\text{s}^{-1}$ with NaOCl; GSH reacts 100x faster than the biosensor, hence it immediately outcompetes it.

2. After fitting with equation 2 from the Storkey et al. paper, NCT provided relatively good linear response plots for GSH and NAC, which resulted in $k_{\text{biosensor}}$ values of $\sim 10^4 \text{ M}^{-1}\text{s}^{-1}$. These k -values are similar to the one determined by the stopped-flow experiments (6.1 ± 0.3) $\times 10^4 \text{ M}^{-1}\text{s}^{-1}$.

Slope: $k_{\text{GSH}}/k_{\text{biosensor}} = 0.003565$
 $1.15 \times 10^2 \text{ M}^{-1}\text{s}^{-1} / k_{\text{biosensor}} = 0.003565$
 $k_{\text{biosensor}} = 3.22 \times 10^4 \text{ M}^{-1}\text{s}^{-1}$

Slope: $k_{\text{NAC}}/k_{\text{biosensor}}$
 For NCT:
 $4.6 \times 10 \text{ M}^{-1}\text{s}^{-1} / k_{\text{biosensor}} = 0.003072$
 $k_{\text{biosensor}} = 4.34 \times 10^4 \text{ M}^{-1}\text{s}^{-1}$

With this experiment, we confirmed the stopped flow determined reaction rate of Hypocrites with NCT.

3. For confirming the k -value of the biosensor for HOCl, we used Ala.

Slope: $k_{\text{Ala}}/k_{\text{biosensor}}$
 For NaOCl:
 $5.4 \times 10^4 \text{ M}^{-1}\text{s}^{-1} / k_{\text{biosensor}} = 0.0006883$
 $k_{\text{biosensor}} = 7.8 \times 10^7 \text{ M}^{-1}\text{s}^{-1}$

For Ala, we obtained a slightly higher value than the one determined with stopped-flow (1.4 ± 0.056) $\times 10^6 \text{ M}^{-1}\text{s}^{-1}$. Considering that with this method we depend on the correctness of the published k_{Ala} value for NaOCl, we are confident that the second order rate constants determined by stopped flow using a double exponential fit are correct.

To further confirm the kinetic behavior of Hypocrates in the presence of NaOCl, we performed competition with ascorbate, NAC, and GSH (the N-acetylTrp and N-acetylTyr – like suggested by the Reviewer - were not directly available). We generated qualitative data by measuring the reactions under steady-state V_{max} kinetic conditions of NaOCl (being determined to be $10 \mu\text{M}$). The set-up is here to measure the effect of these compounds on the read-out of Hypocrates.

1. We checked whether these potential competitors influenced the read-out of the biosensor ($0.5 \mu\text{M}$) in the absence of NaOCl. No effect was observed. This also means that these compounds under these buffer solution conditions do not affect pH.

2. We checked kinetic competition for $10 \mu\text{M}$ NaOCl at high concentrations of NAC, Asc (1 mM, 2000 higher than the biosensor concentration – $0.5 \mu\text{M}$), and GSH (10 mM, 20000 higher).

In the presence of 10 mM GSH and 1 mM NAC, Hypocrates does not detect NaOCl anymore. This is not a surprise knowing the k -values of GSH and NAC for NaOCl. For Asc, we still see a ratiometric response of Hypocrates. These experiments were started by adding NaOCl to the well in which biosensor (0.5 μ M) and compound were already present.

3. Under conditions with equimolar concentrations of biosensor and competitor (0.5 μ M), the biosensor is still detecting NaOCl.

In general, we can conclude that Hypocrates' read-out strongly depends on the local concentrations of potential thiol competitors, and if the local concentrations of these competitors are high, Hypocrates will be outcompeted. If the local concentration of competitor is low, Hypocrates will visualize hypohalous stress.

Therefore, we decided to add the following in the discussion:

“It might seem, that having the reaction rate of $\sim 10^6 \text{ M}^{-1}\text{s}^{-1}$ for NaOCl, Hypocrates will not be able to visualize hypohalous stress efficiently, since it will be outcompeted by other sulfur-containing compounds, as some of them are present at millimolar concentrations and operate at 10^7 - $10^8 \text{ M}^{-1}\text{s}^{-1}$. However, being an extremely reactive compound, HOCl hardly induces its biological effects directly. Apparently, HOCl is locally converted into less aggressive halamines, like NCT, and these compounds, in turn, might affect the cellular metabolism. Therefore, the ability of Hypocrates to sense NCT with a relatively high reaction rate makes it an efficient tool for the visualization of hypohalous stress.”

Comment 2 (R1): Re original comment 5. Lines 384-385 and legend to Fig 5. Rather than the statement about cell number in the text (which would need also to mention volume), it would be more useful to readers if the legend for d and e stated the number of nmol HOCl per 10^5 cells. Is the accuracy of the cell number perhaps somewhat excessive? Clarify if SEM refers to within or between experiments. Range not SEM should be used when $n=2$.

Response 2 (R1): In line with your reasoning, we have changed panels c and d in Fig. 5 as well as their description:

The new description is: “(d) Upper part: Hypocrates fluorescence changes induced by 17 nmol NaOCl/(10⁵ cells) (values \pm SEM, N = 2 experiments, n \geq 30 cells per experiment). Lower part: Images of Hypocrates in transiently transfected HeLa Kyoto cells exposed to 17 nmol/(10⁵ cells) NaOCl at different time points. Scale bar = 50 μ m. The lookup table indicates changes in the Ex₅₀₀/Ex₄₂₅ ratio.”

We also adapted the main text: “The minimal NaOCl addition that induced detectable changes of the sensor fluorescence was approximately 4.2 nmol/(10⁵ cells) (Fig. 5c). Exposure to 17 nmol/(10⁵ cells) NaOCl led to a signal change of 1.8-fold, which is similar to the saturating response obtained with purified protein and in *E. coli* suspension (Fig. 1e,f).” and “Exposure to 17 nmol/(10⁵ cells) NaOCl did not significantly affect the signal of both

probes (Supplementary Fig. 13b), indicating that the behavior of Hypocrates, observed in this system, specifically reflects a NaOCl-induced response.”

We have made the same corrections to the Supplementary Figure 13 and its description:

The new description is: “Supplementary Figure 13. Hypocrates in eukaryotic cell culture. (a) Hypocrates Ex₅₀₀/Ex₄₂₅ ratio changes after several serial additions of 10.5 nmol/(10⁵ cells) NaOCl (values ± SEM, N = 2 experiments, n = 26 cells). The response of the sensor is reversible; therefore, Hypocrates is capable of registering multiple oxidation/reduction events. (b) Hypocrates, HypocratesCS and SypHer3s Ex₅₀₀/Ex₄₂₅ ratio changes after addition of 17 nmol/(10⁵ cells) NaOCl (values ± SEM, N = 3 experiments, n ≥ 28 cells per experiment).”

The Methods section was also corrected: “The desired amount of NaOCl stock was diluted in 100 μl of PBS just before addition, and the final concentration of NaOCl in the sample was in the range of 10-40 μM (~4.2-17 nmol/(10⁵ cells)).”

Since we have recalculated molar concentrations to nmol/cells ratios, the mentioning of the average cells number per dish as well as SEM were removed from the manuscript.

Reviewer 2

Comment 1 (R2): The Figure 1b might be modified to understate formation of sulfenamide. This reviewer understand that this subfigure is just a proposed scheme and reasonable during sensor design. Yet it would be better to tone down the description on formation of sulfenamide as the new experiment results indicated that Lys424 was not involved in hypochlorous acids sensing of Hypocrates.

Response 1 (R2): We fully agree with this comment. We have simplified this scheme.

Comment 2 (R2): Page 12, line 3, the description of crystal cell dimensions should be the same with that in the Supplementary table 3.

Response 2 (R2): We have corrected this misprint. The new text is: “Only HypocratesCS gave diffraction-quality crystals. The orthorhombic crystals ($C222_1$, $a=90.23$, $b=95.44$, $c=106.25$, $\alpha=\beta=\gamma=90^\circ$) contain one molecule of the biosensor per asymmetric unit and diffract to a resolution of 2.2 Å (Supplementary Table 3).”

Comment 3 (R2): Page 18, line 481, "By analogy with the C355A mutation", should be C355S here?

Response 3 (R2): We have corrected this misprint. The new text is: "By analogy with the C355S mutation, we expected that a K424A substitution would completely inactivate Hypocrates."

We have also corrected some other minor typos in the text.

Reviewers' Comments:

Reviewer #1:

Remarks to the Author:

I appreciate the efforts made by the authors in expanding and consolidating their rate data. Indeed the evidence does favour a relatively slow reaction of HOCl with Hypocrates, and a relatively fast reaction with TauCL. On that basis I agree with their suggestion that in biological systems it is probably more likely that Hypocrates picks up secondary chloramines rather than HOCl itself.

Regarding their additional competition experiments, I think that the Figure shown as point 2, page 7 of the response is useful information and recommend that it be included in the supplementary material.

Regarding the other competition experiments, I would note that the alanine experiment (point 3 page 7) is confounded, because the product, alanine chloramine, should react with the probe. This may explain the high rate constant. I would also query the value of the experiment shown as point 3 page 9, as the high excess of HOCl over GSH means that scavenging would have a negligible effect on the amount of HOCl available to react with the probe. This experiment would need to be done with excess target concentrations over oxidant. (I am not proposing this be done.)

However, there is one point that I raised that the authors have not addressed and still needs attention:

There is also an unexpected y intercept in the kinetic data in Fig 3E&G.

With HOCl and HOBr there is a large negative intercept on the y-axis (~ 6 and 12 s^{-1}). This may not be readily apparent as the zero points of the x-axes are not shown. Presumably with zero oxidant, there was minimal fluorescence change, in which case the slope would become non-linear at lower HOCl or HOBr concentrations. This is interesting and important and I would like to see these data shown and commented upon. Does it imply that lower concentrations are consumed by the protein solution? Could it be that there are other reactive sites on the protein (eg Met) that react first so that there is a threshold amount of oxidant that must be overcome before it affects fluorescence? Or might small amounts of these reactive oxidants be consumed non-specifically independent of the protein? Please address this issue.

Reviewer #2:

Remarks to the Author:

The issues raised by this reviewer have been all addressed in the revised paper. In addition, for the competition experiments in the rebuttal letter, the authors should notice that the "equation 2" from the Storkey et al. 2014 paper is based on the approximations that 1) the concentrations of both sensor and scavenger are treated as constant throughout the reaction; 2) the effect of reverse reactions is negligible. Nevertheless, this may not contradict with the conclusions made in the rebuttal letter since the difference of the calculated k value may not exceed one log, and the results of the competition experiments are not reported in the manuscript. Therefore, this reviewer considers that the current manuscript is appropriately revised.

№4 Response to the Reviewers

Dear Reviewers,

We would like to thank you again for the careful review of our paper. Here we answer to all the questions.

Reviewer 1

Comment: Regarding their additional competition experiments, I think that the Figure shown as point 2, page 7 of the response is useful information and recommend that it be included in the supplementary material.

Response: We respectfully tend to disagree to add this information to the manuscript, because, as correctly noted by Reviewer 2, the competitive kinetics experiments that we presented in the previous revision of the manuscript (0.5 μM biosensor with 5 μM NaOCl (i.e. V_{max} conditions)), do not fulfil the conditions required for the utilization of "equation 2" from the Storkey et al. 2014 paper, which are that the concentrations of both sensor and scavenger are treated as constant throughout the reaction (i.e. are in excess compared to the concentration of NaOCl). This is because treating 5 μM sensor with 0.5 μM NaOCl gives a very small read-out window that is within experimental error.

Nevertheless, to make sure that the competitive kinetics results we presented previously were statistically significant, we performed additional competitive kinetic experiments with the same settings (0.5 μM biosensor with 5 μM NCT or NaOCl). In addition, we took into account the comment of the reviewer on the unsuitability of the usage of alanine as a competitor for Hypocrates in experiments with HOCl, and instead used GSH, NAC and dithiodipropionic acid (DTP).

As a reminder, the goal of these experiments was to confirm the k -values, obtained for Hypocrates by stopped-flow kinetics, which follows the cpYFP intensimetric response.

The rate constants for the reactions of the competitors with NCT and NaOCl have been determined previously (Pattison and Davies, 2001, doi:10.1021/tx0155451; Pattison and Davies, 2006, doi:10.2174/092986706778773095; Storkey et al. 2014).

Secondary k -values:

GSH for NaOCl: $1.2 \times 10^8 \text{ M}^{-1}\text{s}^{-1}$; and for NCT: $1.15 \times 10^2 \text{ M}^{-1}\text{s}^{-1}$

NAC for NaOCl: $2.9 \times 10^7 \text{ M}^{-1}\text{s}^{-1}$; and for NCT: $4.6 \times 10 \text{ M}^{-1}\text{s}^{-1}$

DTP for NaOCl: $1.6 \times 10^5 \text{ M}^{-1}\text{s}^{-1}$

Slope: $k_{\text{NAC}}/k_{\text{biosensor}}$
For NCT:
 $4.6 \times 10 \text{ M}^{-1}\text{s}^{-1}/k_{\text{biosensor}} = 0.002912$
 $k_{\text{biosensor}} = 1.58 \times 10^4 \text{ M}^{-1}\text{s}^{-1}$

Slope: $k_{\text{GSH}}/k_{\text{biosensor}} = 0.002321$
 $1.15 \times 10^2 \text{ M}^{-1}\text{s}^{-1}/k_{\text{biosensor}} = 0.002321$
 $k_{\text{biosensor}} = 4.95 \times 10^4 \text{ M}^{-1}\text{s}^{-1}$

Slope: $k_{\text{GSH}}/k_{\text{biosensor}} = 0.001211$
 $2.9 \times 10^7 \text{ M}^{-1}\text{s}^{-1} / k_{\text{biosensor}} = 0.001211$
 $k_{\text{biosensor}} = 2.39 \times 10^{10} \text{ M}^{-1}\text{s}^{-1}$

Slope: $k_{\text{GSH}}/k_{\text{biosensor}} = 0.006983$
 $1.2 \times 10^8 \text{ M}^{-1}\text{s}^{-1} / k_{\text{biosensor}} = 0.006983$
 $k_{\text{biosensor}} = 1.72 \times 10^{10} \text{ M}^{-1}\text{s}^{-1}$

Slope: $k_{\text{GSH}}/k_{\text{biosensor}} = 1.84 \times 10^{-5}$
 $1.6 \times 10^5 \text{ M}^{-1}\text{s}^{-1} / k_{\text{biosensor}} = 1.84 \times 10^{-5}$
 $k_{\text{biosensor}} = 8.69 \times 10^9 \text{ M}^{-1}\text{s}^{-1}$

For NCT, we could confirm a rate constant in the range of $\sim 10^4 \text{ M}^{-1}\text{s}^{-1}$. However, for NaOCl we obtained with all three competitors a diffusion-controlled secondary rate constant of $\sim 10^{10} \text{ M}^{-1}\text{s}^{-1}$, which is extremely fast. The results obtained are reproducible (triplicates), confirmed with all three competitors, and the controls without NaOCl gave no biosensor response. Still, we must be careful (see also remark of Reviewer 2 - both competitor and biosensor concentrations should not change to allow the equation 2 from the Storkey et al. paper be applied), and we must consider that we cannot guarantee that the concentrations of biosensor and competitor stay constant during the assay.

Despite this, if we discuss the data obtained qualitatively, we can see that for a compound that has a rate constant that is just 10x lower than the one we obtained for Hypocrates by stopped-flow (DTP), 1000x the amount is required to show some degree of competition. At the same time, Hypocrates can compete with GSH and NAC, which are supposed to react 100x and 10x faster, respectively, even while being at a concentration that is at least 10x lower (0.5 μM vs 10-80 μM). In the case of NCT, for example, GSH has a 100x lower rate constant than Hypocrates, but can compete when used in the 10-80 μM concentration range. Hence, DTP, which is supposed to be 10x slower than Hypocrates in reacting with HOCl, would also be expected to exhibit competition with Hypocrates when used in the 10-80 μM range. Therefore, the reaction of Hypocrates with NaOCl is likely even higher than $10^6 \text{ M}^{-1}\text{s}^{-1}$, but we cannot obtain a more accurate value due to technical limitations (more precisely, dead times), of the stopped-flow instrument. To accomplish this, we would need to perform competitive stopped-flow experiments, which is already beyond the scope of the present manuscript, whose aim is to report on the first biosensor for (pseudo)hypohalous acids and demonstrate that it works in biological contexts (like cell cultures or model organisms).

Another possible explanation to those discrepancies could be that, as NaOCl is very reactive it can modify several residues and our mass spec data also show this (the raw mass spectrometry data are available via ProteomeXchange with identifier PXD029624). The analysis showed bromination and chlorination on several residues. Special attention should go to the chlorination of Y106, located close to C355.

Mass spectrometric results highlighting the residues that were found chlorinated/brominated or oxidized in Hypocrates. The oxidized methionines are not indicated in this figure (see below)

Chlorination of Y106 might also result in a biosensor response. It could well be that the rate of $10^6 \text{ M}^{-1}\text{s}^{-1}$ is not reflecting the rate of C355 oxidation, but Y106 chlorination (our mass spec data showed Y106 chlorination with NCT as well as with NaOCl).

Comment: Regarding the other competition experiments, I would note that the alanine experiment (point 3 page 7) is confounded, because the product, alanine chloramine, should react with the probe. This may explain the high rate constant.

Response: We agree with this point. Therefore, we decided to repeat the experiments with GSH, NAC, and DTP (see above).

Comment: I would also query the value of the experiment shown as point 3 page 9, as the high excess of HOCl over GSH means that scavenging would have a negligible effect on the amount of HOCl available to react with the probe. This experiment would need to be done with excess target concentrations over oxidant. (I am not proposing this be done.)

Response: The general idea of this experiment was to show that under competition conditions (same concentrations of biosensor and competitor) with a fixed high concentration of NaOCl, the biosensor is still picking up the signal. In any case, this experiment is not included to the manuscript.

Comment: However, there is one point that I raised that the authors have not addressed and still needs attention:

There is also an unexpected y intercept in the kinetic data in Fig 3E&G. With HOCl and HOBr there is a large negative intercept on the y-axis (~ -6 and 12 s^{-1}). This may not be readily apparent as the zero points of the x-axes are not shown. Presumably with zero oxidant, there was minimal fluorescence change, in which case the slope would become non-linear at lower HOCl or HOBr concentrations. This is interesting and important and I would like to see these data shown and commented upon. Does it imply that lower concentrations are consumed by the protein solution? Could it be that there are other reactive sites on the protein (eg Met) that react first so that there is a threshold amount of oxidant that must be overcome before it affects fluorescence? Or might small

amounts of these reactive oxidants be consumed non-specifically independent of the protein? Please address this issue.

Response: We agree that this interesting phenomenon needs explanation. It might well be that some portions of NaOCl and NaOBr were already consumed by residues of Hypocrates located far enough from the chromophore for their chlorination/oxidation not to affect fluorescence. Indeed, as mentioned above, we observe the chlorination/bromination of different regions in our mass spec data. Further, the methionine oxidation in our mass spec experiment is for sure present, but none of the Met are in close contact with chromophore or with C355/Y106 (see below). We also have distinguished the Met residues, which are truly oxidized, from those that represent MS oxidation artefacts.

The oxidized methionine residues in Hypocrates (in red sticks) are located far away from the sensing domain (C355, Y106 indicated in orange sticks) and the chromophore (CR2 in orange sticks).

To give the kinetic interested reader of our manuscript more insight into the observed results we added the following to the main text:

“The negative y-intercept might be the result of several chlorination/oxidation events and some of these modifications might not directly be in the environment of the tryptophan, which was used as reporter amino acid for determining the second-order rate constant of NemR^{C106}.”

“By using the intensimetric cpYFP fluorescence as a read-out, we obtain a negative k_{off} for NaOCl and NaOBr and relatively high k_{fast} values, which could be explained by the observed chlorination/bromination on several amino acids and methionine oxidation by mass spectrometry (**Supplementary Table 3**). Especially Tyr106, located close to Cys355, might be involved in coupling the chlorination and bromination to the chromophore (**Supplementary Fig. 10**).”

“Next to Cys355 oxidation, our kinetic and mass spectrometric studies point in the direction of a potential alternative chlorination/bromination mechanism. A potential culprit residue in the active site of the sensing domain, located within hydrogen bonding distance of Cys355, is Tyr106 (**Supplementary Fig. 10**). This tyrosine is found to be chlorinated and brominated (**Supplementary**

Table 3). It is tempting to speculate that the formation of a chloramine on the flexible Lys424 (observed in two different side chain orientations) via Cys355 could catalyze Tyr106 carbon 3 and carbon 5 chlorination⁵⁷⁻⁵⁹. To facilitate this process, a deprotonated sulfur of Cys355, located N-terminally of a short helix, and the Nε2 of His63 would help to delocalize electrons in the direction of the OH of Tyr106 making both carbons on the Tyr phenol ring more susceptible for chlorination. This chlorination event takes place in a region of the sensing domain which could couple to the hydrogen bonding network surrounding the chromophore resulting in a reversible ratiometric response.”

We have also expanded the **Supplementary Figure 14** (now, it became Supplementary Fig. 10), in which we now also show Y106 and its relative position towards C355, H63, and K424.

We also included a new **Supplementary Table 3** showing the mass spectrometric results for all analyzed samples and uploaded the raw data to ProteomeXchange with identifier PXD029624. Reviewer account details:

Username: reviewer_pxd029624@ebi.ac.uk

Password: MMXpABSN

Supplementary Table 3: Mass spectrometry summary table. The percentages are calculated based on the number of peptide spectral matches (#PSM) for the chlorination/bromination events and based on Area Under the Curve (AUC) for the Cys and Met oxidations. The methionine oxidation was corrected for mass spectrometric oxidation artefacts using retention times. The raw data are available via ProteomeXchange with identifier PXD029624.

	control	NaOCl	NCT	NaOBr	HOSCN
chlorination	H232 (16%)	H232 (25%)	H232 (20%)	Y112 (3%)	H232 (28%)
		Y350 (13%)	Y106 (4%)		
		Y246 (7%)			
		Y198 (7%)			
		Y106 (7%)			
bromination				H438 (5%)	
				Y246 (10%)	
				Y299 (4%)	
				Y198 (4%)	
				Y106 (5%)	
di/tri-oxidation	C355 (7%)	C355 (76%)	C355 (98%)	C355 (57%)	C355 (5%)
oxidation	M59 (3%)	M59 (44%)	M59 (12%)	M59 (51%)	M59 (7%)
	M371 (3%)	M371 (36%)	M371 (9%)	M371 (24%)	M371 (6%)

M295 (3%)	M295 (4%)	M295 (2%)	M295 (5%)	M295 (5%)
M179 (0.7%)	M179 (35%)	M179 (24%)	M179 (24%)	M179 (5%)
M29 (8%)	M29 (76%)	M29 (38%)	M29 (67%)	M29 (14%)
M114 (7%)	M114 (69%)	M114 (76%)	M114 (91%)	M114 (7%)

The modified His, Tyr, and Met are surface exposed residues. Y106 and C355 are in red and located close together in the sensing domain of Hypocrates.

Reviewer #2

Comment: The issues raised by this reviewer have been all addressed in the revised paper. In addition, for the competition experiments in the rebuttal letter, the authors should notice that the "equation 2" from the Storkey et al. 2014 paper is based on the approximations that 1) the concentrations of both sensor and scavenger are treated as constant throughout the reaction; 2) the effect of reverse reactions is negligible. Nevertheless, this may not contradict with the conclusions made in the rebuttal letter since the difference of the calculated k value may not exceed one log, and the results of the competition experiments are not reported in the manuscript. Therefore, this reviewer considers that the current manuscript is appropriately revised.

Response: We agree with the remark of Reviewer 2 and want to emphasize again that it is unfortunately not possible to work under the conditions of the Storkey et al. 2014 paper. However, as we could not obtain pseudo first order measurements with a high concentration of biosensor (read-out window for the biosensor became too small to be reliable), we decided to work under V_{max} conditions, which are in a way also pseudo first order as the concentration of the oxidant will not change a lot and the differential equation will only become dependent on the biosensor concentration, which we kept at 0.5 μM . It will give us at least an assumption of whether the obtained stopped-flow k-values are correct, but as we cannot fulfill the conditions to apply 'equation 2', we decided not to add these values to the manuscript.

Finally, we have introduced several small modifications to the manuscript (corrected typos and updated information).

Reviewers' Comments:

Reviewer #1:

Remarks to the Author:

I appreciate the efforts the authors have gone to. The kinetics of the probe are obviously complex and difficult to sort out. I am happy with their explanation and their acknowledgement of uncertainty, and with the way they have addressed my point about negative intercepts.